# Bayesian Active Causal Discovery with Multi-Fidelity Experiments

**Zeyu Zhang** *
Gaoling School of Artificial Intelligence
Renmin University of China
zeyuzhang@ruc.edu.cn

**Chaozhuo Li** *
Microsoft Research Asia
cli@microsoft.com

**Xu Chen** †
Gaoling School of Artificial Intelligence
Renmin University of China
xu.chen@ruc.edu.cn

**Xing Xie**
Microsoft Research Asia
xingx@microsoft.com

## Abstract

This paper studies the problem of active causal discovery when the experiments can be done based on multi-fidelity oracles, where higher fidelity experiments are more precise and expensive, while the lower ones are cheaper but less accurate. In this paper, we formally define the task of multi-fidelity active causal discovery, and design a probabilistic model for solving this problem. In specific, we first introduce a mutual-information based acquisition function to determine which variable should be intervened at which fidelity, and then a cascading model is proposed to capture the correlations between different fidelity oracles. Beyond the above basic framework, we also extend it to the batch intervention scenario. We find that the theoretical foundations behind the widely used and efficient greedy method do not hold in our problem. To solve this problem, we introduce a new concept called $\epsilon$-submodular, and design a constraint based fidelity model to theoretically validate the greedy method. We conduct extensive experiments to demonstrate the effectiveness of our model.

## 1 Introduction

Causal discovery aims to learn the causal structure of a set of variables, which is fundamental for many real-world applications, including health caring [1], education [2] drug discovery [3] and protein synthesis [4]. In general, causal structure learning is an NP-hard problem [5], and purely based on the observational datasets, one cannot identify the unique causal structure, where the best result is discovering its Markov equivalence class [6].

To more accurately identify the unique causal structure, a promising direction is active causal discovery (ACD), where the model is allowed to actively intervene the causal structure to query key information for orienting the causal relations between different variables. For example, to study the causal relations between the drugs and diseases, one can conduct clinical tests via selectively administering the medicines to the patients. The key of active causal discovery is how to design effective experiments when the total cost (*e.g.*, the number of experiments) is limited. To achieve this goal, recent years have witnessed many promising models. For example, Agrawal et al. [7] proposes to intervene on the variables which can orient as many as possible undirected edges. Tigas et al. [8]

---

*Co-first authors.

†Corresponding author.

37th Conference on Neural Information Processing Systems (NeurIPS 2023).

designs a mutual information based method to determine the variables and values to be intervened, and study both single and batch intervention scenarios.

While the above models have achieved remarkable successes, they only allow to query a single oracle (*e.g.*, the real causal structure) for the experiments[3]. However, in many real-world applications, the experiments can be done via different methods. For example, to investigate the drug-disease causal relations, in addition to the clinical tests, one can also build simulators to obtain the medicine effects on the patients [9]. Usually, each experimental method corresponds to a unique oracle, and different oracles have various fidelites. Higher-fidelity experiments are more accurate but expensive, for example, administering drugs to the real patients. Lower-fidelity experiments are cheaper but inaccurate, for example, using patient simulators. These different fidelity oracles may offer better cost-benefit trade-offs, which, however, cannot be handled by existing active causal discovery models.

To bridge the above gap, in this paper, we formally define the task of active causal discovery with multi-fidelity oracles, where the model has to actively select which variables and values to intervene at which fidelities. This task is non-trivial due to the following reasons: to begin with, because of the introduction of multi-fidelity oracles, the model has to strategically choose the lower-cost and informative enough experiments to uncover the real causal structure, which needs our special designs. Then, given the experiment results with different fidelities, how to infer the real causal structure is also not easy, since the experiment results can be not produced from the oracle corresponding to the real causal structure. In addition, in practice, an efficient experiment should allow simultaneously intervening multiple variables [10]. However, how to extend our model to the batch intervention scenario is still not clear.

To overcome the above challenges, we design a Bayesian active causal discovery model, which is composed of two components. The first one is a mutual information (MI) based acquisition function. It aims to select the interventional variables, values and fidelities which are more informative for the real causal structure. The second one is a cascading fidelity model. In specific, we first regard the highest fidelity oracle as the real causal structure, and then a cascading model is built to correlate different fidelity oracles, so that the experiment results at one fidelity can be leveraged to infer the oracle at another fidelity. To achieve more efficient experiments, we also extend our model to the batch intervention scenario. Previously, the greedy method is demonstrated to be an efficient and effective strategy for batch intervention [8]. However, we found that, by allowing multi-fidelity oracles, the theoretical foundations behind the greedy method do not hold. For alleviating this problem, we introduce a new concept called $\epsilon$-submodular, and design a constraint-based fidelity model to theoretically validate the greedy method.

The main contributions of this paper are summarized as follows: (1) we formally define the task of active causal discovery with multi-fidelity oracles, which, to our knowledge, is the first time in the field of causal discovery. (2) To solve the above task, we propose a Bayesian framework, which is composed of a mutual information based acquisition function and a cascading fidelity model. (3) To extend our framework to the batch intervention scenario, we introduce a constraint-based fidelity model, which provides theoretical guarantees for the efficient greedy method. (4) We conduct extensive experiments to demonstrate the effectiveness of our model.

## 2 Preliminaries

### 2.1 Structure Causal Model

Structure causal model (SCM) is an effective language for describing and learning the causal relations between different random variables [11]. Usually, SCM is composed of a causal graph and a set of structure equation models (SEM).

For the causal graph, we denote it by $G = \langle V, \mathbf{E} \rangle$, where $V$ is the node set, and $\mathbf{E}$ is the adjacency matrix. Each node in $V$ corresponds to a variable. Suppose there are $d$ variables in our studied problem, then we use $X_V = [X_1, X_2, \ldots, X_d]$ to denote the variable set. The adjacency matrix $\mathbf{E} \in \{0, 1\}^{d \times d}$ describes the causal relations between different variables. $\mathbf{E}_{ij} = 1$ means that $X_i$ is a

---

[3]In the following, we may interchangeably use "oracle", "causal structure" and "structure causal model" to represent the underlying model for generating the results of the experiments.

parent of $X_j$, and there exists an edge from $X_i$ to $X_j$, while $\mathbf{E}_{ij} = 0$ indicates that there is no edge between $X_i$ and $X_j$.

For the structure equation models, we denote them by $\boldsymbol{g} = \{g_1, g_2, ..., g_d\}$, where each $g_i$ quantitatively describes the relation between $X_i$ and its parents. Formally, we implement $F$ with the commonly used additive noise models (ANM) [12], that is:

$$X_i = g_i(pa(i); \gamma_i) + \epsilon_i, \quad \epsilon_i \sim \mathcal{N}(0, \sigma_i^2), \tag{1}$$

where $\gamma_i$ is the parameter set of $g_i$, and the noise term $\epsilon_i$ follows the Gaussian distribution with $\sigma_i^2$ as the variance. We denote the complete parameter set of $\boldsymbol{g}$ as $\boldsymbol{\theta} = \{\boldsymbol{\gamma}, \boldsymbol{\sigma}\}$, where $\boldsymbol{\gamma} = \{\gamma_1, \gamma_2, \ldots, \gamma_d\}$ and $\boldsymbol{\sigma} = \{\sigma_1, \sigma_2, \ldots, \sigma_d\}$. Noted that, given the above equation, we can easily derive the distribution of $X_V$, that is, $p(X_V) = \prod_{i=1}^d \mathcal{N}(g_i(pa(i); \gamma_i), \sigma_i^2)$.

Based on the above formulation, given an observational dataset $D = \{\boldsymbol{x}_k\}_{k=1}^N \sim p(X_V)$, causal discovery aims to learn the adjacency matrix $\mathbf{E}$, or more generally, simultaneously identify $\mathbf{E}$ and the SEM parameter $\boldsymbol{\theta}$. In this paper, we focus on the general case, and denote $\boldsymbol{\phi} = (\boldsymbol{\theta}, \mathbf{E})$.

## 2.2 Active Causal Discovery

Previous work has demonstrated that, purely based on the observational dataset, the real causal graph can only be identifiable to its Markov equivalence class. Active causal discovery holds the promise of identifying more accurate causal graph via designing interventional experiments.

Formally, an interventional experiment is represented by $\boldsymbol{e} = \{(j, v)\}$, which means cutting all the causal relations pointing to $X_j$, and fixing the value of $X_j$ as $v$. In causal learning, the experiment $\boldsymbol{e}$ can also be represented by $\mathrm{do}(X_j = v)$. Obviously, the distribution of $X_V$ is changed after the experiment $\boldsymbol{e}$, and we denote the experiment-induced distribution by $p(X_V | \mathrm{do}(X_j = v))$. In practice, we cannot access the implementation of $p$, but can only observe the experiment result sampled from $p(X_V | \mathrm{do}(X_j = v))$. The key of active causal discovery is to design a series of experiments within limited budgets, such that the results can be better leveraged to identify $\boldsymbol{\phi}$.

## 2.3 Multi-Fidelity Active Causal Discovery

Existing ACD models mostly obtain the experiment results via interacting with the real SCM. However, in practice, the experiments can be done in different ways (*e.g.*, real clinical tests or patient simulators). Each type of experiment corresponds to an underlying oracle, which produces the results of the experiments. Different oracles may provide better cost-benefit trade-offs for the experiment designs, which are failed to be considered by the previous work.

Formally, suppose we have $M$ oracles, and the parameters of the $i$th oracle is denoted by $\boldsymbol{\phi}_i$. Let the experiment cost of the $i$th oracle be $\lambda_i$, and without loss of generality, we assume $\lambda_1 \leq \lambda_2 \leq \cdots \leq \lambda_M$. Intuitively, if an oracle is more accurate (*i.e.*, has higher fidelity), then it should be more expensive[4]. We regard the real SCM as the most accurate oracle, thus we set $\boldsymbol{\phi}_M = \boldsymbol{\phi}$. We denote all the oracle parameters and costs as $\boldsymbol{\Phi} = \{\boldsymbol{\phi}_1, \boldsymbol{\phi}_2, ..., \boldsymbol{\phi}_M\}$ and $\boldsymbol{\Lambda} = \{\lambda_1, \lambda_2, ..., \lambda_M\}$, respectively.

Due to the introduction of multi-fidelity oracles, the experiment in traditional active causal discovery is extended to be a triplet $\boldsymbol{e} = \{(j, v), m\}$, where in addition to the intervention pair $(j, v)$, the fidelity $m$ should also be considered in the experiment design. We define the dataset for model training as $D = \{\boldsymbol{e}_t, \boldsymbol{x}_t\}_{t=1}^T$, where $\boldsymbol{e}_t = \{(j, v), m\}$ indicates the distribution for generating $\boldsymbol{x}_t$. In general, $\{(j, v), m\}$ means that $\boldsymbol{x}_t$ is generated from $p_m(X_V | \mathrm{do}(X_j = v))$, which is induced by $\boldsymbol{\phi}_m$ and the intervention $(j, v)$. If $(j, v) = \emptyset$, then $\boldsymbol{x}_t$ is an observational data, which is sampled from $\boldsymbol{\phi}_m$ without any intervention. Finally, we define the task of multi-fidelity active causal discovery as follows:

**Definition 1** (Multi-Fidelity Active Causal Discovery (MFACD)). Given $M$ oracles with different costs $\boldsymbol{\Lambda}$, we need to design a model $f$, which can strategically determine the intervention pair $(j, v)$ and fidelity $m$ to achieve better cost-benefit trade-off in terms of identifying the real SCM $\boldsymbol{\phi}_M$.

---

[4]Noted that if some oracle costs more than another one with higher fidelity, then this oracle is useless, since one may always choose to query the higher fidelity oracle.

# 3 The Licence Model

For solving the task of MFACD, we design a Bayesian framework called Licence (Multi-fidelity active learning for causal discovery), which is composed of two components. The first one is an acquisition function based on mutual information, which is responsible for designing the experiments. The second one is a cascaded fidelity model, which is designed to capture the correlation between different $\phi_i$'s. The experiment results obtained from the first component are leveraged to update the fidelity model, and $\phi_M$ in the fidelity model is the finally predicted result. At last, we introduce how to extend our model to the batch intervention scenario.

## 3.1 MI-based Acquisition Function

Intuitively, a better experiment should leverage little cost to reveal as much as possible information about the real SCM $\phi_M$. Thus, we design the following acquisition function:

$$f(j, v, m) = \frac{I(\boldsymbol{x}; \boldsymbol{\phi}_M | \boldsymbol{e}, D)}{\lambda_m},$$

where $\boldsymbol{e} = \{(j, v), m\}$ is the experiment to be designed. $D$ is the dataset already collected, which will be enlarged after each experiment. $I(\boldsymbol{x}; \boldsymbol{\phi}_M | \boldsymbol{e}, D)$ is the mutual information, which indicates that if we conduct experiment $\boldsymbol{e}$, then how much information the experiment result may share with the target parameter $\boldsymbol{\phi}_M$. Obviously, we should select $\boldsymbol{e}$ which can lead to larger $I(\boldsymbol{x}; \boldsymbol{\phi}_M | \boldsymbol{e}, D)$. $\lambda_m$ is the cost of $\boldsymbol{e}$. By dividing $I(\boldsymbol{x}; \boldsymbol{\phi}_M | \boldsymbol{e}, D)$ with $\lambda_m$, we trade-off the experiment informativeness and cost. To determine $(j, v, m)$, we derive an estimator for $f(j, v, m)$ following the idea of Bayesian Active Learning by Disagreement (BALD) [13], that is:

$$
\begin{aligned}
f(j, v, m) &= \frac{H(\boldsymbol{x}|\boldsymbol{e}, D) - H(\boldsymbol{x}|\boldsymbol{\phi}_M, \boldsymbol{e}, D)}{\lambda_m} \\
&= \frac{-\mathbb{E}_{p(\boldsymbol{x}|\boldsymbol{e},D)}\left[\log \mathbb{E}_{p(\boldsymbol{\phi}_m|\boldsymbol{e},D)}[p(\boldsymbol{x}|\boldsymbol{e}, \boldsymbol{\phi}_m)]\right] + \mathbb{E}_{p(\boldsymbol{\phi}_M|D)}\left[\mathbb{E}_{p(\boldsymbol{x}|\boldsymbol{e},\boldsymbol{\phi}_M)}[\log p(\boldsymbol{x}|\boldsymbol{e}, \boldsymbol{\phi}_M)]\right]}{\lambda_m},
\end{aligned}
\tag{2}
$$

where $p(\boldsymbol{x}|\boldsymbol{e}, D)$ and $p(\boldsymbol{\phi}_m|\boldsymbol{e}, D)$ correspond to the distributions of $\boldsymbol{x}$ and $\boldsymbol{\phi}_m$ after observing $D$ under the intervention $(j, v)$. $p(\boldsymbol{\phi}_M|D)$ is the posterior of $\boldsymbol{\phi}_M$ after observing $D$. $p(\boldsymbol{x}|\boldsymbol{e}, \boldsymbol{\phi}_m)$ is the probability of $\boldsymbol{x}$ based on $\boldsymbol{\phi}_m$ intervened by $(j, v)$. We approximate the expectation operator based Monte Carlo sampling. The detailed derivation process can be seen in the appendix.

Obviously, determining the best $(j, v, m)$ equals to solving the following problem:

$$\{j^*, v^*, m^*\} = \arg \max_{\{j,v,m\}} f(j, v, m). \tag{3}$$

In our task, the intervention value $v$ is continuous. Following the previous work, we firstly learn the optimal $v$ for each $(j, m)$ pair based on Bayesian optimization (BO). Then, we compare all the results, and select the solution which can lead to the largest $f(j, v, m)$. We present the detailed Bayesian optimization process in the appendix. It should be noted that one can also leverage more advanced BO methods for jointly learning $(j, v, m)$ [14]. However, we do not find significant performance improvements by these models.

## 3.2 Cascaded Fidelity Model

Intuitively, the experiment results from different oracles may share common information. The samples at one fidelity may help to infer the oracles at other fidelities. To capture the correlations between different oracles, we build a cascaded probabilistic model (see Figure 1), where the oracles with different fidelities are successively connected, and the observed samples are only determined by their corresponding oracles.

Formally, to achieve more robust optimization, we first regard the discrete adjacency matrix $\mathbf{E}$ as the samples from Bernoulli distribution. In specific, we let $\mathbf{E}_{ij} \sim \text{Bernoulli}(\sigma(\mathbf{S}_{\cdot i}^T \mathbf{T}_{\cdot j}))$, where $\mathbf{S}, \mathbf{T} \in \mathbb{R}^{K \times d}$ ($K \ll d$) are two continuous matrices. $\mathbf{S}_{\cdot i}$ and $\mathbf{T}_{\cdot j}$ are the $i$th and $j$th columns of $\mathbf{S}$ and $\mathbf{T}$, respectively. By replacing $\mathbf{E}$ with $\mathbf{S}$ and $\mathbf{T}$, we revise the parameter set $\phi$ as $(\boldsymbol{\theta}, \mathbf{S}, \mathbf{T})$.

For each fidelity $m$, we assign a prior distribution of $\boldsymbol{\phi}_m$ given $\boldsymbol{\phi}_{m-1}$ as follows:

$$
\begin{aligned}
p(\boldsymbol{\phi}_1) &\propto e^{-\beta \cdot f(\mathbf{S}_1, \mathbf{T}_1)} \cdot \mathcal{N}(\mathbf{0}, \boldsymbol{I}), \\
p(\boldsymbol{\phi}_m | \boldsymbol{\phi}_{m-1}) &\propto e^{-\beta \cdot f(\mathbf{S}_m, \mathbf{T}_m)} \cdot \mathcal{N}(\boldsymbol{a}\boldsymbol{\phi}_{m-1} + \boldsymbol{b}, \boldsymbol{\sigma}^2 \boldsymbol{I}), \quad m \geq 2,
\end{aligned}
\tag{4}
$$

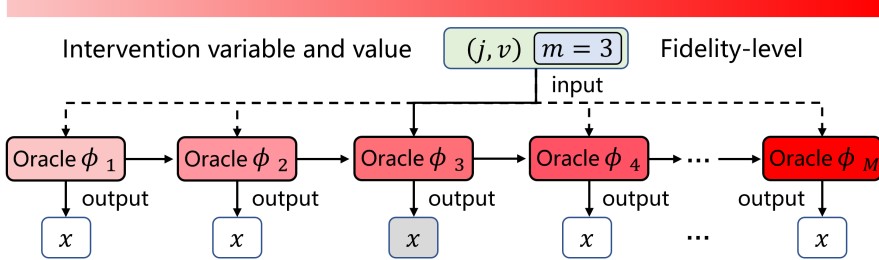

Figure 1: The cascaded probabilistic model is shown above. Different fidelity oracles are successively connected, and the observed samples are only determined by their corresponding oracles.

where we add subscript $m$ to indicate different fidelities. $f(\mathbf{S}_m, \mathbf{T}_m) = \mathbb{E}_{p(\mathbf{E}|\mathbf{S}_m, \mathbf{T}_m)}[\lambda_1 \cdot \{\text{trace}(e^{\mathbf{E}}) - d\} + \lambda_2 \cdot ||\mathbf{E}||_1]$ is a regularizer to encourage $\mathbf{E}$ to be a sparse and directed acyclic graph [15]. $\lambda_1$, $\lambda_2$, $\boldsymbol{a}$, $\boldsymbol{b}$ and $\boldsymbol{\sigma}^2$ are hyper-parameters.

Since the real SCM is $\boldsymbol{\phi}_M$, we need to infer the posterior $p(\boldsymbol{\phi}_M|D)$, where $D$ is the initially observational dataset or experiment results. Directly computing $p(\boldsymbol{\phi}_M|D)$ is not easy, since the dataset $D$ may contain samples from different fidelity oracles. To solve this problem, we first obtain the joint distribution $p(\boldsymbol{\Phi}|D)$, where $\boldsymbol{\Phi} = \{\boldsymbol{\phi}_1, \boldsymbol{\phi}_2, ..., \boldsymbol{\phi}_M\}$ is the collection of all the oracle parameters. Then we derive $p(\boldsymbol{\phi}_M|D)$ by marginalizing out $\{\boldsymbol{\phi}_1, \boldsymbol{\phi}_2, ..., \boldsymbol{\phi}_{M-1}\}$.

To efficiently compute and sample from $p(\boldsymbol{\Phi}|D)$, we introduce a variational approximator $q(\boldsymbol{\Phi})$, which is specified as follows:

$$q(\boldsymbol{\phi}_1) \sim \mathcal{N}(\mathbf{0}, \boldsymbol{I}),$$
$$q_{\psi_m}(\boldsymbol{\phi}_m|\boldsymbol{\phi}_{m-1}) \sim \mathcal{N}(\boldsymbol{c}_m\boldsymbol{\phi}_{m-1} + \boldsymbol{d}_m, \boldsymbol{\sigma}_m^2\boldsymbol{I}), \quad m \geq 2,$$
$$q(\boldsymbol{\Phi}) = q(\boldsymbol{\phi}_1) \prod_{m=2}^{M} q(\boldsymbol{\phi}_m|\boldsymbol{\phi}_{m-1}), \tag{5}$$

where $\psi_m = \{\boldsymbol{c}_m, \boldsymbol{d}_m, \boldsymbol{\sigma}_m\}$ is the set of learnable parameters, and we denote $\boldsymbol{\Psi} = \{\psi_m\}_{m=2}^{M}$. According to the theory of variational inference [16], we maximize the following evidence lower bound (ELBO) [16] to learn $\boldsymbol{\Psi}$:

$$\begin{aligned} \text{ELBO} &= \mathbb{E}_{\boldsymbol{\Phi} \sim q(\boldsymbol{\Phi})}\left[\log p(D|\boldsymbol{\Phi}) - \text{KL}(q(\boldsymbol{\Phi})||p(\boldsymbol{\Phi}))\right], \\ &= \mathbb{E}_{\boldsymbol{\Phi} \sim q(\boldsymbol{\Phi})}\left[\log p(D|\boldsymbol{\Phi}) - \log q(\boldsymbol{\Phi}) + \log p(\boldsymbol{\Phi})\right], \end{aligned} \tag{6}$$

where the likelihood $p(D|\boldsymbol{\Phi})$ can be easily obtained based on equation (1). More detailed derivation on the ELBO can be seen in the appendix. Once we have learned $\boldsymbol{\Psi}$, the posterior $p(\boldsymbol{\Phi}|D)$ is approximated by $q(\boldsymbol{\Phi})$, and further, we have the following theory:

**Theorem 1.** *If $p(\boldsymbol{\Phi}|D) \approx q(\boldsymbol{\Phi})$, then for any variable sets $A, B \subseteq \boldsymbol{\Phi}$, $p(A|B, D) \approx q(A|B)$. As a special case $p(\boldsymbol{\phi}_M|D) \approx q(\boldsymbol{\phi}_M)$.*

The proof of this theory is immediate, since

$$\begin{aligned} p(A|B, D) &= \frac{p(A, B|D)}{p(B|D)} = \frac{\int_{\boldsymbol{\Phi}_{-A-B}} p(\boldsymbol{\Phi}|D)\mathrm{d}\boldsymbol{\Phi}_{-A-B}}{\int_{\boldsymbol{\Phi}_{-B}} p(\boldsymbol{\Phi}|D)\mathrm{d}\boldsymbol{\Phi}_{-B}} \approx \frac{\int_{\boldsymbol{\Phi}_{-A-B}} q(\boldsymbol{\Phi})\mathrm{d}\boldsymbol{\Phi}_{-A-B}}{\int_{\boldsymbol{\Phi}_{-B}} q(\boldsymbol{\Phi})\mathrm{d}\boldsymbol{\Phi}_{-B}} \\ &= \frac{q(A, B)}{q(B)} = q(A|B) \end{aligned}$$

Let $A = \boldsymbol{\phi}_M$ and $B = \emptyset$, we have $p(\boldsymbol{\phi}_M|D) \approx q(\boldsymbol{\phi}_M)$. Based on this theory, we leverage $q$ to replace $p$ in (2) for easy sampling.

*Remark.* According to the specification of $q_{\psi_m}(\boldsymbol{\phi}_m|\boldsymbol{\phi}_{m-1})$, we can easily demonstrate that $q(\boldsymbol{\Phi})$ follows Gaussian distribution. Such property enables us to use the reparameterization trick [17] to relate $\boldsymbol{\Phi}$ with $\boldsymbol{\Psi}$. Different from traditional variational models, in our objective, the adjacency matrix

$\boldsymbol{E}$ in $f(\mathbf{S}_m, \mathbf{T}_m)$ is sampled from a discrete distribution, which cuts down the back-propagation signal. To solve this problem, we leverage gumbel-softmax to further associate $\boldsymbol{E}$ with $(\mathbf{S}_m, \mathbf{T}_m) \in \boldsymbol{\Phi}$. Since $\boldsymbol{\Phi}$ can be further represented by $\boldsymbol{\Psi}$, all the variables in (6) can be reparameterized by $\boldsymbol{\Psi}$, which enables us to optimize it in an end to end manner. We present the detailed derivation process and the complete learning algorithm in the appendix.

### 3.3 Extension to Batch Intervention

In practice, simultaneously intervening multiple variables can be more efficient due to the lower frequency on interacting with the oracles. However, under the setting of batch intervention, the candidate intervention space exponentially increases with respect to the number of targets. Suppose we need to select $c$ out of $d$ variables for intervention, then the size of the candidate space is $d^c$. Previous work found that the greedy strategy is a both efficient and effective method for the batch intervention scenario [8]. From the efficiency perspective, the greedy method only need to search in a $cd$-sized candidate space. From the effectiveness perspective, people demonstrate that the mutual information obtained by the greedy strategy is not worse than that of the optimal solution multiplied by $(1 - \frac{1}{e})$ [18]. In the following, we introduce how to extend our model to the batch intervention scenario, and leverage the greedy strategy to solve our task.

**Objective for batch intervention in MFACD**. We use the following objective for batch intervention scenario:

$$\arg\max_{\{\boldsymbol{e}_i\}_{i=1}^n} I(\{\boldsymbol{x}_i\}_{i=1}^n; \boldsymbol{\phi}_M | \{\boldsymbol{e}_i\}_{i=1}^n, D),$$

$$\text{s.t.} \sum_{i=1}^n \lambda_i \leq C, \tag{7}$$

where $\boldsymbol{e}_i = \{(j_i, v_i), m_i\}$ is an experiment, $\boldsymbol{x}_i$ is the observed sample from the experiment $\boldsymbol{e}_i$, that is, $\boldsymbol{x}_i \sim p_{m_i}(X_V | do(X_{j_i} = v_i))$. $\lambda_i$ is the cost of experiment $\boldsymbol{e}_i$. This objective aims to design a series of experiments with budget C, which can reveal the information about $\boldsymbol{\phi}_M$ as much as possible. The number of intervention targets $n$ is not a fixed value, which is constrained by the total budget.

**The greedy method for MFACD**. The greedy method designs each experiment independently, and at each step, it selects the experiment which can maximize the average information gain. Following [19], the $k$th experiment is determined based on the following objective:

$$\arg\max_{\boldsymbol{e}_k} \frac{I(\{\boldsymbol{x}_i\}_{i=1}^{k-1} \cup \boldsymbol{x}_k; \boldsymbol{\phi}_M | \{\boldsymbol{e}_i\}_{i=1}^{k-1} \cup \boldsymbol{e}_k, D) - I(\{\boldsymbol{x}_i\}_{i=1}^{k-1}; \boldsymbol{\phi}_M | \{\boldsymbol{e}_i\}_{i=1}^{k-1}, D))}{\lambda_m}$$

$$s.t. \sum_{m=1}^{k-1} \lambda_m + \lambda_k \leq C, \tag{8}$$

where $\{\boldsymbol{e}_i\}_{i=1}^{k-1}$ is the previously designed experiments, and is fixed when learning $\boldsymbol{e}_k$.

**What's wrong with the greedy method**. The theoretical foundations of the greedy method is demonstrated by the previous work [19] as follows:

**Theorem 2.** *If $I(\boldsymbol{x}; \boldsymbol{\phi}_M | \boldsymbol{e}, D)$ is (1) submodular and (2) non-decreasing, then*

$$I(\{\boldsymbol{x}_i^g\}_{i=1}^n; \boldsymbol{\phi}_M | \{\boldsymbol{e}_i^g\}_{i=1}^n, D) \geq (1 - \frac{1}{e}) I(\{\boldsymbol{x}_i^*\}_{i=1}^n; \boldsymbol{\phi}_M | \{\boldsymbol{e}_i^*\}_{i=1}^n, D), \tag{9}$$

*where $\{\boldsymbol{e}_i^g\}_{i=1}^n$ is the solution obtained from the greedy method, and $\{\boldsymbol{e}_i^*\}_{i=1}^n$ is the optimal solution.*

However, due to the introduction of multi-fidelity, in our model, $I(\boldsymbol{x}; \boldsymbol{\phi}_M | \boldsymbol{e}, D)$ is actually not submodular. More specifically, in the proof of submodular, two samples $\boldsymbol{x}_s$ and $\boldsymbol{x}_t$ has to be independent given $\boldsymbol{\phi}_M$ (*e.g.*, see B.4 in [8]). In the single-fidelity setting, this requirement naturally holds, since $\boldsymbol{\phi}_M$ is exactly the parameter used to sample $\boldsymbol{x}_s$ and $\boldsymbol{x}_t$. However, when we introduce multi-fidelity, $\boldsymbol{x}_s$ and $\boldsymbol{x}_t$ may not independent given $\boldsymbol{\phi}_M$, since they are only directly influenced by their own oracles (see Figure 1).

**An improved greedy method tailored to MFACD**. To alleviate the above problem, in this section we design an improved greedy method tailored to our task. In specific, we first define several new concepts, and then build theories based on these concepts to inspire our model designs.

**Definition 2** ($\epsilon$-independent). For random variables $A$, $B$ and $C$, if their mutual information satisfy $I(A; B|C) \leq \epsilon$, then we say $A$ and $B$ are $\epsilon$-independent given $C$.

**Definition 3** ($\epsilon$-submodular). Suppose $f(\cdot)$ is a set function on $\Omega$. If for any $A, B \subseteq \Omega$, $A \subseteq B$ and $x \in \Omega \backslash B$, $f(A \cup \{x\}) - f(A) \geq f(B \cup \{x\}) - f(B) - \epsilon$, then we say $f$ is $\epsilon$-submodular on $\Omega$.

Based on the above two definitions, we have:

**Theorem 3.** *For any two experiments $e_s$ and $e_t$, if the corresponding samples $x_s$ and $x_t$ are $\epsilon$-independent given $\phi_M, \{e_s, e_t\}$ and $D$, then $I(\cdot; \phi_M|\cdot, D)$ is $\epsilon$-submodular.*

**Theorem 4.** *If $I(\cdot; \phi_M|\cdot, D)$ is $\epsilon$-submodular on $X$ and non-decrease, for any $i, j$, $\frac{\lambda_i}{\lambda_j} \leq B_\lambda$, then*

$$I(\{x_i^g\}_{i=1}^n; \phi_M|\{e_i^g\}_{i=1}^n, D) \geq (1 - e^{-\frac{1}{B_\lambda}})I(\{x_i^*\}_{i=1}^n; \phi_M|\{e_i^*\}_{i=1}^n, D) - B,$$

*where $B = \frac{\epsilon}{B_\lambda} \cdot \sum_{i=1}^n (1 - \frac{1}{B_\lambda n})^{i-1}$.*

Following this theory, we improve objective (6) to a constaint-based ELBO as follows to capture the degree of independence between different experiment results:

$$\max \mathbb{E}_{\Phi \sim q(\Phi)} \left[ \log p(D|\Phi) - \log q(\Phi) + \log p(\Phi) \right],$$
$$\text{s.t.} \sum_{\{e_s, e_t\}} I(x_s; x_t|\phi_M, \{e_s, e_t\}, D) \leq \epsilon. \tag{10}$$

The proofs of the above theories are presented in the appendix, and similar to objective (2), we use Monte Carlo method to approximate the mutual information in objective (10).

## 4   Related Works

**Bayesian Active Causal Discovery.** Causal discovery [20–22] refers to recovering causality in a set of variables, especially trying to find a directed acyclic graph (DAG) that can represent the relationship between variables in a system. Active causal discovery was first proposed in [23, 24] with the assumption that the data is discrete-valued. In active causal discovery, the experimenter attempts to intervene on the variables in the system at each step, utilizes the interventional data to recover causal relation, and finally identifies the entire causal structure with minimal cost [25]. Many methods have been discussed in different settings over the past decades, including continuous linear Bayesian networks [26, 27], non-linear causal models [28], and large-scale causal models [8].

**Multi-fidelity Settings.** The fidelity commonly refers to how accurate the model or environment can be when providing information. Higher-fidelity models are more accuracy but cost much, while lower ones are less accurate but cheaper. In order to combine the strength of each model, multi-fidelity models are proposed to achieve an accurate model with lower costs. Multi-fidelity models can be divided into two main categories [29]: Multi-fidelity Surrogate Models (MFSM) [30–32] and Multi-fidelity Hierarchical Models (MFHM) [33–35]. In addition to optimization, multi-fidelity models can also be used for uncertainty propagation [36] and statistical inference [37]. Recent works also start to study the multi-fidelity settings when conducting Bayesian experiments [38–40], and adopt deep learning frameworks to solve corresponding problems.

Our paper makes a first step towards multi-fidelity active causal discovery, and solve the non-trivial challenges when combining the above two fields, which, to the best of our knowledge, is the first time in the causal inference domain.

## 5   Experiments

In this section, we conduct experiments to demonstrate the effectiveness of our model, where we focus on the following problems: (1) whether our model can achieve better performance than the previous ACD methods? (2) Whether the constraint in objective (10) in necessary? (3) How the DAG regularization coefficient influence the model performance? In the following, we first introduce the experiment setup, and then present and analyze the results.

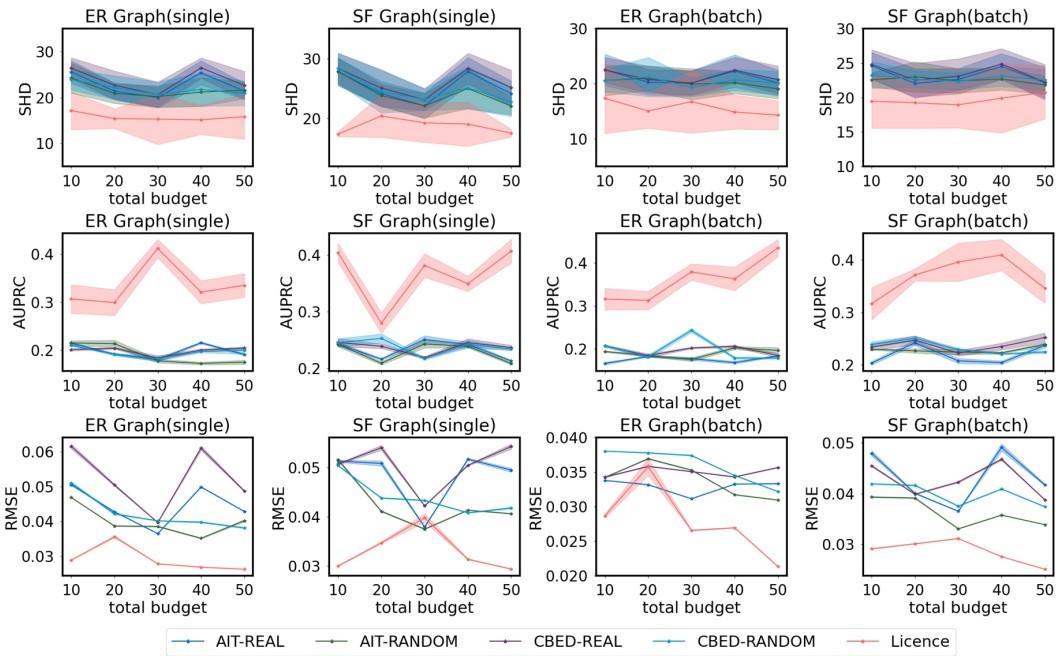

Figure 2: Results of the overall performance on different datasets and budgets. Lower SHD, RMSE or larger AUPRC indicate better performances. We conduct each experiment for ten times, and report the average performances and error bars.

## 5.1 Experimental Setup

We experiment with three commonly used causal discovery datasets, including Erdős-Rényi graph (**ER**) [41], Scale-Free graph (**SF**) [42] and DREAM [43]. To demonstrate the effectiveness of our model, we compare it with AIT [44] and CBED [8], which are the recent state-of-the-art models in this field. Since they cannot select different fidelities, we design two variants for each of the baseline, that is, **X-REAL** and **X-RANDOM**, which means that the model always interacts with the ground truth oracle $\phi_M$ or randomly select the oracles. Here "X" is AIT or CBED. For the evaluation metrics, we use SHD [45], AUPRC [46] and RMSE to evaluate different models. The first two metrics aim to evaluate the accuracy of the learned topological structure, and the last one measures the performance of functional relations. For single intervention, we first generate several observational samples to initialize the model. Then, we indicate a total intervention budget, and let the model interact with the causal graph with different fidelities until the budget runs out. For each interaction, the model will provide an intervention, and correspondingly obtain a sample from the oracles, which is used to update the model. Finally, the model outputs the estimated causal graph, which is leveraged to evaluate the performance. For batch intervention, we indicate the total budget for each intervention step, and the model determines $n$ interventions simultaneously, which are delivered to the oracles to obtain the samples. We present more detailed settings in the appendix.

## 5.2 Overall Performance

In this experiment, we evaluate the models under different total budgets, and we present the results on ER and SF datasets with 10 graph nodes. The experiments on DREAM and more nodes are presented in the appendix. From the results shown in Figure 2, we can see: as the total budget becomes larger, the performances of all the models tend to increase on both datasets. This is not surprising, since more experiment budgets enable us to conduct more interventions or query more accurate oracles, which can reveal more information about the ground truth and facilitate more accurate causal discovery. In most cases, our model can perform better than the baselines across different datasets, evaluation metrics and intervention budgets. These results demonstrate the effectiveness of our model. In specific, on the metrics of SHD, AUPRC and RMSE, our model can on average improve performance of the best baseline by about 27.74%, 82.35% and 22.74% on ER, and 17.69%, 60.27% and 21.62%

| (a) Batch Intervention with 30 Budget | | | |
|---|---|---|---|
| Model | SHD $\downarrow$ | AUPRC (%) $\uparrow$ | RMSE (%) $\downarrow$ |
| AIT-REAL | $20.10 \pm 2.26$ | $17.74 \pm 0.32$ | $3.11 \pm 0.00$ |
| AIT-RANDOM | $19.92 \pm 2.81$ | $17.57 \pm 0.43$ | $3.53 \pm 0.01$ |
| CBED-REAL | $20.12 \pm 2.17$ | $20.25 \pm 0.19$ | $3.51 \pm 0.00$ |
| CBED-RANDOM | $19.34 \pm 1.72$ | $24.32 \pm 0.47$ | $3.74 \pm 0.01$ |
| Licence (w/o reg) | $17.10 \pm 3.71$ | $32.55 \pm 1.81$ | $3.12 \pm 0.02$ |
| Licence | $\mathbf{16.75 \pm 5.68}$ | $\mathbf{37.94 \pm 1.90}$ | $\mathbf{2.66 \pm 0.01}$ |

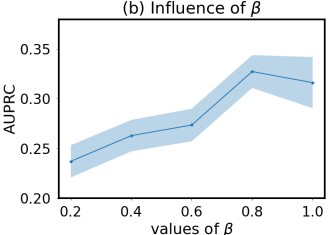

Figure 3: (a) The results of experiments on ER graph with 10 graph nodes under the batch intervention scenario. The average performance and error bars are provided. (b) The results of Licence model with different DAG regulation coefficient $\beta$'s. The experiment is conducted based on ER graph with 10 graph nodes.

on SF, respectively. If we look more carefully, we find that, for both AIT and CBED, randomly querying the oracles can sometimes perform better than always interacting with the ground truth oracles. This result suggests that lower fidelity oracles can be helpful to trade-off the performance and cost. However, the random method is still suboptimal, and designing more principled and tailored strategies to select the fidelities is necessary, which is evidenced by the lowered performance of "X-RANDOM" than our model.

### 5.3 Necessity of the Mutual Information Constraint in Objective (10)

In our model, the mutual information constraint in objective (10) aims to make the greedy method validate. In this section, we study whether it is necessary by experiments. To achieve this goal, we introduce a variant of our model **Licence (w/o reg)**, where we remove the mutual information constraint. We evaluate the models based on the dataset of ER with 10 graph nodes, and the total budget is set as 30. The results are presented in Figure 3(a). We can see, in some cases, "X-RANDOM" performs better than "X-REAL", which is consistent with the above experiments, and demonstrates that always querying the ground truth oracle may not lead to better performance under limited budget. By comparing our model with the variant Licence (w/o reg), we find that our model can consistently achieve better performances on all the evaluation metrics. In specific, Licence can improve the performance of Licence (w/o reg) by about 2.05%, 16.57% and 14.92% on SHD, AUPRC and RMSE respectively. These results demonstrate that the mutual information constraint is necessary in our model, which empirically verifies the theories proposed in section 3.3.

### 5.4 Influence of the DAG regularization coefficient

In this section, we analysis the influence of the DAG regulation coefficient $\beta$ in equation (4). The results are reported based on AUPRC. The coefficient $\beta$ reflects the importance of DAG regulation when updating the model. As $\beta$ increases, the DAGness is more emphasized for the causal graph. In this subsection, we conduct experiments for various $\beta$, ranging from 0.0 to 1.0, and the results are shown in Figure 3(b). We can see as $\beta$ increases, the performance of AUPRC improves as well, and peaks at $\beta = 0.8$. That is probably because lower $\beta$ will decrease the acyclic property of causal graphs, which is incompatible with the prior knowledge of true causal graphs. However, higher coefficient may also impact the optimization process, which leads to sub-optimal results. We think a trade-off between mild constraint for easy optimization and solid constraint for DAG property is supposed to take into consider for different tasks and settings.

## 6 Conclusion

This paper formally defines the task of active causal discovery with multi-fidelity oracles, which, to our knowledge, is the first time in the causal discovery domain. To solve this task, we propose a Bayesian framework, which is composed of a mutual information based acquisition function and a cascading fidelity model. We also extend our framework to the batch intervention scenario, and propose a constraint-based fidelity model to validate the efficient greedy method.

This paper actually makes an initial step toward considering different oracles in active causal discovery. There is much room left for improvement. To begin with, one can design more advanced batch intervention strategies, which can bypass the greedy method and does not need to introduce the mutual information constraint in the fidelity model. In addition, since the experiments in active causal discovery are conducted sequentially, and the former experiment results may influence the latter ones, it is interesting to consider the experiment designs as a Markov decision process, and leverage reinforcement learning to optimize the total information gains of all the experiments in a more principled manner.

## ACKNOWLEDGMENTS

This work is supported in part by National Key R&D Program of China (2022ZD0120103), National Natural Science Foundation of China (No. 62102420), Beijing Outstanding Young Scientist Program NO. BJJWZYJH012019100020098, Intelligent Social Governance Platform, Major Innovation & Planning Interdisciplinary Platform for the "Double-First Class" Initiative, Renmin University of China, Public Computing Cloud, Renmin University of China, fund for building world-class universities (disciplines) of Renmin University of China, Intelligent Social Governance Platform.

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

# Appendix

# A Monte Carlo Approximation for $f(j, v, m)$

## A.1 Derivation Process for $f(j, v, m)$

Considering that the mutual information is not directly tractable, we approximate $f(j, v, m)$ by:

$$f(j, v, m) = -\frac{1}{\lambda_m \cdot K_1 \cdot L_1} \sum_{k_1=1}^{K_1} \sum_{l_1=1}^{L_1} \log \left[ \frac{1}{C} \sum_{c_1=1}^{C_1} p(\boldsymbol{x}_m^{(k_1, l_1)} | \boldsymbol{\phi}_m^{(c_1)}, \boldsymbol{e}) \right]$$

$$+ \frac{1}{\lambda_m \cdot K_2 \cdot L_2 \cdot C_2} \sum_{k_2=1}^{K_2} \sum_{l_2=1}^{L_2} \sum_{c_2=1}^{C_2} \log \left[ p(\boldsymbol{x}_m^{(k_2, l_2, c_2)} | \boldsymbol{\phi}_M^{(k_2)}, \boldsymbol{e}) \right],$$

where $\boldsymbol{e} = \{(j, v), m\}$ is the experiment to be designed, $\boldsymbol{\phi}_m^{(c_1)}, \boldsymbol{\phi}_m^{(k_1)} \sim p(\boldsymbol{\phi}_m | D)$, $\boldsymbol{x}_m^{(k_1, l_1)} \sim p(\boldsymbol{x} | \boldsymbol{\phi}_m^{(k_1)}, \boldsymbol{e})$, $\boldsymbol{\phi}_M^{(k_2)} \sim p(\boldsymbol{\phi}_M | D)$, $\boldsymbol{\phi}_m^{(k_2, l_2)} \sim p(\boldsymbol{\phi}_m | \boldsymbol{\phi}_M^{(k_2)}, D)$ and $\boldsymbol{x}_m^{(k_2, l_2, c_2)} \sim p(\boldsymbol{x} | \boldsymbol{\phi}_m^{(k_2, l_2)}, \boldsymbol{e})$.

We present the detailed approximation process as follows:

$$f(j, v, m) = \frac{1}{\lambda_m} I(\boldsymbol{x}; \boldsymbol{\phi}_M | \boldsymbol{e}, D)$$

$$= \frac{1}{\lambda_m} \left[ H(\boldsymbol{x} | \boldsymbol{e}, D) - H(\boldsymbol{x} | \boldsymbol{\phi}_M, \boldsymbol{e}, D) \right]$$

$$= \frac{1}{\lambda_m} \left[ -\mathbb{E}_{p(\boldsymbol{x}|\boldsymbol{e}, D)} \left[ \log p(\boldsymbol{x} | \boldsymbol{e}, D) + \mathbb{E}_{p(\boldsymbol{\phi}_M | D)} \left[ \mathbb{E}_{p(\boldsymbol{x}|\boldsymbol{\phi}_M, \boldsymbol{e})} \left[ \log p(\boldsymbol{x} | \boldsymbol{e}, \boldsymbol{\phi}_M) \right] \right] \right] \right]$$

$$= \underbrace{\frac{1}{\lambda_m} \left[ -\mathbb{E}_{p(\boldsymbol{x}|\boldsymbol{e}, D)} \left[ \log \mathbb{E}_{p(\boldsymbol{\phi}_m|\boldsymbol{e}, D)} \left[ p(\boldsymbol{x}|\boldsymbol{e}, \boldsymbol{\phi}_m) \right] \right] \right]}_{E}$$

$$+ \underbrace{\frac{1}{\lambda_m} \left[ \mathbb{E}_{p(\boldsymbol{\phi}_M | D)} \left[ \mathbb{E}_{p(\boldsymbol{x}|\boldsymbol{e}, \boldsymbol{\phi}_M)} \left[ \log p(\boldsymbol{x}|\boldsymbol{e}, \boldsymbol{\phi}_M) \right] \right] \right]}_{F}$$

For part $E$, we can estimate it by

$$E = -\frac{1}{\lambda_m \cdot K_1 \cdot L_1} \sum_{k_1=1}^{K_1} \sum_{l_1=1}^{L_1} \log \left[ \frac{1}{C} \sum_{c_1=1}^{C_1} p(\boldsymbol{x}_m^{(k_1, l_1)} | \boldsymbol{\phi}_m^{(c_1)}, \boldsymbol{e}) \right],$$

where for the first expectation on $p(\boldsymbol{\phi}_m | \boldsymbol{e}, D)$, we first sample $\boldsymbol{\phi}_m^{(k_1)}$ from $\boldsymbol{\phi}_m^{(k_1)} \sim p(\boldsymbol{\phi}_m | \boldsymbol{e}, D)$ for $K_1$ times, and then for each $\boldsymbol{\phi}_m^{(k_1)}$, we sample $\boldsymbol{x}_m^{(k_1, l_1)}$ from $\boldsymbol{x}_m^{(k_1, l_1)} \sim p(\boldsymbol{x} | \boldsymbol{\phi}_m^{(k_1)}, \boldsymbol{e})$ for $L_1$ times. For the second expectation on $p(\boldsymbol{\phi}_m | \boldsymbol{e}, D)$, we sample $\boldsymbol{\phi}_m^{(c_1)} \sim p(\boldsymbol{\phi}_m | \boldsymbol{e}, D)$ for $C_1$ times.

For part $F$, we have

$$F = \frac{1}{\lambda_m} \cdot \left[ \mathbb{E}_{p(\boldsymbol{\phi}_M | D)} \left[ \mathbb{E}_{p(\boldsymbol{x}|\boldsymbol{\phi}_M, \boldsymbol{e})} \left[ \log p(\boldsymbol{x}|\boldsymbol{\phi}_M, \boldsymbol{e}) \right] \right] \right]$$

$$= \frac{1}{\lambda_m} \cdot \left[ \mathbb{E}_{p(\boldsymbol{\phi}_M | D)} \left[ \int p(\boldsymbol{x}|\boldsymbol{\phi}_M, \boldsymbol{e}) \log p(\boldsymbol{x}|\boldsymbol{\phi}_M, \boldsymbol{e}) \, d\boldsymbol{x} \right] \right]$$

$$= \frac{1}{\lambda_m} \cdot \left[ \mathbb{E}_{p(\boldsymbol{\phi}_M | D)} \left[ \int \int p(\boldsymbol{x}|\boldsymbol{\phi}_m, \boldsymbol{e}) p(\boldsymbol{\phi}_m|\boldsymbol{\phi}_M) \, d\boldsymbol{\phi}_m \log p(\boldsymbol{x}|\boldsymbol{\phi}_M, \boldsymbol{e}) \, d\boldsymbol{x} \right] \right]$$

$$= \frac{1}{\lambda_m} \cdot \left[ \mathbb{E}_{p(\boldsymbol{\phi}_M | D, \boldsymbol{e})} \left[ \int \int p(\boldsymbol{x}|\boldsymbol{\phi}_m, \boldsymbol{e}) p(\boldsymbol{\phi}_m|\boldsymbol{\phi}_M) \log p(\boldsymbol{x}|\boldsymbol{\phi}_M, \boldsymbol{e}) \, d\boldsymbol{x} \, d\boldsymbol{\phi}_m \right] \right]$$

$$= \frac{1}{\lambda_m} \cdot \left[ \mathbb{E}_{p(\boldsymbol{\phi}_M | D, \boldsymbol{e})} \left[ \int \mathbb{E}_{p(\boldsymbol{x}|\boldsymbol{\phi}_m, \boldsymbol{e})} \left[ p(\boldsymbol{\phi}_m|\boldsymbol{\phi}_M) \log p(\boldsymbol{x}|\boldsymbol{\phi}_M, \boldsymbol{e}) \right] d\boldsymbol{\phi}_m \right] \right]$$

$$= \frac{1}{\lambda_m} \cdot \left[ \mathbb{E}_{p(\boldsymbol{\phi}_M | D, \boldsymbol{e})} \left[ \mathbb{E}_{p(\boldsymbol{\phi}_m|\boldsymbol{\phi}_M)} \left[ \mathbb{E}_{p(\boldsymbol{x}|\boldsymbol{\phi}_m, \boldsymbol{e})} \left[ \log p(\boldsymbol{x}|\boldsymbol{\phi}_M, \boldsymbol{e}) \right] \right] \right] \right].$$

It can be estimated by

$$\frac{1}{\lambda_m \cdot K_2 \cdot L_2 \cdot C_2} \sum_{k_2=1}^{K_2} \sum_{l_2=1}^{L_2} \sum_{c_2=1}^{C_2} \log \left[ p(\boldsymbol{x}_m^{(k_2, l_2, c_2)} | \boldsymbol{\phi}_M^{(k_2)}, \boldsymbol{e}) \right],$$

where for the expectation on $p(\phi_M | D, e)$, we sample $\phi_M^{(k_2)}$ from $\phi_M^{(k_2)} \sim p(\phi_M | e, D)$ for $K_2$ times. For the expectation on $p(\phi_m | \phi_M)$, for each $\phi_M^{(k_2)}$, we sample $\phi_m^{(k_2, l_2)}$ from $\phi_m^{(k_2, l_2)} \sim p(\phi_m | \phi_M^{(k_2)})$ for $L_2$ times. For the expectation on $p(\boldsymbol{x} | \phi_m, e)$, for each $\phi_M^{(k_2)}$ and $\phi_m^{(k_2, l_2)}$, we sample $\boldsymbol{x}_m^{(k_2, l_2, c_2)}$ from $\boldsymbol{x}_m^{(k_2, l_2, c_2)} \sim p(\boldsymbol{x} | \phi_m^{(k_2, l_2)}, e)$ for $C_2$ times.

Therefore, we can conclude that $f(j, v, m)$ can be estimated by

$$
f(j, v, m) = -\frac{1}{\lambda_m \cdot K_1 \cdot L_1} \sum_{k_1=1}^{K_1} \sum_{l_1=1}^{L_1} \log \left[ \frac{1}{C} \sum_{c_1=1}^{C_1} p(\boldsymbol{x}_m^{(k_1, l_1)} | \phi_m^{(c_1)}, e) \right]
$$

$$
+ \frac{1}{\lambda_m \cdot K_2 \cdot L_2 \cdot C_2} \sum_{k_2=1}^{K_2} \sum_{l_2=1}^{L_2} \sum_{c_2=1}^{C_2} \log \left[ p(\boldsymbol{x}_m^{(k_2, l_2, c_2)} | \phi_M^{(k_2)}, e) \right],
$$

where $\phi_m^{(c_1)}, \phi_m^{(k_1)} \sim p(\phi_m | D)$, $\boldsymbol{x}_m^{(k_1, l_1)} \sim p(\boldsymbol{x} | \phi_m^{(k_1)}, e)$, $\phi_M^{(k_2)} \sim p(\phi_M | D)$, $\phi_m^{(k_2, l_2)} \sim p(\phi_m | \phi_M^{(k_2)}, D)$ and $\boldsymbol{x}_m^{(k_2, l_2, c_2)} \sim p(\boldsymbol{x} | \phi_m^{(k_2, l_2)}, e)$.

Obviously, the above approximation of $f(j, v, m)$ only depends on $p(\phi_m | D)$, $p(\phi_m | \phi_M, D)$ and $p(\boldsymbol{x} | \phi_m, e)$. In the next, we show how to sample from them in Section A.2, A.3 and A.4, respectively.

### A.2  Sampling from $p(\phi_m | D)$

Basically, sampling from the posterior of "$p(\cdot | D)$" is not easy. To solve this problem, as mentioned in the main paper, we introduce a variational probability "$q$" to approximate "$p$". In specific, in order to sample from $p(\phi_m | D)$, we first obtain a sample $\phi_1$ from $\phi_1 \sim \mathcal{N}(\boldsymbol{0}, \boldsymbol{I})$, and then get $\phi_m$ from the distribution $q(\phi_m | \phi_1)$.

Since

$$
q(\phi_m | \phi_1) = \int_{\phi_{m-1}} \cdots \int_{\phi_2} q(\phi_m, \phi_{m-1}, \ldots, \phi_2 | \phi_1) \, d\phi_{m-1} \, \ldots \, d\phi_2.
$$

and

$$
q(\phi_m, \phi_{m-1}, \ldots, \phi_2 | \phi_1) = \prod_{i=2}^{m} q(\phi_i | \phi_{i-1})
$$

$$
q(\phi_i | \phi_{i-1}) = \mathcal{N}(\boldsymbol{c}_i \phi_{i-1} + \boldsymbol{d}_i, \sigma_i^2 \boldsymbol{I}),
$$

we have $q(\phi_m | \phi_1)$ is a Gaussian distribution, which is easy for sampling.

In our model, $\boldsymbol{c}_i$ and $\sigma_i^2 \boldsymbol{I}$ are diagonal matrices, which means that the dimensions in $\phi_i$ are independent with each other. We denote $\phi_i = [\phi_{i,1}, \phi_{i,2} \ldots \phi_{i,d}]$, where $\phi_{i,j}$ is the $j$th element of $\phi_i$. Then, we have

$$
q(\phi_m, \phi_{m-1}, \ldots, \phi_2 | \phi_1) = \prod_{j=1}^{d} q(\phi_{m,j}, \phi_{m-1,j}, \ldots, \phi_{2,j} | \phi_{1,j}).
$$

So our target can be converted to calculate the probability $q(\phi_{m,j}, \phi_{m-1,j}, \ldots, \phi_{2,j} | \phi_{1,j})$ for all dimensions $\forall 1 \leq j \leq d$. Let $c_{i,j}, d_{i,j}$ and $\sigma_{i,j}^2$ be the $j$th element of $\boldsymbol{c}_i, \boldsymbol{d}_i$ and $\sigma_i^2$, respectvely. We assume that $\sigma_{i,j} = \sqrt{c_{i-1,j}^2 + 1} \cdot \sigma_{i-1,j}$ $(i \geq 4)$ and $\sigma_{3,j} = \sigma_{2,j} = e$, where $e$ is the hyper-parameter. Suppose $\mu_{i,j}$ is the mean of the Gaussian distribution for $q(\phi_{i,j} | \phi_{i-1,j})$, that is,

$$
\mu_{i,j} = c_{i,j} \phi_{i-1,j} + d_{i,j} \ (i \geq 2),
$$

then, the approximated joint distribution can be represented as

$$
q(\phi_{m,j}, \phi_{m-1,j}, \ldots, \phi_{2,j} | \phi_{1,j}) = \prod_{i=2}^{m} q(\phi_{i,j} | \phi_{i-1,j})
$$

$$
= \prod_{i=2}^{m} \frac{1}{\sqrt{2\pi}\sigma_{i,j}} \cdot e^{\frac{-1}{2\sigma_{i,j}^2}(\phi_{i,j} - \mu_{i,j})^2}.
$$

Then, we integrate $\phi_{2,j}, \phi_{3,j}, \ldots, \phi_{m-1,j}$ sequentially to obtain $q(\phi_{m,j}|\phi_{1,j})$.

First of all, we integrate $\phi_{2,j}$ for the joint distribution, where we have:

$$
\begin{aligned}
&q(\phi_{m,j}, \phi_{m-1,j}, \ldots, \phi_{3,j}|\phi_{1,j}) \\
&= \int q(\phi_{m,j}, \phi_{m-1,j}, \ldots, \phi_{3,j}, \phi_{2,j}|\phi_{1,j}) \, d\phi_{2,j} \\
&= \int \prod_{i=2}^{m} \frac{1}{\sqrt{2\pi}\sigma_{i,j}} \cdot e^{\frac{-1}{2\sigma_{i,j}^2}(\phi_{i,j}-\mu_{i,j})^2} \, d\phi_{2,j} \\
&= \prod_{i=4}^{m} \frac{1}{\sqrt{2\pi}\sigma_{i,j}} \cdot e^{\frac{-1}{2\sigma_{i,j}^2}(\phi_{i,j}-\mu_{i,j})^2} \cdot \frac{1}{\sqrt{2\pi}\sigma_{3,j}} \cdot \frac{1}{\sqrt{2\pi}\sigma_{2,j}} \cdot \int e^{\frac{-1}{2\sigma_{3,j}^2}[\phi_{3,j}-(c_{3,j}\phi_{2,j}+d_{3,j})]^2} \cdot \\
&\quad e^{\frac{-1}{2\sigma_{2,j}^2}[\phi_{2,j}-(w_2\phi_{1,j}+d_{3,j})]^2} \, d\phi_{2,j}.
\end{aligned}
$$

Denote $\bar{c}_{2,j} = c_{2,j}$ and $\bar{d}_{2,j} = d_{2,j}$, and because of $\sigma_{3,j} = \sigma_{2,j}$, we have

$$
\begin{aligned}
&q(\phi_{m,j}, \phi_{m-1,j}, \ldots, \phi_{3,j}|\phi_{1,j}) \\
&= \frac{1}{\sqrt{2\pi}\sigma_{2,j}} \cdot \prod_{i=4}^{m} \frac{1}{\sqrt{2\pi}\sigma_{i,j}} \cdot e^{\frac{-1}{2\sigma_{i,j}^2}(\phi_{i,j}-\mu_{i,j})^2} \cdot \frac{1}{\sqrt{2\pi}\sigma_{3,j}} \int e^{\frac{-1}{2\sigma_{3,j}^2}[\phi_{3,j}-(c_{3,j}\phi_{2,j}+d_{3,j})]^2} \cdot \\
&\quad e^{\frac{-1}{2\sigma_{3,j}^2}\left[\phi_{2,j}-(\bar{c}_{2,j}\phi_{1,j}+\bar{d}_{2,j})\right]^2} \, d\phi_{2,j} \\
&= \frac{1}{\sqrt{2\pi}\sigma_{2,j}} \cdot \prod_{i=4}^{m} \frac{1}{\sqrt{2\pi}\sigma_{i,j}} \cdot e^{\frac{-1}{2\sigma_{i,j}^2}(\phi_{i,j}-\mu_{i,j})^2} \cdot \frac{1}{\sqrt{2\pi}\sigma_{3,j}} \int \cdot \\
&\quad e^{\frac{-1}{2\sigma_{3,j}^2} \underbrace{\left\{[\phi_{3,j} - (c_{3,j}\phi_{2,j}+d_{3,j})]^2 + \left[\phi_{2,j} - (\bar{c}_{2,j}\phi_{1,j}+\bar{d}_{2,j})\right]^2\right\}}_{S_1}} \, d\phi_{2,j}.
\end{aligned}
$$

For $S_1$, we have

$$
\begin{aligned}
S_1 &= [\phi_{3,j} - (c_{3,j}\phi_{2,j}+d_{3,j})]^2 + \left[\phi_{2,j} - (\bar{c}_{2,j}\phi_{1,j}+\bar{d}_{2,j})\right]^2 \\
&= \phi_{3,j}^2 + c_{3,j}^2\phi_{2,j}^2 + d_{3,j}^2 + 2d_{3,j}c_{3,j}\phi_{2,j} - 2c_{3,j}\phi_{3,j}\phi_{2,j} - 2d_{3,j}\phi_{3,j} + \phi_{2,j}^2 \\
&\quad + \bar{c}_{2,j}^2\phi_{1,j}^2 + \bar{d}_{2,j}^2 + 2\bar{d}_{2,j}\bar{c}_{2,j}\phi_{1,j} - 2\bar{c}_{2,j}\phi_{1,j} - 2d_{3,j}\phi_{3,j} \\
&= (c_{3,j}^2 + 1) \cdot \\
&\quad \left[\phi_{2,j}^2 + \frac{2(d_{3,j}c_{3,j} - c_{3,j}\phi_{3,j} - \bar{c}_{2,j}\phi_{1,j} - \bar{d}_{2,j})}{c_{3,j}^2+1} + \left(\frac{d_{3,j}c_{3,j} - c_{3,j}\phi_{3,j} - \bar{c}_{2,j}\phi_{1,j} - \bar{d}_{2,j}}{c_{3,j}^2+1}\right)^2\right] \\
&\quad - \frac{(d_{3,j}c_{3,j} - c_{3,j}\phi_{3,j} - \bar{c}_{2,j}\phi_{1,j} - \bar{d}_{2,j})^2}{c_{3,j}^2+1} + \phi_{3,j}^2 + \bar{c}_{2,j}^2\phi_{1,j}^2 + \bar{d}_{2,j}^2 + 2\bar{d}_{2,j}\bar{c}_{2,j}\phi_{1,j} - 2d_{3,j}\phi_{3,j} \\
&= (c_{3,j}^2 + 1) \cdot \left(\phi_{2,j} - \frac{c_{3,j}\phi_{3,j} + \bar{c}_{2,j}\phi_{1,j} + \bar{d}_{2,j} - d_{3,j}c_{3,j}}{c_{3,j}^2+1}\right)^2 \\
&\quad + \frac{c_{3,j}^2\phi_{3,j}^2 + \bar{c}_{2,j}^2c_{3,j}^2\phi_{1,j}^2 + \bar{d}_{2,j}^2c_{3,j}^2 + 2d_{3,j}\bar{c}_{2,j}c_{3,j}\phi_{1,j} - \bar{c}_{2,j}^2\phi_{1,j}^2 - \bar{d}_{2,j}^2 - 2\bar{d}_{2,j}\bar{c}_{2,j}\phi_{1,j}}{c_{3,j}^2+1} \\
&\quad + \frac{\phi_{3,j}^2 + \bar{c}_{2,j}^2\phi_{1,j}^2 + \bar{d}_{2,j}^2 + 2\bar{d}_{2,j}\bar{c}_{2,j}\phi_{1,j} - 2d_{3,j}\phi_{3,j} - c_{3,j}^2d_{3,j}^2 - c_{3,j}^2\phi_{3,j}^2 + 2c_{3,j}^2d_{3,j}\phi_{3,j}}{c_{3,j}^2+1} \\
&\quad + \frac{2c_{3,j}\bar{d}_{2,j}d_{3,j} - 2c_{3,j}\phi_{3,j}\bar{c}_{2,j}\phi_{1,j} - 2c_{3,j}\phi_{3,j}\bar{d}_{2,j} + 2\bar{d}_{2,j}\bar{c}_{2,j}c_{3,j}^2\phi_{1,j} - 2d_{3,j}c_{3,j}^2\phi_{3,j}}{c_{3,j}^2+1}.
\end{aligned}
$$

Then we have

$$
\begin{aligned}
S_1 =& (c_{3,j}^2 + 1) \cdot \left( \phi_{2,j} - \frac{c_{3,j}\phi_{3,j} + \bar{c}_{2,j}\phi_{1,j} + \bar{d}_{2,j} - d_{3,j}c_{3,j}}{c_{3,j}^2 + 1} \right)^2 \\
& + \frac{\phi_{3,j}^2 - 2(\bar{c}_{2,j}c_{3,j}\phi_{1,j} + \bar{d}_{2,j}c_{3,j} + d_{3,j})\phi_{3,j}}{c_{3,j}^2 + 1} \\
& + \frac{\left(\bar{c}_{2,j}c_{3,j}\phi_{1,j} + \bar{d}_{2,j}c_{3,j} + d_{3,j}\right)^2 - d_{3,j}^2 \cdot (c_{3,j}^2 + 1)}{c_{3,j}^2 + 1} \\
=& (c_{3,j}^2 + 1) \cdot \left( \phi_{2,j} - \frac{c_{3,j}\phi_{3,j} + \bar{c}_{2,j}\phi_{1,j} + \bar{d}_{2,j} - d_{3,j}c_{3,j}}{c_{3,j}^2 + 1} \right)^2 \\
& + \frac{1}{c_{3,j}^2 + 1} \cdot \left[ \phi_{3,j} - (c_{3,j}\bar{c}_{2,j}\phi_{1,j} + \bar{d}_{2,j}c_{3,j} + d_{3,j}) \right]^2 - d_{3,j}^2.
\end{aligned}
$$

Therefore we have

$$
\begin{aligned}
& q(\phi_{m,j}, \phi_{m-1,j}, \ldots, \phi_{3,j}|\phi_{1,j}) \\
=& \frac{1}{\sqrt{2\pi}\sigma_{2,j}} \cdot \prod_{i=4}^{m} \frac{1}{\sqrt{2\pi}\sigma_{i,j}} \cdot e^{\frac{-1}{2\sigma_{i,j}^2}(\phi_{i,j} - \mu_{i,j})^2} \cdot \\
& \underbrace{\left[ \int \frac{\sqrt{c_{3,j}^2 + 1}}{\sqrt{2\pi}\sigma_{3,j}} e^{\frac{-(c_{3,j}^2 + 1)}{2\sigma_{3,j}^2}\left( \phi_{2,j} - \frac{c_{3,j}\phi_{3,j} + \bar{c}_{2,j}\phi_{1,j} + \bar{d}_{2,j} - d_{3,j}c_{3,j}}{c_{3,j}^2 + 1} \right)^2} d\phi_{2,j} \right]}_{S_2} \cdot \\
& e^{\frac{-1}{2\sigma_{3,j}^2 \cdot (c_{3,j}^2 + 1)}\left[ \phi_{3,j} - (c_{3,j}\bar{c}_{2,j}\phi_{1,j} + \bar{d}_{2,j}c_{3,j} + d_{3,j}) \right]^2} \cdot e^{\frac{d_{3,j}^2}{2\sigma_{3,j}^2}} \cdot \frac{1}{\sqrt{c_{3,j}^2 + 1}}
\end{aligned}
$$

The $S_2$ part is the integration form of $\phi_{2,j} \sim \mathcal{N}(\frac{c_{3,j}\phi_{3,j} + \bar{c}_{2,j}\phi_{1,j} + \bar{d}_{2,j} - d_{3,j}c_{3,j}}{c_{3,j}^2 + 1}, \frac{\sigma_{3,j}^2}{c_{3,j}^2 + 1})$, which is equal to 1, so we have

$$
\begin{aligned}
& q(\phi_{m,j}, \phi_{m-1,j}, \ldots, \phi_{3,j}|\phi_{1,j}) \\
=& \frac{1}{\sqrt{2\pi}\sigma_{2,j}} \cdot \prod_{i=4}^{m} \frac{1}{\sqrt{2\pi}\sigma_{i,j}} \cdot e^{\frac{-1}{2\sigma_{i,j}^2}(\phi_{i,j} - \mu_{i,j})^2} \cdot e^{\frac{-1}{2\sigma_{3,j}^2 \cdot (c_{3,j}^2 + 1)}\left[ \phi_{3,j} - (c_{3,j}\bar{c}_{2,j}\phi_{1,j} + \bar{d}_{2,j}c_{3,j} + d_{3,j}) \right]^2} \cdot \\
& e^{\frac{d_{3,j}^2}{2\sigma_{3,j}^2}} \cdot \frac{1}{\sqrt{c_{3,j}^2 + 1}}.
\end{aligned}
$$

We denote $\bar{c}_{3,j} = c_{3,j}\bar{c}_{2,j}$ and $\bar{d}_{3,j} = \bar{d}_{2,j}c_{3,j} + d_{3,j}$, and denote $r_{2,j} = e^{\frac{d_{3,j}^2}{2\sigma_{3,j}^2}} \cdot \frac{1}{\sqrt{c_{3,j}^2 + 1}}$, so we have

$$
\begin{aligned}
& q(\phi_{m,j}, \phi_{m-1,j}, \ldots, \phi_{3,j}|\phi_{1,j}) \\
=& \frac{1}{\sqrt{2\pi}\sigma_{2,j}} \cdot \prod_{i=4}^{m} \frac{1}{\sqrt{2\pi}\sigma_{i,j}} \cdot e^{\frac{-1}{2\sigma_{i,j}^2}(\phi_{i,j} - \mu_{i,j})^2} \cdot e^{\frac{-1}{2\sigma_{3,j}^2 \cdot (c_{3,j}^2 + 1)}\left[ \phi_{3,j} - (\bar{c}_{3,j}\phi_{1,j} + \bar{d}_{3,j}) \right]^2} \cdot r_{2,j} \\
=& \frac{1}{\sqrt{2\pi}\sigma_{2,j}} \cdot \prod_{i=5}^{m} \frac{1}{\sqrt{2\pi}\sigma_{i,j}} \cdot e^{\frac{-1}{2\sigma_{i,j}^2}(\phi_{i,j} - \mu_{i,j})^2} \cdot \frac{1}{\sqrt{2\pi}\sigma_4} \cdot e^{\frac{-1}{2\sigma_4^2}[\phi_4 - (c_4\phi_{3,j} + d_4)]^2} \cdot \\
& e^{\frac{-1}{2\sigma_{3,j}^2 \cdot (c_{3,j}^2 + 1)}\left[ \phi_{3,j} - (\bar{c}_{3,j}\phi_{1,j} + \bar{d}_{3,j}) \right]^2} \cdot r_{2,j} \\
=& \frac{1}{\sqrt{2\pi}\sigma_{2,j}} \cdot \prod_{i=5}^{m} \frac{1}{\sqrt{2\pi}\sigma_{i,j}} \cdot e^{\frac{-1}{2\sigma_{i,j}^2}(\phi_{i,j} - \mu_{i,j})^2} \cdot \frac{1}{\sqrt{2\pi}\sigma_4} \cdot e^{\frac{-1}{2\sigma_4^2}[\phi_4 - (c_4\phi_{3,j} + d_4)]^2} \cdot \\
& e^{\frac{-1}{2\sigma_4^2}\left[ \phi_{3,j} - (\bar{c}_{3,j}\phi_{1,j} + \bar{d}_{3,j}) \right]^2} \cdot r_{2,j}
\end{aligned}
$$

Similarly, then we integrate $\phi_{3,j}$

$$q(\phi_{m,j}, \phi_{m-1,j}, \ldots, \phi_4 | \phi_{1,j})$$

$$= \int q(\phi_{m,j}, \phi_{m-1,j}, \ldots, \phi_{3,j} | \phi_{1,j}) \, d\phi_{3,j}$$

$$= \frac{r_{2,j}}{\sqrt{2\pi}\sigma_{2,j}} \cdot \prod_{i=5}^{m} \frac{1}{\sqrt{2\pi}\sigma_{i,j}} \cdot e^{\frac{-1}{2\sigma_{i,j}^2}(\phi_{i,j}-\mu_{i,j})^2} \cdot \int \frac{1}{\sqrt{2\pi}\sigma_4} \cdot e^{\frac{-1}{2\sigma_4^2}[\phi_4 - (c_4\phi_{3,j}+d_4)]^2} \cdot$$

$$e^{\frac{-1}{2\sigma_4^2}\left[\phi_{3,j}-(\bar{c}_{3,j}\phi_{1,j}+\bar{d}_{3,j})\right]^2} \, d\phi_{2,j}.$$

The formulation is similar to the previous one, so we can utilize the process above to integrate succesively, and we finally obtain

$$q(\phi_{m,j}|\phi_{1,j}) = \frac{\prod_{i=2}^{m-1} r_{i,j}}{\sqrt{2\pi}\sigma_{2,j}} \cdot e^{\frac{-1}{2\sigma_{m,j}^2 \cdot (c_{m,j}^2+1)}\left[\phi_{m,j}-(\bar{c}_{m,j}\phi_{1,j}+\bar{d}_{m,j})\right]^2},$$

which indicates

$$p(\phi_{m,j}|\phi_{1,j}, D) \approx \frac{\prod_{i=2}^{m-1} r_{i,j}}{\sqrt{2\pi}\sigma_{2,j}} \cdot e^{\frac{-1}{2\sigma_{m,j}^2 \cdot (c_{m,j}^2+1)}\left[\phi_{m,j}-(\bar{c}_m\phi_{1,j}+\bar{d}_{m,j})\right]^2},$$

where we have the iterative calculation by

$$r_{i,j} = e^{\frac{d_{i+1,j}^2}{2\sigma_{i+1,j}^2}} \cdot \frac{1}{\sqrt{c_{i+1,j}^2+1}},$$

$$\bar{c}_{i,j} = c_{i,j}\bar{c}_{i-1,j}, \ (i \geq 3),$$

$$\bar{d}_{i,j} = \bar{d}_{i-1,j}c_{i,j} + d_{i,j}, \ (i \geq 3).$$

## A.3 Sampling from $p(\phi_m | \phi_M, D)$

To sample from the distribution $p(\phi_m | \phi_M, D)$, we first obtain a sample $\phi_1$ from the prior distribution (*i.e.*, $\phi_1 \sim \mathcal{N}(\mathbf{0}, \mathbf{I})$), then get $\phi_m$ from a consecutive sampling process:

$$\phi_{M-1} \sim p(\phi_{M-1}|\phi_M, \phi_1, D),$$
$$\phi_{M-2} \sim p(\phi_{M-2}|\phi_{M-1}, \phi_1, D),$$
$$\vdots$$
$$\phi_m \sim p(\phi_m|\phi_{m+1}, \phi_1, D),$$

because of the Markov property in our cascaded model. So our target is obtaining the distributions $p(\phi_{i-1}|\phi_i, \phi_1, D)$. For a certain $p(\phi_{i-1}|\phi_i, \phi_1, D)$, according to the Bayes rule, we have

$$p(\phi_{i-1}|\phi_i, \phi_1, D) = \frac{p(\phi_i|\phi_{i-1}, \phi_1, D) \cdot p(\phi_{i-1}|\phi_1, D)}{p(\phi_i|\phi_1, D)}.$$

Similarly with the last section, we use non-bold symbols to represent one dimension of the multi-dimension parameters, where they are able to transfer independently, and finally construct the eventual parameters by concatenating, that is,

$$p(\phi_{i-1}|\phi_i, \phi_1, D) = \prod_{j=1}^{d} p(\phi_{i,j}|\phi_{i-1,j}, \phi_{1,j}, D).$$

So our target can be converted to calculate the probability $p(\phi_{i,j}|\phi_{i-1,j}, \phi_{1,j}, D)$ for all dimensions $\forall 1 \leq j \leq d$. According to the Markov property and the transportation probability, we have

$$p(\phi_{i,j}|\phi_{i-1,j}, \phi_{1,j}, D) \approx q(\phi_{i,j}|\phi_{i-1,j}, \phi_{1,j})$$

$$= \frac{1}{\sqrt{2\pi}\sigma_{i,j}} \cdot e^{\frac{-1}{2\sigma_{i,j}^2} \cdot [\phi_{i,j}-(c_i\phi_{i-1,j}+d_{i,j})]^2}.$$

According to the previous section, we have

$$p(\phi_{i,j}|\phi_{1,j}, D) \approx q(\phi_{i,j}|\phi_{1,j}) = \frac{\prod_{i=2}^{i-1} r_{i,j}}{\sqrt{2\pi}\sigma_{2,j}} \cdot e^{\frac{-1}{2\sigma_{i,j}^2 \cdot (c_{i,j}^2+1)}\left[\phi_{i,j}-(\bar{c}_i\phi_{1,j}+\bar{d}_i)\right]^2},$$

$$p(\phi_{i-1,j}|\phi_{1,j}, D) \approx q(\phi_{i-1,j}|\phi_{1,j}) = \frac{\prod_{i=2}^{i-2} r_{i,j}}{\sqrt{2\pi}\sigma_{2,j}} \cdot e^{\frac{-1}{2\sigma_{i-1,j}^2 \cdot (c_{i-1,j}^2+1)}\left[\phi_{i-1,j}-(\bar{c}_{i-1,j}\phi_{1,j}+\bar{d}_{i-1,j})\right]^2}.$$

Then we have

$$p(\phi_{i-1,j}|\phi_{i,j}, \phi_{1,j}, D) \approx \frac{q(\phi_{i,j}|\phi_{i-1,j}, \phi_{1,j}) \cdot q(\phi_{i-1,j}|\phi_{1,j})}{q(\phi_{i,j}|\phi_{1,j})}$$

$$= \frac{1}{\sqrt{2\pi}\sigma_{i,j} \cdot r_{i-1}} \cdot e^{\frac{-1}{2\sigma_{i,j}^2}\cdot[\phi_{i,j}-(c_{i,j}\phi_{i-1,j}+d_{i,j})]^2} \cdot \frac{e^{\frac{[\phi_{i-1,j}-(\bar{c}_{i-1,j}\phi_{1,j}+\bar{d}_{i-1,j})]^2}{-2\sigma_{i-1,j}^2\cdot(c_{i-1,j}^2+1)}}}{e^{\frac{[\phi_{i,j}-(c_{i,j}\phi_{i-1,j}+d_{i,j})]^2}{-2\sigma_{i,j}^2\cdot(c_{i,j}^2+1)}}}$$

$$= \sqrt{\frac{c_{i,j}+1}{2\pi\sigma_{i,j}^2}} \cdot e^{\frac{2\sigma_{i,j}^2}{d_{i,j}^2}} \cdot e^{\frac{[\phi_{i,j}-(c_{i,j}\phi_{i-1,j}+d_{i,j})]^2}{2\sigma_{i+1,j}^2}} \cdot e^{\frac{[\phi_{i,j}-(c_{i,j}\phi_{i-1,j}+d_{i,j})]^2}{-2\sigma_{i,j}^2}} \cdot e^{\frac{[\phi_{i-1,j}-(\bar{c}_{i-1,j}\phi_{1,j}+\bar{d}_{i-1,j})]^2}{-2\sigma_{i,j}^2}}$$

$$= \sqrt{\frac{c_{i,j}+1}{2\pi\sigma_{i,j}^2}} \cdot e^{\frac{2\sigma_{i,j}^2}{d_{i,j}^2}+\frac{[\phi_{i,j}-(c_{i,j}\phi_{i-1,j}+d_{i,j})]^2}{2\sigma_{i+1,j}^2}} \cdot$$

$$e^{\frac{1}{-2\sigma_{i,j}^2}\cdot\underbrace{\left\{\left[\phi_{i,j}-(c_{i,j}\phi_{i-1,j}+d_{i,j})\right]^2+\left[\phi_{i-1,j}-(\bar{c}_{i-1,j}\phi_{1,j}+\bar{d}_{i-1,j})\right]^2\right\}}_{C}}.$$

Then we calculate the part $C$ as

$$\begin{aligned}
C &= [\phi_{i,j}-(c_{i,j}\phi_{i-1,j}+d_{i,j})]^2 + \left[\phi_{i-1,j}-(\bar{c}_{i-1,j}\phi_{1,j}+\bar{d}_{i-1,j})\right]^2 \\
&= \phi_{i,j}^2 + (c_{i,j}\phi_{i-1,j}+d_{i,j})^2 - 2(c_{i,j}\phi_{i-1,j}+d_{i,j})\phi_{i,j} \\
&\quad + \phi_{i-1,j}^2 + (\bar{w}_{i-1}\phi_{1,j}+\bar{d}_{i-1,j})^2 - 2\phi_{i-1,j}(\bar{c}_{i-1,j}\phi+\bar{d}_{i-1,j}) \\
&= \phi_{i,j}^2 + c_{i,j}^2\phi_{i-1,j}^2 + d_{i,j}^2 + 2d_{i,j}c_{i,j}\phi_{i-1,j} - 2c_{i,j}\phi_{i-1,j}\phi_{i,j} - 2d_{i,j}\phi_{i,j} + \phi_{i-1,j}^2 \\
&\quad + \bar{c}_{i-1,j}^2\phi_{1,j}^2 + \bar{d}_{i-1,j}^2 + 2\bar{c}_{i-1,j}\bar{d}_{i-1,j}\phi_{1,j} - 2\bar{c}_{i-1,j}\phi_{1,j}\phi_{i-1,j} - 2\bar{d}_{i-1,j}\phi_{i-1,j} \\
&= (\bar{w}_{i-1}^2+1)\cdot\left[\phi_{i-1,j}-\frac{c_{i,j}\phi_{i,j}+\bar{c}_{i-1,j}\phi_{1,j}+\bar{d}_{i-1,j}-d_{i,j}c_{i,j}}{c_{i,j}^2+1}\right]^2 + B,
\end{aligned}$$

where $B$ does not include $\phi_i$, which indicates

$$p(\phi_{i-1,j}|\phi_{i,j}, \phi_{1,j}, D) \sim \mathcal{N}(\frac{c_{i,j}\phi_{i,j}+\bar{c}_{i-1,j}\phi_{1,j}+\bar{d}_{i-1,j}-d_{i,j}w_{i-2}}{c_{i,j}^2+1}, \frac{\sigma_{i,j}^2}{c_{i,j}^2+1}).$$

### A.4   Calculation of $p(\boldsymbol{x}|\boldsymbol{e},\boldsymbol{\phi}_m)$

In this section, we will show how to calculate the graph probability $p(\boldsymbol{x}|\boldsymbol{e}, \boldsymbol{\phi}_m)$. Remember the graph parameters $\boldsymbol{\phi}_m = [\boldsymbol{\theta}_m; \boldsymbol{S}_m; \boldsymbol{T}_m]$, so we have

$$\begin{aligned}
p(\boldsymbol{x}|\boldsymbol{e}, \boldsymbol{\phi}_m) &= \int_{\mathbf{E}} p(\boldsymbol{x}|\boldsymbol{e}, \boldsymbol{\theta}_m, \mathbf{E}) \cdot p(\mathbf{E}|\boldsymbol{e}, \boldsymbol{S}_m, \boldsymbol{T}_m) \, d\mathbf{E} \\
&= \mathbb{E}_{\mathbf{E}\sim p(\mathbf{E}|\boldsymbol{S}_m, \boldsymbol{T}_m)}\left[p(\boldsymbol{x}|\boldsymbol{e}, \boldsymbol{\theta}_m, \mathbf{E})\right].
\end{aligned}$$

According to Monte Carlo sampling, we have

$$p(\boldsymbol{x}|\boldsymbol{e}, \boldsymbol{\phi}_m) = \frac{1}{K} \cdot \sum_{l=1}^{K} p(\boldsymbol{x}|\boldsymbol{e}, \boldsymbol{\theta}_m, \mathbf{E}_l),$$

where $\mathbf{E}_l[i,j] \sim \text{Bernoulli}(\sigma(\mathbf{S}_m^T[i] \cdot \mathbf{T}_m[j]))$. In order to conduct intervention process, we change the $j$th column of $\mathbf{E}_l$ to zeros, and represent it with $\tilde{\mathbf{E}}_l$. Moreover, we replace the $j$th element of $\boldsymbol{x}$

with $v$, and get the result $\tilde{x}$. We change the $j$th element of $\epsilon_m$ with zero, and get the result $\tilde{\epsilon}_m$. Then according the definition of causal graphs, we have

$$p(\boldsymbol{x}|e, \boldsymbol{\phi}_m) = \frac{1}{K} \sum_{l=1}^{K} \mathcal{N}(\boldsymbol{x}; \boldsymbol{f}(\tilde{\boldsymbol{x}}; \tilde{\mathbf{E}}_l, \boldsymbol{\gamma}_m), \tilde{\boldsymbol{\epsilon}}_m),$$

where $\boldsymbol{f}$ is the causal function that depends on the parameter $\boldsymbol{\gamma}_m$.

## B   Bayesian Optimization for Determining $(j^*, v^*, m^*)$

We intend to find the best tuple for acquisition, that is,

$$(j^*, v^*, m^*) = \underset{(j,v,m)}{\arg\max} f(j, v, m).$$

We define the best interventional value $v$ under interventional node $j$ and fidelity $m$ as

$$v^*(j, m) = \underset{v}{\arg\max} f(j, v, m)$$

$$= \underset{v}{\arg\max} f_{j,m}(v).$$

where $f_{j,m}(v)$ is rewritten from $f(j, v, m)$ under given $j, m$. Therefore, our task is calculating $v^*(j, m)$ for $\forall j \in [d], m \in [M]$ with Bayesian optimization [47]. We utilize a Gaussian Process (GP) [48] to model surrogate function distributions for each $v^*(j, m)$. We denote $f \sim \mathcal{GP}(\mathbf{0}, \mathcal{K}(v_i, v_j))$, and $\mathcal{K}(v_i, v_j)$ is the kernel of GP. We sequentially find $v_t$ and calculate $f_{j,m}(v_t)$ to direct the process. According to GP, the previous $t$ functions and the $t+1$ function are multivariate Gaussian distribution,

$$\begin{bmatrix} \mathbf{F}_{1:t} \\ f_{t+1} \end{bmatrix} \sim \mathcal{N}\left(\mathbf{0}, \begin{bmatrix} \mathbf{K}_t & \boldsymbol{k}_{t+1} \\ \boldsymbol{k}_{t+1}^T & \mathcal{K}(v_{t+1}, v_{t+1}) \end{bmatrix}\right),$$

where we define

$$\mathbf{F}_{1:t} = [f_1, f_2, \ldots, f_t],$$

$$\boldsymbol{k}_{t+1} = [\mathcal{K}(v_{t+1}, v_1), \mathcal{K}(v_{t+1}, v_2), \ldots, \mathcal{K}(v_{t+1}, v_{t+1})]^T,$$

$$\mathbf{K}_t = \begin{bmatrix} \mathcal{K}(v_1, v_1) & \cdots & \mathcal{K}(v_t, v_1) \\ \vdots & \ddots & \vdots \\ \mathcal{K}(v_t, v_1) & \cdots & \mathcal{K}(v_t, v_t) \end{bmatrix}. \tag{11}$$

Given previous $t$ steps, we have the posterior probability is

$$p(f_{t+1}|\{(v_i, f_{j,m}(v_i))\}_{i=1}^t, v_{t+1}) = \mathcal{N}(\mu_t(v_{t+1}), \sigma_t^2(v_{t+1})),$$

with the non-parametric means and variances

$$\mu_t(v_{t+1}) = \boldsymbol{k}_{t+1}^T(\mathbf{K} + \mathbf{I})^{-1}\mathbf{F}_{1:t}, \tag{12}$$

$$\sigma_t^2(v_{t+1}) = \mathcal{K}(v_{v+1}, v_{t+1}) - \boldsymbol{k}_{t+1}^T(\mathbf{K} + \mathbf{I})^{-1}\boldsymbol{k}_{t+1}. \tag{13}$$

We acquire the next $v_{t+1}$ with GP-UCB [49] function

$$a_{t+1}(v) = \mu_t(v) + \beta_{ac} \cdot \sqrt{\sigma_t^2(v)},$$

$$v_{t+1} = \underset{v}{\arg\max} a_{t+1}(v).$$

where $\beta_{ac}$ is a hyper-parameter. Suppose the maximum of steps is $T$, the final output of function $v^*(j, m)$ is

$$v^*(j, m) = \underset{v}{\arg\max} \mu_T(v).$$

Then we choose the best interventional node $j$ and fidelity $m$ by their best values under $\mathcal{O}(d \cdot M)$

$$j^*, m^* = \underset{j,m}{\arg\max} v^*(j, m),$$

$$v^* = v^*(j^*, m^*).$$

# C  Detailed Training Process of ELBO

## C.1  Derivation Process of ELBO

Because we use the distribution $q(\phi_m)$ to approximate the distribution $p(\phi_m)$, then we intend to minimize the distance between these two distributions optimize the parameters of $q(\phi_m)$, where we utilize KL divergence to measure the distance, that is,

$$\Psi^* = \arg\min_{\Psi} \text{KL}[q(\boldsymbol{\Phi})||p(\boldsymbol{\Phi}|D)].$$

According to the variational inference, we have

$$
\begin{aligned}
&\text{KL}\left[q(\boldsymbol{\Phi})||p(\boldsymbol{\Phi}|D)\right] \\
&= \int q(\boldsymbol{\Phi}) \log \frac{q(\boldsymbol{\Phi})}{p(\boldsymbol{\Phi}|D)} \, d\boldsymbol{\Phi} \\
&= \int q(\boldsymbol{\Phi}) \log q(\boldsymbol{\Phi}) \, d\boldsymbol{\Phi} - \int q(\boldsymbol{\Phi}) \log p(\boldsymbol{\Phi}|D) \, d\boldsymbol{\Phi} \\
&= \mathbb{E}_{\boldsymbol{\Phi}\sim q(\boldsymbol{\Phi})}\left[\log q(\boldsymbol{\Phi})\right] - \int q(\boldsymbol{\Phi}) \log \frac{p(\boldsymbol{\Phi}, D)}{p(D)} \, d\boldsymbol{\Phi} \\
&= \mathbb{E}_{\boldsymbol{\Phi}\sim q(\boldsymbol{\Phi})}\left[\log q(\boldsymbol{\Phi})\right] - \int q(\boldsymbol{\Phi}) \log p(\boldsymbol{\Phi}, D) \, d\boldsymbol{\Phi} + \int q(\boldsymbol{\Phi}) \log p(D) \, d\boldsymbol{\Phi} \\
&= \mathbb{E}_{\boldsymbol{\Phi}\sim q(\boldsymbol{\Phi})}\left[\log q(\boldsymbol{\Phi})\right] - \mathbb{E}_{\boldsymbol{\Phi}\sim q(\boldsymbol{\Phi})}\left[\log p(\boldsymbol{\Phi}, D)\right] + \int q(\boldsymbol{\Phi}) \log p(D) \, d\boldsymbol{\Phi} \\
&= \underbrace{\mathbb{E}_{\boldsymbol{\Phi}\sim q(\boldsymbol{\Phi})}\left[\log q(\boldsymbol{\Phi})\right] - \mathbb{E}_{\boldsymbol{\Phi}\sim q(\boldsymbol{\Phi})}\left[\log p(\boldsymbol{\Phi}, D)\right]}_{-\text{ELBO}} + \log p(D).
\end{aligned}
$$

Because $\log p(D)$ is not related to $\Psi$, minimizing $\text{KL}\left[q(\boldsymbol{\Phi})||p(\boldsymbol{\Phi}|D)\right]$ is equivalent to maximizing the ELBO part, and we have

$$
\begin{aligned}
\text{ELBO} &= \mathbb{E}_{\boldsymbol{\Phi}\sim q(\boldsymbol{\Phi})}\left[\log p(\boldsymbol{\Phi}, D)\right] - \mathbb{E}_{\boldsymbol{\Phi}\sim q(\boldsymbol{\Phi})}\left[\log q(\boldsymbol{\Phi})\right] \\
&= \mathbb{E}_{\boldsymbol{\Phi}\sim q(\boldsymbol{\Phi})}\left[\log p(D|\boldsymbol{\Phi})\right] + \mathbb{E}_{\boldsymbol{\Phi}\sim q(\boldsymbol{\Phi})}\left[\log p(\boldsymbol{\Phi})\right] - \mathbb{E}_{\boldsymbol{\Phi}\sim q(\boldsymbol{\Phi})}\left[\log q(\boldsymbol{\Phi})\right] \\
&= \mathbb{E}_{\boldsymbol{\Phi}\sim q(\boldsymbol{\Phi})}\left[\log p(D|\boldsymbol{\Phi}) - \log q(\boldsymbol{\Phi}) + \log p(\boldsymbol{\Phi})\right]
\end{aligned}
$$

Above all, we can conclude that

$$\Psi^* = \arg\min_{\Psi} \text{KL}[q(\boldsymbol{\Phi}||p(\boldsymbol{\Phi}|D)]$$

is equivalent to maximize evidence lower bound

$$
\begin{aligned}
\Psi^* &= \arg\max_{\Psi} \text{ELBO} \\
&= \arg\max_{\Psi} \mathbb{E}_{\boldsymbol{\Phi}\sim q(\boldsymbol{\Phi})}\left[\log p(D|\boldsymbol{\Phi}) - \log q(\boldsymbol{\Phi}) + \log p(\boldsymbol{\Phi})\right].
\end{aligned}
$$

## C.2  Estimation of ELBO

We represent the equation of ELBO as

$$
\begin{aligned}
\text{ELBO} &= \mathbb{E}_{\boldsymbol{\Phi}\sim q(\boldsymbol{\Phi})}\left[\log p(D|\boldsymbol{\Phi}) - \log q(\boldsymbol{\Phi}) + \log p(\boldsymbol{\Phi})\right] \\
&= \underbrace{\mathbb{E}_{\boldsymbol{\Phi}\sim q(\boldsymbol{\Phi})}\left[\log p(D|\boldsymbol{\Phi})\right]}_{A} - \underbrace{\mathbb{E}_{\boldsymbol{\Phi}\sim q(\boldsymbol{\Phi})}\left[\log q(\boldsymbol{\Phi}) - \log p(\boldsymbol{\Phi})\right]}_{B}.
\end{aligned}
$$

For the part $A$, we have

$$A = \mathbb{E}_{\boldsymbol{\Phi}\sim q(\boldsymbol{\Phi})}\left[\log \prod_{i=1}^{N} p(\boldsymbol{x}^{(i)}|j^{(i)}, v^{(i)}, m^{(i)}, \boldsymbol{\Phi})\right],$$

where $N$ is the current number of samples in buffer. Then we have

$$A = \mathbb{E}_{\boldsymbol{\Phi} \sim q(\boldsymbol{\Phi})} \left[ \log \prod_{i=1}^{N} p(\boldsymbol{x}^{(i)} | j^{(i)}, v^{(i)}, m^{(i)}, \boldsymbol{\Phi}) \right]$$

$$= \mathbb{E}_{\boldsymbol{\Phi} \sim q(\boldsymbol{\Phi})} \left[ \sum_{i=1}^{N} \log p(\boldsymbol{x}^{(i)} | j^{(i)}, v^{(i)}, m^{(i)}, \boldsymbol{\Phi}) \right]$$

$$= \sum_{i=1}^{N} \mathbb{E}_{\boldsymbol{\Phi} \sim q(\boldsymbol{\Phi})} \left[ \log p(\boldsymbol{x}^{(i)} | j^{(i)}, v^{(i)}, m^{(i)}, \boldsymbol{\Phi}) \right]$$

$$= \sum_{i=1}^{N} \mathbb{E}_{\boldsymbol{\phi}_{m^{(i)}} \sim q(\boldsymbol{\phi}_{m^{(i)}})} \left[ \log p(\boldsymbol{x}^{(i)} | j^{(i)}, v^{(i)}, m^{(i)}, \boldsymbol{\phi}_{m^{(i)}}) \right].$$

Using Monte Carlo sampling [50], we can calculate the expectation by $N_S$ samples for each point.

$$A = \sum_{i=1}^{N} \sum_{j=1}^{N_S} \log p(\boldsymbol{x}^{(i)} | j^{(i)}, v^{(i)}, m^{(i)}, \boldsymbol{\phi}_{m^{(i)}}^{(j)}),$$

where we sample $\boldsymbol{\phi}_{m^{(i)}}^{(j)} \sim q(\boldsymbol{\phi}_{m^{(i)}})$ with size $N_S$.

Then we denote the distribution $q(\boldsymbol{\Phi}) = \mathcal{N}(\tilde{\boldsymbol{\mu}}_{all}, \tilde{\boldsymbol{\Sigma}}_{all})$, and similarly, we have $p(\boldsymbol{\Phi}) = \prod_{m=1}^{M} e^{-\beta \cdot f(\mathbf{S}_m, \mathbf{T}_m)} \cdot \mathcal{N}(\boldsymbol{\mu}_{all}, \boldsymbol{\Sigma}_{all})$. Both the parameter $\tilde{\boldsymbol{\mu}}_{all}, \tilde{\boldsymbol{\Sigma}}_{all}$ can be represented by the parameters in $\Psi$, while $\boldsymbol{\mu}_{all}$ and $\boldsymbol{\Sigma}_{all}$ are constant. Then we calculate part $B$

$$B = \mathbb{E}_{\boldsymbol{\Phi} \sim q(\boldsymbol{\Phi})} \left[ \log q(\boldsymbol{\Phi}) - \log p(\boldsymbol{\Phi}) \right]$$

$$= \int_{\boldsymbol{\Phi}} \mathcal{N}(\tilde{\boldsymbol{\mu}}_{all}, \tilde{\boldsymbol{\Sigma}}_{all}) \log \frac{\mathcal{N}(\tilde{\boldsymbol{\mu}}_{all}, \tilde{\boldsymbol{\Sigma}}_{all})}{\prod_{m=1}^{M} e^{-\beta \cdot f(\mathbf{S}_m, \mathbf{T}_m)} \cdot \mathcal{N}(\boldsymbol{\mu}_{all}, \boldsymbol{\Sigma}_{all})} \, d\boldsymbol{\Phi}$$

$$= \int_{\boldsymbol{\Phi}} \mathcal{N}(\tilde{\boldsymbol{\mu}}_{all}, \tilde{\boldsymbol{\Sigma}}_{all}) \log \frac{\mathcal{N}(\tilde{\boldsymbol{\mu}}_{all}, \tilde{\boldsymbol{\Sigma}}_{all})}{\mathcal{N}(\boldsymbol{\mu}_{all}, \boldsymbol{\Sigma}_{all})} \, d\boldsymbol{\Phi} + \int_{\boldsymbol{\Phi}} \mathcal{N}(\tilde{\boldsymbol{\mu}}_{all}, \tilde{\boldsymbol{\Sigma}}_{all}) \log \frac{1}{\prod_{m=1}^{M} e^{-\beta \cdot f(\mathbf{S}_m, \mathbf{T}_m)}} \, d\boldsymbol{\Phi}$$

$$= \underbrace{\mathrm{KL}[\mathcal{N}(\tilde{\boldsymbol{\mu}}_{all}, \tilde{\boldsymbol{\Sigma}}_{all}) || \mathcal{N}(\boldsymbol{\mu}_{all}, \boldsymbol{\Sigma}_{all})]}_{C} + \underbrace{\int_{\boldsymbol{\Phi}} \mathcal{N}(\tilde{\boldsymbol{\mu}}_{all}, \tilde{\boldsymbol{\Sigma}}_{all}) \log \prod_{m=1}^{M} e^{\beta \cdot f(\mathbf{S}_m, \mathbf{T}_m)} \, d\boldsymbol{\Phi}}_{D}.$$

According to KL divergence of Gaussian distribution, we can calculate $C$ in a close-form.

$$C = \mathrm{KL}[\mathcal{N}(\tilde{\boldsymbol{\mu}}_{all}, \tilde{\boldsymbol{\Sigma}}_{all}) || \mathcal{N}(\boldsymbol{\mu}_{all}, \boldsymbol{\Sigma}_{all})]$$

$$= \frac{1}{2} \left[ \log \frac{||\boldsymbol{\Sigma}_{all}||}{||\tilde{\boldsymbol{\Sigma}}_{all}||} - d + \mathrm{tr}(\boldsymbol{\Sigma}_{all}^{-1} \tilde{\boldsymbol{\Sigma}}_{all}) + (\tilde{\boldsymbol{\mu}}_{all} - \boldsymbol{\mu}_{all})^T \boldsymbol{\Sigma}_{all}^{-1} (\tilde{\boldsymbol{\mu}}_{all} - \boldsymbol{\mu}_{all}) \right].$$

Then we calculate $D$ by the following steps:

$$D = \int_{\boldsymbol{\Phi}} \mathcal{N}(\boldsymbol{\mu}_{all}, \boldsymbol{\Sigma}_{all}) \log \prod_{m=1}^{M} e^{\beta \cdot f(\mathbf{S}_m, \mathbf{T}_m)} \, d\boldsymbol{\Phi}$$

$$= \int_{\boldsymbol{\Phi}} \mathcal{N}(\boldsymbol{\mu}_{all}, \boldsymbol{\Sigma}_{all}) \sum_{m=1}^{M} \log e^{\beta \cdot f(\mathbf{S}_m, \mathbf{T}_m)} \, d\boldsymbol{\Phi}$$

$$= \cdot \int_{\boldsymbol{\Phi}} \mathcal{N}(\boldsymbol{\mu}_{all}, \boldsymbol{\Sigma}_{all}) \sum_{m=1}^{M} \beta \cdot f(\mathbf{S}_m, \mathbf{T}_m) \, d\boldsymbol{\Phi}$$

$$= \beta \cdot \mathbb{E}_{\boldsymbol{\Phi} \sim \mathcal{N}(\boldsymbol{\mu}_{all}, \boldsymbol{\Sigma}_{all})} \left[ \sum_{m=1}^{M} f(\mathbf{S}_m, \mathbf{T}_m) \right].$$

Using Monte Carlo sampling, we can calculate the expectation by $N_D$ samples for each point.

$$D = \beta \cdot \sum_{i=1}^{N_D} \sum_{m=1}^{M} f(\mathbf{S}_m^{(i)}, \mathbf{T}_m^{(i)}).$$

$$= \beta \cdot \sum_{i=1}^{N_D} \sum_{m=1}^{M} \mathbb{E}_{p(\mathbf{E}|\mathbf{S}_m^{(i)}, \mathbf{T}_m^{(i)})} \left[ \lambda_1 \cdot \left[ \mathrm{tr}\left(e^{\mathbf{E}}\right) - d \right] + \lambda_2 \cdot ||\mathbf{E}|| \right],$$

where we samples $\boldsymbol{\Phi}^{(i)} \sim \mathcal{N}(\boldsymbol{\mu}_{all}, \boldsymbol{\Sigma}_{all})$ with size $N_D$. Using Monte Carlo sampling again, we can calculate the expectation by $N_E$ samples.

$$D = \beta \cdot \sum_{i=1}^{N_D} \sum_{m=1}^{M} \sum_{j=1}^{N_E} \left[ \lambda_1 \cdot \left[ \mathrm{tr}\left(e^{\mathbf{E}}\right) - d \right] + \lambda_2 \cdot ||\mathbf{E}|| \right],$$

where we samples $\mathbf{E}^{(j)} \sim p(\mathbf{E}|\mathbf{S}_m^{(i)}, \mathbf{T}_m^{(i)})$ with size $N_E$.

Finally, we obtain the estimation

$$\begin{aligned}
\mathrm{ELBO} = &\sum_{i=1}^{N} \sum_{j=1}^{N_S} \log p(x^{(i)}|j^{(i)}, v^{(i)}, m^{(i)}, \boldsymbol{\phi}_{m^{(i)}}^{(j)}) \\
&- \frac{1}{2} \left[ \log \frac{||\boldsymbol{\Sigma}_{all}||}{||\tilde{\boldsymbol{\Sigma}}_{all}||} - d + \mathrm{tr}(\boldsymbol{\Sigma}_{all}^{-1} \tilde{\boldsymbol{\Sigma}}_{all}) + (\tilde{\boldsymbol{\mu}}_{all} - \boldsymbol{\mu}_{all})^T \boldsymbol{\Sigma}_{all}^{-1} (\tilde{\boldsymbol{\mu}}_{all} - \boldsymbol{\mu}_{all}) \right] \\
&- \beta \cdot \sum_{i=1}^{N_D} \sum_{m=1}^{M} \sum_{j=1}^{N_E} \left[ \lambda_1 \cdot \left[ \mathrm{tr}\left(e^{\mathbf{E}}\right) - d \right] + \lambda_2 \cdot ||\mathbf{E}|| \right].
\end{aligned}$$

### C.3 Gaussian Reparameterization Trick

In the last section, we derive the objection function for optimizing the model parameters, where we can use methods of the gradient decent to solve it. However, a significant problem rises due to the sampling process, because the gradient of model parameters can not pass backward from the naive sampling process(*i.e.,* untraceable). Therefore, we use Gaussian reparameterization trick to make the Gaussian sampling process traceable.

In specific, we will demonstrate the traceable calculation of $\phi$ by Gaussian reparameterization trick. In order to sample $\phi \sim \mathcal{N}(\boldsymbol{\mu}, \boldsymbol{\Sigma})$, we first sample $\boldsymbol{\delta} \sim \mathcal{N}(\mathbf{0}, \mathbf{I})$ instead, and then obtain $\phi = \boldsymbol{\mu} + \boldsymbol{\delta} \odot \boldsymbol{\Sigma}$. Therefore, the gradient can be traced from $\phi$ to $\boldsymbol{\mu}$ and $\boldsymbol{\Sigma}$. In specific, both $\boldsymbol{\mu}$ and $\boldsymbol{\Sigma}$ can be represented with the function of learnable parameter $\Psi$.

### C.4 Gumbel-softmax Reparameterization Trick

Besides of the Gaussian sampling process, the Bernoulli sampling in our equation is not traceable either, so we utilize Gumbel-softmax reparameterization trick to make it traceable.

We demonstrate the traceable calculation of $\mathbf{E} \sim p(\mathbf{E}|\mathbf{S}, \mathbf{T})$ by Gumbel-max reparameterization trick. According to Gumbel-max [51], we have

$$\mathrm{Bernoulli}(p) \iff \mathbf{1}\left[ G_1 + \log p > G_0 + \log(1-p) \right], \quad G_0, G_1 \sim \mathrm{Gumbel}(0, 1).$$

Instead of using unit step function, we utilize sigmoid function

$$\sigma(G_1 + \log p > G_0 + \log(1-p)).$$

Therefore, we have

$$\mathbf{E}_{i,j} = \sigma(\mathbf{L}_{i,j} + \mathbf{S}_i^T \cdot \mathbf{T}_j),$$

where $\mathbf{L}_{i,j} \sim L(0, 1)$. Therefore, we sample $\mathbf{L}_{i,j} \sim L(0, 1)$ instead, where $L(0, 1)$ is logistic distribution, and calculate $\mathbf{E}_{i,j} = \sigma(\mathbf{L}_{i,j} + \mathbf{S}_i^T \cdot \mathbf{T}_j)$ to trace gradients. Specifically, both $\boldsymbol{S}_i$ and $\boldsymbol{T}_i$ can be represented with the function of learnable parameter $\Psi$.

## C.5 Optimization of ELBO

With the estimation and reparameterization trick, we are able to conduct gradient descent methods to optimize our parameters with the objection function

$$\Psi^* = \arg \max_{\Psi} \text{ELBO}.$$

The format of stochastic gradient descent (SGD) is

$$\Psi \leftarrow \Psi + \gamma \cdot \frac{\partial \text{ELBO}}{\partial \Psi},$$

where $\gamma$ is the learning rate.

## D    Training Process of Constraint based ELBO

We intend to optimize our parameter with

$$\Psi^* = \arg \max_{\Psi} \mathbb{E}_{\boldsymbol{\Phi} \sim q(\boldsymbol{\Phi})} \left[ \log p(D|\boldsymbol{\Phi}) - \log q(\boldsymbol{\Phi}) + \log p(\boldsymbol{\Phi}) \right],$$

$$\text{s.t.} \sum_{\{\boldsymbol{e}_s, \boldsymbol{e}_t\}} I(\boldsymbol{x}_s; \boldsymbol{x}_t | \boldsymbol{\phi}_M, \{\boldsymbol{e}_s, \boldsymbol{e}_t\}, D) \leq \epsilon.$$

However, the objection has a constraint, which is hard to optimize with gradient descent methods. So we utilize Lagrange multiplier [52] to convert it to a constraint-free method:

$$\Psi^* = \arg \max_{\Psi} \mathbb{E}_{\boldsymbol{\Phi} \sim q(\boldsymbol{\Phi})} \left[ \log p(D|\boldsymbol{\Phi}) - \log q(\boldsymbol{\Phi}) + \log p(\boldsymbol{\Phi}) \right] + \lambda \cdot \sum_{\{\boldsymbol{e}_s, \boldsymbol{e}_t\}} I(\boldsymbol{x}_s; \boldsymbol{x}_t | \boldsymbol{\phi}_M, \{\boldsymbol{e}_s, \boldsymbol{e}_t\}, D),$$

where $\lambda$ is the Lagrange multiplier. Then, we intend to calculate the constraint part.

First of all, we have

$$\begin{aligned}
& I(\boldsymbol{x}_s; \boldsymbol{x}_t | \boldsymbol{\phi}_M, \{\boldsymbol{e}_s, \boldsymbol{e}_t\}, D) \\
=& H(\boldsymbol{x}_s | \boldsymbol{\phi}_M, \{\boldsymbol{e}_s, \boldsymbol{e}_t\}, D) + H(\boldsymbol{x}_t | \boldsymbol{\phi}_M, \{\boldsymbol{e}_s, \boldsymbol{e}_t\}, D) - H(\boldsymbol{x}_s, \boldsymbol{x}_t | \boldsymbol{\phi}_M, \{\boldsymbol{e}_s, \boldsymbol{e}_t\}, D) \\
=& H(\boldsymbol{x}_s | \boldsymbol{\phi}_M, \boldsymbol{e}_s, D) + H(\boldsymbol{x}_t | \boldsymbol{\phi}_M, \boldsymbol{e}_t, D) - H(\boldsymbol{x}_s, \boldsymbol{x}_t | \boldsymbol{\phi}_M, \{\boldsymbol{e}_s, \boldsymbol{e}_t\}, D),
\end{aligned}$$

For the term $H(\boldsymbol{x}_s | \boldsymbol{\phi}_M, \boldsymbol{e}_s, D)$, we have

$$\begin{aligned}
H(\boldsymbol{x}_s | \boldsymbol{\phi}_M, \boldsymbol{e}_s, D) &= - \int p(\boldsymbol{x}_s | \boldsymbol{\phi}_M, \boldsymbol{e}_s, D) \log p(\boldsymbol{x}_s | \boldsymbol{\phi}_M, \boldsymbol{e}_s, D) \, d\boldsymbol{x}_s \\
&= - \mathbb{E}_{p(\boldsymbol{x}_s | \boldsymbol{\phi}_M, \boldsymbol{e}_s, D)} \left[ \log p(\boldsymbol{x}_s | \boldsymbol{\phi}_M, \boldsymbol{e}_s, D) \right].
\end{aligned}$$

We use Monte Carlo sampling to estimate $H(\boldsymbol{x}_s | \boldsymbol{\phi}_M, \boldsymbol{e}_s, D)$, and we have

$$H(\boldsymbol{x}_s | \boldsymbol{\phi}_M, \boldsymbol{e}_s, D) \approx \frac{1}{K_1 \cdot K_2} \sum_{k_1=1}^{K_1} \sum_{k_2=1}^{K_2} \log p(\boldsymbol{x}^{(k_1, k_2)} | \boldsymbol{e}^s, \boldsymbol{\phi}_M),$$

where we sample graphs $\boldsymbol{\phi}_m^{k_1} \sim q(\boldsymbol{\phi}_m | \boldsymbol{e}^s, \boldsymbol{\phi}_M)$, and obtain samples $\boldsymbol{x}^{(k_1, k_2)} \sim p(\boldsymbol{x} | \boldsymbol{e}^s, \boldsymbol{\phi}_m^{k_1})$. Similarly, we can calculate

$$H(\boldsymbol{x}_t | \boldsymbol{\phi}_M, \boldsymbol{e}_t, D) \approx \frac{1}{K_1 \cdot K_2} \sum_{k_1=1}^{K_1} \sum_{k_2=1}^{K_2} \log p(\boldsymbol{x}^{(k_1, k_2)} | \boldsymbol{e}^t, \boldsymbol{\phi}_M),$$

where we sample graphs $\boldsymbol{\phi}_m^{k_1} \sim q(\boldsymbol{\phi}_m | \boldsymbol{e}^t, \boldsymbol{\phi}_M)$, and obtain samples $\boldsymbol{x}^{(k_1, k_2)} \sim p(\boldsymbol{x} | \boldsymbol{e}^t, \boldsymbol{\phi}_m^{k_1})$.

And we have

$$H(\boldsymbol{x}_s, \boldsymbol{x}_t | \boldsymbol{\phi}_M, \{\boldsymbol{e}_s, \boldsymbol{e}_t\}, D) \approx \frac{1}{K_1 \cdot K_2 \cdot K_3} \sum_{k_1=1}^{K_1} \sum_{k_2^1=1}^{K_2} \sum_{k_2^2=1}^{K_2} \log p(\boldsymbol{x}^{(k_1, k_2^1)}, \boldsymbol{x}^{(k_1, k_2^2)} | \{\boldsymbol{e}_s, \boldsymbol{e}_t\}, \boldsymbol{\phi}_M),$$

where we sample graphs $\phi_m^{k_1} \sim q(\phi_m|\{e_s, e_t\}, \phi_M)$, obtain samples $x^{(k_1,k_2^1)} \sim p(x|e^s, \phi_m^{k_1})$, and obtain samples $x^{(k_1,k_2^2)} \sim p(x|e^t, \phi_m^{k_1})$.

Therefore, we add constraint on the original loss function to obtained the estimation of constraint based ELBO, that is,

$$
\begin{aligned}
\text{ELBO} = &\sum_{i=1}^{N} \sum_{j=1}^{N_S} \log p(x^{(i)}|j^{(i)}, v^{(i)}, m^{(i)}, \boldsymbol{\phi}_{m^{(i)}}^{(j)}) \\
&- \frac{1}{2} \left[ \log \frac{||\boldsymbol{\Sigma}_{all}||}{||\tilde{\boldsymbol{\Sigma}}_{all}||} - d + \text{tr}(\boldsymbol{\Sigma}_{all}^{-1}\tilde{\boldsymbol{\Sigma}}_{all}) + (\tilde{\boldsymbol{\mu}}_{all} - \boldsymbol{\mu}_{all})^T \boldsymbol{\Sigma}_{all}^{-1}(\tilde{\boldsymbol{\mu}}_{all} - \boldsymbol{\mu}_{all}) \right] \\
&- \beta \cdot \sum_{i=1}^{N_D} \sum_{m=1}^{M} \sum_{j=1}^{N_E} \left[ \lambda_1 \cdot \left[ \text{tr}\left(e^{\mathbf{E}}\right) - d \right] + \lambda_2 \cdot ||\mathbf{E}|| \right] + \lambda \cdot \left[ I(\boldsymbol{x}_s; \boldsymbol{x}_t|\boldsymbol{\phi}_M, \{\boldsymbol{e}_s, \boldsymbol{e}_t\}, D) \right].
\end{aligned}
$$

## E Proof of Theory 3

*Proof.* To begin with, we introduce two anchor variables $\boldsymbol{x}, \boldsymbol{e}$, indicating existing samples and experiments in the system, which are independent with the following experiments. Since $\boldsymbol{x}_s, \boldsymbol{x}_t$ are $\epsilon$-independent given $\boldsymbol{\phi}_M, \{\boldsymbol{e}_s, \boldsymbol{e}_t\}$ and $D$, we have:

$$
\begin{aligned}
&I(\boldsymbol{x}_s; \boldsymbol{x}_t|\boldsymbol{\phi}_M, \{\boldsymbol{e}_s, \boldsymbol{e}_t\}, D) = I(\boldsymbol{x}_s; \boldsymbol{x}_t|\boldsymbol{\phi}_M, \boldsymbol{x}, \boldsymbol{e} \cup \{\boldsymbol{e}_s, \boldsymbol{e}_t\}, D) \le \epsilon \\
\Leftrightarrow\ &H(\boldsymbol{x}_s|\boldsymbol{\phi}_M, \boldsymbol{x}, \boldsymbol{e} \cup \{\boldsymbol{e}_s, \boldsymbol{e}_t\}, D) + H(\boldsymbol{x}_t|\boldsymbol{\phi}_M, \boldsymbol{x}, \boldsymbol{e} \cup \{\boldsymbol{e}_s, \boldsymbol{e}_t\}, D) \\
&- H(\boldsymbol{x}_s, \boldsymbol{x}_t|\boldsymbol{\phi}_M, \boldsymbol{x}, \boldsymbol{e} \cup \{\boldsymbol{e}_s, \boldsymbol{e}_t\}, D) \le \epsilon,
\end{aligned}
$$

Since

$$
\begin{aligned}
I(\boldsymbol{x}_s; \boldsymbol{\phi}_M|\boldsymbol{x}, \boldsymbol{e} \cup \{\boldsymbol{e}_s, \boldsymbol{e}_t\}, D) =&H(\boldsymbol{x}_s|\boldsymbol{x}, \boldsymbol{e} \cup \{\boldsymbol{e}_s, \boldsymbol{e}_t\}, D) - H(\boldsymbol{x}_s|\boldsymbol{\phi}_M, \boldsymbol{x}, \boldsymbol{e} \cup \{\boldsymbol{e}_s, \boldsymbol{e}_t\}, D) \\
I(\boldsymbol{x}_t; \boldsymbol{\phi}_M|\boldsymbol{x}, \boldsymbol{e} \cup \{\boldsymbol{e}_s, \boldsymbol{e}_t\}, D) =&H(\boldsymbol{x}_t|\boldsymbol{x}, \boldsymbol{e} \cup \{\boldsymbol{e}_t, \boldsymbol{e}_t\}, D) - H(\boldsymbol{x}_t|\boldsymbol{\phi}_M, \boldsymbol{x}, \boldsymbol{e} \cup \{\boldsymbol{e}_s, \boldsymbol{e}_t\}, D)
\end{aligned}
$$

We have:

$$
\begin{aligned}
&I(\boldsymbol{x}_s; \boldsymbol{\phi}_M|\boldsymbol{x}, \boldsymbol{e} \cup \{\boldsymbol{e}_s, \boldsymbol{e}_t\}, D) + I(\boldsymbol{x}_t; \boldsymbol{\phi}_M|\boldsymbol{x}, \boldsymbol{e} \cup \{\boldsymbol{e}_s, \boldsymbol{e}_t\}, D) \\
=&H(\boldsymbol{x}_s|\boldsymbol{x}, \boldsymbol{e} \cup \{\boldsymbol{e}_s, \boldsymbol{e}_t\}, D) + H(\boldsymbol{x}_t|\boldsymbol{x}, \boldsymbol{e} \cup \{\boldsymbol{e}_t, \boldsymbol{e}_t\}, D) \\
&-H(\boldsymbol{x}_s|\boldsymbol{\phi}_M, \boldsymbol{x}, \boldsymbol{e} \cup \{\boldsymbol{e}_s, \boldsymbol{e}_t\}, D) - H(\boldsymbol{x}_t|\boldsymbol{\phi}_M, \boldsymbol{x}, \boldsymbol{e} \cup \{\boldsymbol{e}_s, \boldsymbol{e}_t\}, D) \\
\ge&H(\boldsymbol{x}_s, \boldsymbol{x}_t|\boldsymbol{x}, \boldsymbol{e} \cup \{\boldsymbol{e}_s, \boldsymbol{e}_t\}, D) - H(\boldsymbol{x}_s, \boldsymbol{x}_t|\boldsymbol{\phi}_M, \boldsymbol{x}, \boldsymbol{e} \cup \{\boldsymbol{e}_s, \boldsymbol{e}_t\}, D) - \epsilon \\
=&I(\boldsymbol{x}_s, \boldsymbol{x}_t; \boldsymbol{\phi}_M|\boldsymbol{x}, \boldsymbol{e} \cup \{\boldsymbol{e}_s, \boldsymbol{e}_t\}, D) - \epsilon.
\end{aligned}
$$

According to the basic mutual information property $I(A, B; C) - I(B; C) = I(A; C|B)$, we have:

$$
\begin{aligned}
&I(\boldsymbol{x} \cup \boldsymbol{x}_s; \boldsymbol{\phi}_M|\boldsymbol{e} \cup \{\boldsymbol{e}_s, \boldsymbol{e}_t\}, D) - I(\boldsymbol{x}; \boldsymbol{\phi}_M|\boldsymbol{e} \cup \{\boldsymbol{e}_s, \boldsymbol{e}_t\}, D) \\
+&I(\boldsymbol{x} \cup \boldsymbol{x}_t; \boldsymbol{\phi}_M|\boldsymbol{e} \cup \{\boldsymbol{e}_s, \boldsymbol{e}_t\}, D) - I(\boldsymbol{x}; \boldsymbol{\phi}_M|\boldsymbol{e} \cup \{\boldsymbol{e}_s, \boldsymbol{e}_t\}, D) \\
\ge&I(\boldsymbol{x} \cup \{\boldsymbol{x}_t, \boldsymbol{x}_s\}; \boldsymbol{\phi}_M|\boldsymbol{e} \cup \{\boldsymbol{e}_s, \boldsymbol{e}_t\}, D) - I(\boldsymbol{x}; \boldsymbol{\phi}_M|\boldsymbol{e} \cup \{\boldsymbol{e}_s, \boldsymbol{e}_t\}, D) - \epsilon.
\end{aligned}
$$

Thus, we have:

$$
\begin{aligned}
&I(\boldsymbol{x} \cup \boldsymbol{x}_s; \boldsymbol{\phi}_M|\boldsymbol{e} \cup \{\boldsymbol{e}_s, \boldsymbol{e}_t\}, D) + I(\boldsymbol{x} \cup \boldsymbol{x}_t; \boldsymbol{\phi}_M|\boldsymbol{e} \cup \{\boldsymbol{e}_s, \boldsymbol{e}_t\}, D) \\
\ge&I(\boldsymbol{x} \cup \{\boldsymbol{x}_t, \boldsymbol{x}_s\}; \boldsymbol{\phi}_M|\boldsymbol{e} \cup \{\boldsymbol{e}_s, \boldsymbol{e}_t\}, D) + I(\boldsymbol{x}; \boldsymbol{\phi}_M|\boldsymbol{e} \cup \{\boldsymbol{e}_s, \boldsymbol{e}_t\}, D) - \epsilon.
\end{aligned}
$$

Since different experiments are independent, we have:

$$
\begin{aligned}
&I(\boldsymbol{x} \cup \boldsymbol{x}_s; \boldsymbol{\phi}_M|\boldsymbol{e} \cup \{\boldsymbol{e}_s\}, D) + I(\boldsymbol{x} \cup \boldsymbol{x}_t; \boldsymbol{\phi}_M|\boldsymbol{e} \cup \{\boldsymbol{e}_t\}, D) \\
\ge&I(\boldsymbol{x} \cup \{\boldsymbol{x}_t, \boldsymbol{x}_s\}; \boldsymbol{\phi}_M|\boldsymbol{e} \cup \{\boldsymbol{e}_s, \boldsymbol{e}_t\}, D) + I(\boldsymbol{x}; \boldsymbol{\phi}_M|\boldsymbol{e}, D) - \epsilon.
\end{aligned}
$$

Thus, $I(\cdot; \boldsymbol{\phi}_M|\cdot, D)$ is $\epsilon$-submodular. $\qquad\square$

## F Proof of Theory 4

For clear presentation, we denote $g(\{\boldsymbol{e}_i\}_{i=1}^n) = I(\{\boldsymbol{x}_i\}_{i=1}^n; \boldsymbol{\phi}_M|\{\boldsymbol{e}_i\}_{i=1}^n, D)$, then we need to solve the following problem:

$$
\underset{\{\boldsymbol{e}_i\}_{i=1}^n}{\arg\max}\, g(\{\boldsymbol{e}_i\}_{i=1}^n), \tag{14}
$$

Suppose $S^* = \{e_i^*\}_{i=1}^n$ is the optimal solution for objective (14), and the results of the greedy method is $S = \{e_i\}_{i=1}^n$, where the experiments are sequentially determined from $e_1$ to $e_n$. We denote $S_{1:j} = \{e_i\}_{i=1}^j$, and $\Delta(e|S_{1:j}) = g(S_{1:j} \cup e) - g(S_{1:j})$, according to the greedy method, we have:

$$e_{j+1} = \arg\max_e \frac{\Delta(e|S_{1:j})}{\lambda_e},$$

where $\lambda_e$ is the cost of experiment $e$.

Based on all the above notations, we have:

$$
\begin{aligned}
g(S^*) &\le g(S^* \cup S_{1:j}) \\
&= g(S_{1:j}) + g(S_{1:j} \cup e_1^*) - g(S_{1:j}) \\
&\quad + g(S_{1:j} \cup e_1^* \cup e_2^*) - g(S_{1:j} \cup e_1^*) \\
&\quad + \ldots \\
&\quad + g(S_{1:j} \cup \{e_1^*, ..., e_n^*\}) - g(X_{1:i} \cup \{e_1^*, ..., e_{n-1}^*\}) \\
&= g(S_{1:j}) + \sum_{k=1}^n \left[g(S_{1:j} \cup \{e_1^*, ..., e_k^*\}) - g(X_{1:i} \cup \{e_1^*, ..., e_{k-1}^*\})\right] \\
&\le g(S_{1:j}) + \sum_{k=1}^n [g(S_{1:j} \cup \{e_k^*\}) - g(S_{1:j}) + \epsilon] \\
&= g(S_{1:j}) + \sum_{k=1}^n [\Delta(\{e_k^*\}|S_{1:j}) + \epsilon],
\end{aligned}
$$

where the first inequality holds because of the non-decreasing property, and the second inequality holds because of the $\epsilon$-submodular property.

Since $e_{j+1} = \arg\max_e \frac{\Delta(e|S_{1:j})}{\lambda_e}$, we have $\frac{\Delta(e|S_{1:j})}{\lambda_e} \le \frac{\Delta(e_{j+1}|S_{1:j})}{\lambda_{e_{j+1}}}$ for any $e$, thus $\Delta(e|S_{1:j}) \le \frac{\lambda_e}{\lambda_{e_{j+1}}}\Delta(e_{j+1}|S_{1:j}) \le B_\lambda \Delta(e_{j+1}|S_{1:j})$. By bringing this result into the above equation, we have:

$$
\begin{aligned}
g(S^*) &\le g(S_{1:j}) + \sum_{k=1}^n [\Delta(\{e_k^*\}|S_{1:j}) + \epsilon] \\
&\le g(S_{1:j}) + \sum_{k=1}^n [B_\lambda \Delta(e_{j+1}|S_{1:j}) + \epsilon] \\
&= g(S_{1:j}) + nB_\lambda \Delta(e_{j+1}|S_{1:j}) + n\epsilon
\end{aligned}
$$

Let $T_j = g(S^*) - g(S_{1:j})$, we have:

$$T_j - T_{j+1} = g(S_{1:j+1}) - g(S_{1:j}) = \Delta(e_{j+1}|S_{1:j}) \ge \frac{T_j - n\epsilon}{nB_\lambda}$$

Then

$$
\begin{aligned}
T_n &\le (1 - \frac{1}{nB_\lambda})T_{n-1} + \frac{\epsilon}{B_\lambda} \le [(1 - \frac{1}{nB_\lambda})]^2 T_{n-2} + (1 - \frac{1}{nB_\lambda})\frac{\epsilon}{B_\lambda} + \frac{\epsilon}{B_\lambda} \\
&\le ... \le [(1 - \frac{1}{nB_\lambda})]^n T_0 + [(1 - \frac{1}{nB_\lambda})]^{n-1}\frac{\epsilon}{B_\lambda} + ... + \frac{\epsilon}{B_\lambda}
\end{aligned}
$$

Let $B = [(1 - \frac{1}{nB_\lambda})]^{n-1}\frac{\epsilon}{B_\lambda} + ... + \frac{\epsilon}{B_\lambda} = \frac{\epsilon}{B_\lambda}\sum_{i=1}^n[(1 - \frac{1}{nB_\lambda})]^{i-1}$, and considering that $[(1 - \frac{1}{nB_\lambda})]^n = e^{-\frac{1}{B_\lambda}}$, we have:

$$g(S^*) - g(S_{1:n}) \le e^{-\frac{1}{B_\lambda}} g(S^*) + B$$

Thus, we have $g(S_{1:n}) \ge (1 - e^{-\frac{1}{B_\lambda}})g(S^*) - B$.

**Algorithm 1:** Algorithm of Licence for Single Intervention Scenario

---

**Input:** Variable set $X_V$, number of oracles $M$, cost of oracles $\Lambda$, observational data $D^O$, total budget $C$, and learning rate $\eta$.
**Output:** Causal graph $\phi_M$.

1  Initialize the model parameter $\Psi$ .
2  Optimize $\Psi$ with the training process of ELBO under $D^O$.
3  Initialize $D^I = \emptyset$.
4  **while** *Budget $C$ does not run out* **do**
5      Initialize $j^*, m^*, v^*$ and let $\zeta^* = -\infty$.
6      **for** $(j, m)$ *in* $\{1, 2, \ldots, d\} \times \{1, 2, \ldots, M\}$ **do**
7          Calculate $v^*(j, m)$ with BO.
8          **if** $f(j, v^*(j, m), m) > \zeta^*$ **then**
9              Update $j^* \leftarrow j, m^* \leftarrow m$ and $v^* \leftarrow v^*(j, m)$.
10              Update $\zeta^* \leftarrow f(j, v^*(j, m), m)$.
11          **end**
12      **end**
13      Subtract the budget with $C \leftarrow C - \lambda_{m^*}$.
14      Acquire $(j^*, v^*, m^*)$ towards the true causal graph to obtain $\boldsymbol{x}^* \sim p_m(X_V | do(X_j = v))$.
15      Update $D^I \leftarrow D^I \cup \{\boldsymbol{x}^*\}$.
16      Optimize $\Psi$ with training process of ELBO under $D^O \cup D^I$.
17  **end**
18  Sample $\phi_M$ from $p(\phi_M | D)$
19  **return** *Causal graph $\phi_M$.*

---

# G   Algorithm

The algorithm for Licence method for single interventiion scenario is shown in Algorithm 1. Moreover, the algorithm for Licence method for batch interventiion scenario is shown in Algorithm 2.

# H   More Experiments

## H.1   Experimental Settings

### H.1.1   Datasets

The details of our experimental datasets are presented as follows:

• **Erdős-Rényi (ER)** [41] graph is a random graph introduced by Paul Erdős and Alfréd Rényi. For ER graph, a graph with $n$ vertices is generated by connecting each pair of vertices with a probability $p$.

• **Scale-Free (SF)** [42] graph is a type of random graph that has a degree distribution following power law. A small number of vertices in SF graph own a large number of edges, while the vast majority of vertices have relatively few edges.

• **DREAM** [43] is the abbreviation for Dialogue for Reverse Engineering Assessments and Methods, which can estimate the reverse quality that causal discovery methods perform. Specifically, we use a biological graph generator GeneNetWeaver for our experiments, which is a real-word public dataset.

### H.1.2   Baselines

The details of experimental baselines are demonstrated as follows. We utilize DiBS [53] as our basic graph representation component. For acquisition methods, we use AIT and CBED and obtain the query tuples of node and value.

• **AIT** [44] is an active learning method that utilize f-score to select intervention queries.

---

**Algorithm 2:** Algorithm of Licence for Batch Intervention Scenario

---

**Input:** Variable set $X_V$, number of oracles $M$, cost of oracles $\boldsymbol{\Lambda}$, observational data $D^O$, total batch experiment step $T$, total budget $C$, and learning rate $\eta$.

**Output:** Causal graph $\boldsymbol{\phi}_M$.

1   Initialize the model parameter $\Psi$ .

2   Optimize $\Psi$ with training process of constraint based ELBO under $D^O$.

3   Initialize $B^I = \emptyset$

4   **for** $t$ *in* $1, 2, \ldots, T$ **do**

5     **while** *Budget $C$ does not run out* **do**

6       Initialize $j^*, m^*, v^*$ and let $\zeta^* = -\infty$.

7       **for** $(j, m)$ *in* $\{1, 2, \ldots, d\} \times \{1, 2, \ldots, M\}$ **do**

8         Calculate $v^*(j, m)$ with BO.

9         **if** $f(j, v^*(j, m), m) > \zeta^*$ **then**

10           Update $j^* \leftarrow j, m^* \leftarrow m$ and $v^* \leftarrow v^*(j, m)$.

11           Update $\zeta^* \leftarrow f(j, v^*(j, m), m)$.

12         **end**

13       **end**

14       Subtract the budget with $C \leftarrow C - \lambda_{m^*}$.

15       Update $B^I \leftarrow B^I \cup \{(j^*, v^*, m^*)\}$.

16     **end**

17     Acquire $B^I$ towards the true causal graph to obtain $\{\boldsymbol{x}^* \sim p_m(X_V | do(X_j = v))\}_{(j,v,m) \in B^I}$.

18     Update $D^I \leftarrow D^I \cup \{\boldsymbol{x}^*\}_{(j,v,m) \in B^I}$.

19     Optimize $\Psi$ with training process of constraint based ELBO under $D^O \cup D^I$.

20   **end**

21   Sample $\boldsymbol{\phi}_M$ from $p(\boldsymbol{\phi}_M | D)$

22   **return** *Causal graph $\boldsymbol{\phi}_M$.*

---

- **CBED** [8] is based on the calculation of mutual information (MI), which intend to select intervention queries with maximal MI scores after obtaining new samples under current queries.

For the batch intervention scenario, we extend above methods with greedy strategy, which can promise an lower bound for approximation with submodular property. For choosing the fidelities to query, we use two circumstances, *i.e.,* REAL and RANDOM.

- **REAL** fidelity means the model always choose the highest fidelity to conduct experiments. This strategy is aligned with classic causal discovery under active learning paradigm without multi-fidelity settings, which can just choose the most accurate samples to conduct discovery process.

- **RANDOM** fidelity means the model choose different fidelities randomly with uniform probability.

### H.1.3   Metrics

The details of experimental metrics are demonstrated as follows. We utilize SHD and AUPRC to reflect the topological structure discovering performance, and design RMSE to reflex the predicting performance of functional relations.

- **SHD** [45] is the abbreviation for Structural Hamming Distance, and it estimate the topological structure by counting the number of different edges on adjacency matrix. We calculate the expectation of SHD under multiple graph samplings.

- **AUPRC** [46] is the area under precision-recall curve, where we consider entities on the adjacency matrix as binary classification problem. The AUPRC is also under the expectation for multiple graph sampling.

- **RMSE** is designed for estimating the performance of grasping functional relations. We obtain several samples from the true causal graph, and let our model and the true causal function to conduct forward process respectively, then calculate the RMSE between the two results. We calculate RMSE by sampling graphs for multiple times.

Table 1: The left table demonstrate the details of the configuration of device and platform. The right table shows the details of time cost on computation.

| Name | Details | | Model | Time (secs) |
|---|---|---|---|---|
| CPU | Intel Xeon Platinum 8350C 2.60GHz | | AIT-REAL | 7.686 |
| GPU | RTX A5000 (24GB) | | AIT-RANDOM | 7.451 |
| Memory | 42GB RAM | | CBED-REAL | 7.998 |
| Python | Version 3.8 | | CBED-RANDOM | 7.989 |
| Java | Version 1.8.0 (Necessary for DREAM) | | Licence | 8.320 |

Table 2: The details of experimental settings.

| Name | Explanation | Value |
|---|---|---|
| budget | The total budget for interventional experiments, (*i.e.,* C). | 10/20/30/40/50 |
| oracle number | The number of oracles, (*i.e.,* $M$) | 3 |
| oracle cost | The cost for each oracle, (*i.e.,* $\mathbf{\Lambda}$) | 2, 8, 32 |
| oracle noise | The extra additive noise for each oracle. | 0.04, 0.02, 0.00 |
| observation number | The number of observational samples. | 1000 |
| expect edge number | The number of expect edges. | 2 |
| additive noise | The value of additive noise during data generations. | 0.01 |

## H.2 Simulation of Oracles with Different Fidelities

For a given intervention $(j, v)$, suppose we have $M$ oracles $\{\phi_1, \phi_2, ..., \phi_M\}$, then the experiment results $\{x_{j,v,1}, x_{j,v,2}, ..., x_{j,v,M}\}$ are specified as follows:

$$x_{j,v,m} = x_{j,v,M} + \delta_m,$$
$$\delta_m \sim N(0, \sigma_m),$$

where $x_{j,v,M}$ is the ground truth, which can be directly obtained from the datasets. Since $x_{j,v,m}$ is correlated with $x_{j,v,M}$ by the first line, their underlying oracles $\phi_m$ and $\phi_M$ are correlated in our simulation. In our experiment, we set $\delta_1 > \delta_2 > ... > \delta_M = 0$. Suppose the cost of $\phi_m$ as $\lambda_m$, then we set $\lambda_1 < \lambda_2 < ... < \lambda_M$.

## H.3 Details of Configurations and Computation

The details of the configurations of device and platform are demonstrate in Table 1(left). We will show the details of the time cost on computation. We measure the time cost on the generation of each intervention per fidelity for all models, and the results are shown in Figure 1(right). We find that our method cost a little more than the baselines, which is probably due to the more complex sampling process in our model.

We also show the details of experimental settings for our overall experiments in Table 2. We carefully tune the hyper-parameters for baselines and our model, and the final values can be obtained in the configuration file in our codes.

## H.4 Experiments on DREAM Dataset

We conduct experiments on a real-world biological dataset, called DREAM. Note that, DREAM does not support the calculation of RMSE, because of the lack of interface in this real-world dataset. We use two sub-datasets *Ecoli* and *Yeast* as our true causal graphs. The results are shown in Figure 4. We find that our model outperforms that other baselines on both *Ecoli* and *Yeast*, and both single and batch intervention scenario.

## H.5 Experiments on More Nodes

In this section, we conduct further experiments on datasets with more nodes. We extend the number of nodes from 10 to 20, and experiment on the ER graph. The results are shown in Figure 3. We find that our model is still effective on the scenario of more nodes, and is better than baselines.

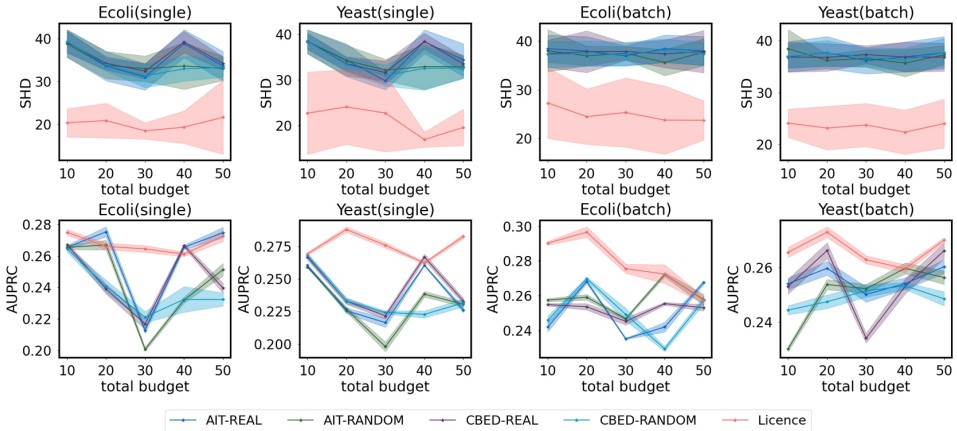

Figure 4: The performance among models on DREAM datasets with different datasets and budgets. Lower SHD, RMSE indicate better performances. We conduct each experiment for ten times, and report the average performances and error bars.

Table 3: SHD results of 20 nodes graphs on different budgets. Lower SHD indicates better performances. We conduct each experiment for ten times, and report average performances and error bars.

| Model | Budget(10) | Budget(20) | Budget(30) | Budget(40) | Budget(50) |
|---|---|---|---|---|---|
| AIT-REAL | 63.36±4.89 | 64.36±5.18 | 64.53±6.83 | 63.28±4.86 | 64.35±5.19 |
| AIT-RANDOM | 63.62±4.61 | 62.16±5.75 | 64.60±5.23 | 66.87±6.47 | 63.53±5.27 |
| DiBS-REAL | 63.58±6.35 | 61.50±7.69 | 63.50±6.86 | 63.56±6.34 | 61.45±7.69 |
| DiBS-RANDOM | 63.68±6.77 | 65.07±6.41 | 63.91±7.14 | 63.99±4.46 | 63.86±3.00 |
| Licence | 49.67±11.64 | 49.61±8.08 | 55.68±8.63 | 51.34±11.24 | 51.36±9.11 |

## H.6 Supplementary Experiments on MAE Metric

We further compare our model with the baselines based on the Mean Absolute Error (MAE) metric. The experiments are conducted based on ER with different total budgets. The results are presented in Table 4.

Table 4: Results of the metric MAE (%).

| Model | Budget(10) | Budget(20) | Budget(30) | Budget(40) | Budget(50) |
|---|---|---|---|---|---|
| AIT-REAL | 3.46±0.01 | 2.43±0.01 | 2.63±0.01 | 3.46±0.01 | 2.42±0.01 |
| AIT-RANDOM | 3.71±0.02 | 2.42±0.01 | 2.82±0.01 | 2.68±0.00 | 2.54±0.01 |
| CBED-REAL | 3.73±0.01 | 2.56±0.00 | 2.54±0.00 | 3.69±0.01 | 2.52±0.00 |
| CBED-RANDOM | 4.00±0.02 | 2.53±0.00 | 2.88±0.00 | 3.14±0.01 | 2.70±0.01 |
| Licence | **2.06±0.00** | **2.20±0.01** | **1.70±0.00** | **1.77±0.00** | **2.07±0.00** |

The results indicate that our model surpasses the baselines in terms of MAE. This further provides evidence that the superior performance of our model is a general conclusion.

## H.7 Supplementary Experiments on Different Oracle Settings

To demonstrate that our model is generally effective for different cost- and noisy-levels. We conduct experiments based on different sets of oracles. In specific, the experiments are conducted based on the following settings in Table 5. The results are presented in Figure 5.

From the results, we can see that our model can always perform better than the baselines on different sets of oracles.

Table 5: Settings for different cost and noise levels.

| | Setting 1 | | | Setting 2 | |
|---|---|---|---|---|---|
| Oracle ($m$) | cost ($\lambda_m$) | noise ($\sigma_m$) | Oracle ($m$) | cost ($\lambda_m$) | noise ($\sigma_m$) |
| 1 | 2 | 0.04 | 1 | 2 | 0.04 |
| 2 | 8 | 0.02 | 2 | 8 | 0.02 |
| 3 | 32 | 0 | 3 | 16 | 0 |
| | Setting 3 | | | Setting 4 | |
| Oracle ($m$) | cost ($\lambda_m$) | noise ($\sigma_m$) | Oracle ($m$) | cost ($\lambda_m$) | noise ($\sigma_m$) |
| 1 | 2 | 0.04 | 1 | 2 | 0.04 |
| 2 | 4 | 0.02 | 2 | 8 | 0.02 |
| 3 | 32 | 0 | 3 | 32 | 0 |
| | Setting 5 | | | Setting 6 | |
| Oracle ($m$) | cost ($\lambda_m$) | noise ($\sigma_m$) | Oracle ($m$) | cost ($\lambda_m$) | noise ($\sigma_m$) |
| 1 | 2 | 0.08 | 1 | 2 | 0.04 |
| 2 | 8 | 0.02 | 2 | 8 | 0.03 |
| 3 | 32 | 0 | 3 | 32 | 0 |

## H.8 Supplementary Experiments on Regularization Coefficient $\lambda$

In our model, the $\epsilon$-independent constraint in Equation 10 is an important contribution. In the optimization process, we convert it to the objective. We study the influence of the coefficient $\lambda$ by tuning it in the range of $[10^{-5}, 10^{-6}, 10^{-7}, 10^{-8}, 10^{-9}]$. The results are presented in Figure 6.

From the results, we can see the performances of our model varies as we set different $\lambda$'s. In most cases, the best performance is achieved when $\lambda$ is moderated (not too large or too small).

## H.9 Supplementary Experiments on Ablation Studies

To study whether the correlation modeling between different oracles are necessary, we first build a variant of our model by regarding different oracles as independent components, that is, removing the links between different $\phi$'s in Figure 1, and then compare our model with such variant. The results are presented in Table 6.

Table 6: Results of the Licence and Licence without the cascaded relation.

| Metrics | Model | Budget(10) | Budget(20) | Budget(30) | Budget(40) | Budget(50) |
|---|---|---|---|---|---|---|
| SHD ↓ | Licence (w/o rel) | **14.61±2.30** | 15.25±2.74 | 15.47±3.74 | 15.17±4.30 | 18.52±4.65 |
| | Licence | 14.67±2.98 | **14.73±2.04** | **14.29±3.20** | **14.83±3.24** | **15.02±3.01** |
| AUPRC (%) ↑ | Licence (w/o rel) | 28.96±1.28 | **35.05±1.35** | 31.74±1.50 | 37.85±2.03 | 33.70±4.36 |
| | Licence | **35.75±2.13** | 25.12±2.01 | **41.79±1.71** | **40.89±2.92** | **41.76±3.19** |
| RMSE (%) ↓ | Licence (w/o rel) | 3.34±0.04 | 3.10±0.02 | 2.46±0.00 | **2.61±0.00** | 2.83±0.00 |
| | Licence | **2.82±0.01** | **2.75±0.00** | **2.40±0.00** | 2.68±0.01 | **2.69±0.01** |

We can see, in most cases, our model can achieve better performance than its variant without modeling the correlations between different oracles.

# I Potentially Negative Social Impact

Causal discovery focuses on understanding causal relationships between variables. While causal discovery has the potential to bring about positive social impacts, it is important to consider both the positive and negative implications of its applications. In this response, I will focus on the negative impact of causal discovery.

• **Reductionism and Oversimplification.** Causal discovery techniques often aim to identify simple cause-and-effect relationships. However, complex social phenomena often involve a multitude of interconnected factors, making it difficult to capture the full complexity of the system. Relying solely on causal discovery may lead to oversimplification and reductionism, neglecting the nuanced interactions between variables.

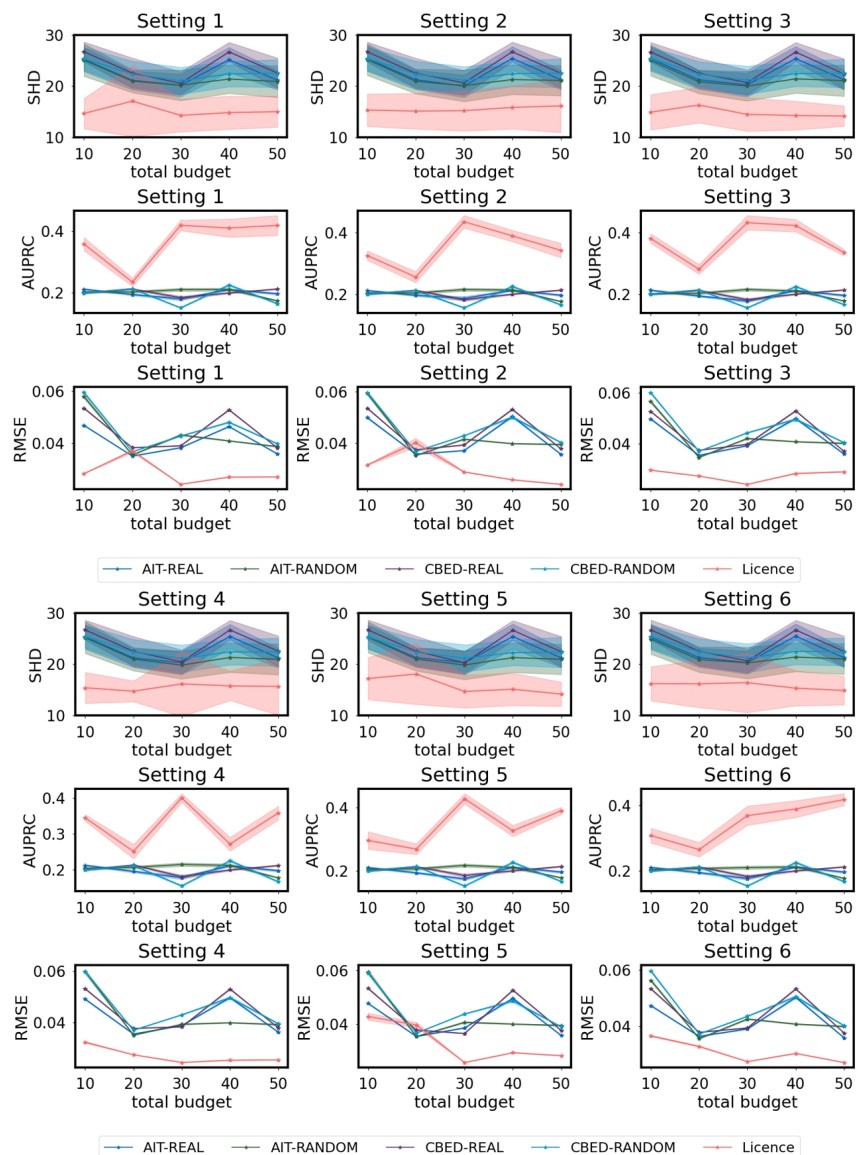

Figure 5: Results of different settings of oracles in terms of cost and noise.

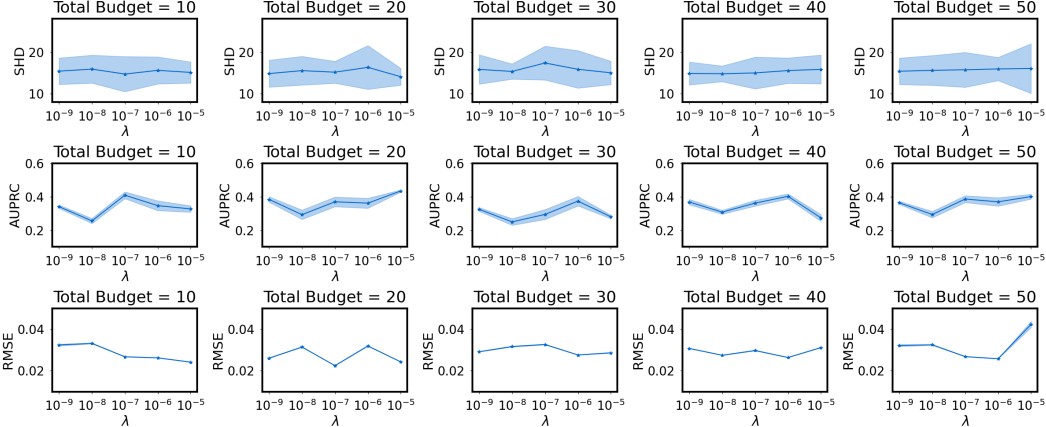

Figure 6: Results of extensive experiments on regularization $\lambda$.

- **Ethical Concerns.** Causal discovery can involve analyzing sensitive data, such as personal information or medical records. If not handled carefully, the use of this data can raise significant ethical concerns related to privacy, consent, and potential discrimination. Improper handling of data could lead to violations of privacy and unfair treatment of individuals or groups.

- **Overreliance on Correlation.** Causal discovery often relies on identifying statistical correlations between variables. However, correlation does not imply causation, and there is a risk of mistakenly inferring causal relationships based solely on correlation. Overreliance on such methods can lead to erroneous conclusions, leading to misguided decision-making and ineffective interventions.

- **Social Bias and Inequality.** Causal discovery relies on the data used for analysis, which can reflect existing biases and inequalities present in society. If the data used is biased, the causal relationships discovered may perpetuate or exacerbate existing social inequalities. Causal discovery methods need to be sensitive to potential biases and strive for fairness and inclusivity in both data collection and analysis.

In conclusion, while causal discovery holds promise in understanding complex systems, it is crucial to consider its potential negative impacts. Oversimplification, ethical concerns, overreliance on correlation, and social bias are all factors that need to be addressed to ensure responsible and beneficial applications of causal discovery. It is essential to approach this field with caution and incorporate broader societal considerations to mitigate the negative impacts and harness its potential for positive social change.

## J   Limitations

In this section, we analyze the limitations of our work, including sub-optimum of greedy method, estimation of mutual information, and scale of causal graph.

- **Sub-optimum of greedy method.** For the whole process of active causal discovery, the interventional data will be acquired successively in the greedy manner. Therefore, even if the strategy for acquisition is the optimal for each current step, the entire trajectory of causal discovery is sub-optimal. A possible solution is finding the best acquisition trajectory by reinforcement learning.

- **Estimation of mutual information.** For different circumstances, the costs, accuracy and data scale can be various. Therefore, the scale of mutual information can be affected as well. So it is important to adjust hyper-parameters accordingly.

- **Scale of causal graph.** It is a classic problem for causal discovery that most existing methods suffer from the difficulty in extending to large-scale graphs. The efficiency and effectiveness are supposed to be further improved, and we will optimize our model as well.

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
