$$
\begin{aligned}
B =&\mathbb{E}_{\mathbf{\Phi}\sim q(\mathbf{\Phi})}\left[\log q(\mathbf{\Phi})-\log p(\mathbf{\Phi})\right]\\
=&\int_{\mathbf{\Phi}}\mathcal{N}(\tilde{\boldsymbol{\mu}}_{all},\tilde{\mathbf{\Sigma}}_{all})\log\frac{\mathcal{N}(\tilde{\boldsymbol{\mu}}_{all},\tilde{\mathbf{\Sigma}}_{all})}{\prod_{m=1}^{M}e^{-\beta\cdot f(\mathbf{S}_m,\mathbf{T}_m)}\cdot\mathcal{N}(\boldsymbol{\mu}_{all},\mathbf{\Sigma}_{all})}\,d\mathbf{\Phi}\\
=&\int_{\mathbf{\Phi}}\mathcal{N}(\tilde{\boldsymbol{\mu}}_{all},\tilde{\mathbf{\Sigma}}_{all})\log\frac{\mathcal{N}(\tilde{\boldsymbol{\mu}}_{all},\tilde{\mathbf{\Sigma}}_{all})}{\mathcal{N}(\boldsymbol{\mu}_{all},\mathbf{\Sigma}_{all})}\,d\mathbf{\Phi}+\int_{\mathbf{\Phi}}\mathcal{N}(\tilde{\boldsymbol{\mu}}_{all},\tilde{\mathbf{\Sigma}}_{all})\log\frac{1}{\prod_{m=1}^{M}e^{-\beta\cdot f(\mathbf{S}_m,\mathbf{T}_m)}}\,d\mathbf{\Phi}\\
=&\underbrace{\mathrm{KL}[\mathcal{N}(\tilde{\boldsymbol{\mu}}_{all},\tilde{\mathbf{\Sigma}}_{all})||\mathcal{N}(\boldsymbol{\mu}_{all},\mathbf{\Sigma}_{all})]}_{C}+\underbrace{\int_{\mathbf{\Phi}}\mathcal{N}(\tilde{\boldsymbol{\mu}}_{all},\tilde{\mathbf{\Sigma}}_{all})\log\prod_{m=1}^{M}e^{\beta\cdot f(\mathbf{S}_m,\mathbf{T}_m)}\,d\mathbf{\Phi}}_{D}.
\end{aligned}
$$

According to KL divergence of Gaussian distribution, we can calculate $C$ in a close-form.

$$
\begin{aligned}
C =&\mathrm{KL}[\mathcal{N}(\tilde{\boldsymbol{\mu}}_{all},\tilde{\mathbf{\Sigma}}_{all})||\mathcal{N}(\boldsymbol{\mu}_{all},\mathbf{\Sigma}_{all})]\\
=&\frac{1}{2}\left[\log\frac{||\mathbf{\Sigma}_{all}||}{||\tilde{\mathbf{\Sigma}}_{all}||}-d+\mathrm{tr}(\mathbf{\Sigma}_{all}^{-1}\tilde{\mathbf{\

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

\Leftrightarrow \quad & H(\boldsymbol{x}_s | \phi_M, \boldsymbol{x}, \boldsymbol{e} \cup \{e_s, e_t\}, D) + H(\boldsymbol{x}_t | \phi_M, \boldsymbol{x}, \boldsymbol{e} \cup \{e_s, e_t\}, D) \\
& - H(\boldsymbol{x}_s, \boldsymbol{x}_t | \phi_M, \boldsymbol{x}, \boldsymbol{e} \cup \{e_s, e_t\}, D) \leq \epsilon,
\end{aligned}
$$

Since

$$
\begin{aligned}
I(\boldsymbol{x}_s; \phi_M | \boldsymbol{x}, \boldsymbol{e} \cup \{e_s, e_t\}, D) = & H(\boldsymbol{x}_s | \boldsymbol{x}, \boldsymbol{e} \cup \{e_s, e_t\}, D) - H(\boldsymbol{x}_s | \phi_M, \boldsymbol{x}, \boldsymbol{e} \cup \{e_s, e_t\}, D) \\
I(\boldsymbol{x}_t; \phi_M | \boldsymbol{x}, \boldsymbol{e} \cup \{e_s, e_t\}, D) = & H(\boldsymbol{x}_t | \boldsymbol{x}, \boldsymbol{e} \cup \{e_t, e_t\}, D) - H(\boldsymbol{x}_t | \phi_M, \boldsymbol{x}, \boldsymbol{e} \cup \{e_s, e_t\}, D)
\end{aligned}
$$

We have:

$$
\begin{aligned}
& I(\boldsymbol{x}_s; \phi_M | \boldsymbol{x}, \boldsymbol{e} \cup \{e_s, e_t\}, D) + I(\boldsymbol{x}_t; \phi_M | \boldsymbol{x}, \boldsymbol{e} \cup \{e_s, e_t\}, D) \\
= & H(\boldsymbol{x}_s | \boldsymbol{x}, \boldsymbol{e} \cup \{e_s, e_t\}, D) + H(\boldsymbol{x}_t | \boldsymbol{x}, \boldsymbol{e} \cup \{e_t, e_t\}, D) \\
& - H(\boldsymbol{x}_s | \phi_M, \boldsymbol{x}, \boldsymbol{e} \cup \{e_s, e_t\}, D) - H(\boldsymbol{x}_t | \phi_M, \boldsymbol{x}, \boldsymbol{e} \cup \{e_s, e_t\}, D) \\
\geq & H(\boldsymbol{x}_s, \boldsymbol{x}_t | \boldsymbol{x}, \boldsymbol{e} \cup \{e_s, e_t\}, D) - H(\boldsymbol{x}_s, \boldsymbol{x}_t | \phi_M, \boldsymbol{x}, \boldsymbol{e} \cup \{e_s, e_t\}, D) - \epsilon \\
= & I(\boldsymbol{x}_s, \boldsymbol{x}_t; \phi_M | \boldsymbol{x}, \boldsymbol{e} \cup \{e_s, e_t\}, D) - \epsilon.
\end{aligned}
$$

According to the basic mutual information property $I(A, B; C) - I(B; C) = I(A; C | B)$, we have:

$$
\begin{aligned}
& I(\boldsymbol{x} \cup \boldsymbol{x}_s; \phi_M | \boldsymbol{e} \cup \{e_s, e_t\}, D) - I(\boldsymbol{x}; \phi_M | \boldsymbol{e} \cup \{e_s, e_t\}, D) \\
+ & I(\boldsymbol{x} \cup \boldsymbol{x}_t; \phi_M | \boldsymbol{e} \cup \{e_s, e_t\}, D) - I(\boldsymbol{x}; \phi_M | \boldsymbol{e} \cup \{e_s, e_t\}, D) \\
\geq & I(\boldsymbol{x} \cup \{\boldsymbol{x}_t, \boldsymbol{x}_s\}; \phi_M | \boldsymbol{e} \cup \{e_s, e_t\}, D) - I(\boldsymbol{x}; \phi_M | \boldsymbol{e} \cup \{e_s, e_t\}, D) - \epsilon.
\end{aligned}
$$

Thus, we have:

$$
\begin{aligned}
& I(\boldsymbol{x} \cup \boldsymbol{x}_s; \phi_M | \boldsymbol{e} \cup \{e_s, e_t\}, D) + I(\boldsymbol{x} \cup \boldsymbol{x}_t; \phi_M | \boldsymbol{e} \cup \{e_s, e_t\}, D) \\
\geq & I(\boldsymbol{x} \cup \{\boldsymbol{x}_t, \boldsymbol{x}_s\}; \phi_M | \boldsymbol{e} \cup \{e_s, e_t\}, D) + I(\boldsymbol{x}; \phi_M | \boldsymbol{e} \cup \{e_s, e_t\}, D) - \epsilon.
\end{aligned}
$$

Since different experiments are independent, we have:

$$
\begin{aligned}
& I(\boldsymbol{x} \cup \boldsymbol{x}_s; \phi_M | \boldsymbol{e} \cup \{e_s\}, D) + I(\boldsymbol{x} \cup \boldsymbol{x}_t; \phi_M | \boldsymbol{e} \cup \{e_t\}, D) \\
\geq & I(\boldsymbol{x} \cup \{\boldsymbol{x}_t, \boldsymbol{x}_s\}; \phi_M | \boldsymbol{e} \cup \{e_s, e_t\}, D) + I(\boldsymbol{x}; \phi_M | \boldsymbol{e}, D) - \epsilon.
\end{aligned}
$$

Thus, $I(\cdot; \phi_M | \cdot, D)$ is $\epsilon$-submodular. $\qquad \square$

## F  Proof of Theory 4

For clear presentation, we denote $g(\{e_i\}_{i=1}^n) = I(\{\boldsymbol{x}_i\}_{i=1}^n; \phi_M | \{e_i\}_{i=1}^n, D)$, then we need to solve the following problem:

$$
\underset{\{\boldsymbol{e}_i\}_{i=1}^n}{\arg\max} \, g(\{e_i\}_{i=1}^n), \tag{4}
$$

198 Suppose $S^* = \{e_i^*\}_{i=1}^n$ is the optimal solution for objective (4), and the results of the greedy method
199 is $S = \{e_i\}_{i=1}^n$, where the experiments are sequentially determined from $e_1$ to $e_n$. We denote
200 $S_{1:j} = \{e_i\}_{i=1}^j$, and $\Delta(e|S_{1:j}) = g(S_{1:j} \cup e) - g(S_{1:j})$, according to the greedy method, we have:

$$e_{j+1} = \arg\max_e \frac{\Delta(e|S_{1:j})}{\lambda_e},$$

201 where $\lambda_e$ is the cost of experiment $e$.

202 Based on all the above notations, we have:

$$
\begin{aligned}
g(S^*) &\leq g(S^* \cup S_{1:j}) \\
&= g(S_{1:j}) + g(S_{1:j} \cup e_1^*) - g(S_{1:j}) \\
&\quad + g(S_{1:j} \cup e_1^* \cup e_2^*) - g(S_{1:j} \cup e_1^*) \\
&\quad + \ldots \\
&\quad + g(S_{1:j} \cup \{e_1^*, ..., e_n^*\}) - g(X_{1:i} \cup \{e_1^*, ..., e_{n-1}^*\}) \\
&= g(S_{1:j}) + \sum_{k=1}^n \left[ g(S_{1:j} \cup \{e_1^*, ..., e_k^*\}) - g(X_{1:i} \cup \{e_1^*, ..., e_{k-1}^*\}) \right] \\
&\leq g(S_{1:j}) + \sum_{k=1}^n \left[ g(S_{1:j} \cup \{e_k^*\}) - g(S_{1:j}) + \epsilon \right] \\
&= g(S_{1:j}) + \sum_{k=1}^n \left[ \Delta(\{e_k^*\}|S_{1:j}) + \epsilon \right],
\end{aligned}
$$

203 where the first inequality holds because of the non-decreasing property, and the second inequality
204 holds because of the $\epsilon$-submodular property.

205 Since $e_{j+1} = \arg\max_e \frac{\Delta(e|S_{1:j})}{\lambda_e}$, we have $\frac{\Delta(e|S_{1:j})}{\lambda_e} \leq \frac{\Delta(e_{j+1}|S_{1:j})}{\lambda_{e_{j+1}}}$ for any $e$, thus $\Delta(e|S_{1:j}) \leq$
206 $\frac{\lambda_e}{\lambda_{e_{j+1}}}\Delta(e_{j+1}|S_{1:j}) \leq B_\lambda \Delta(e_{j+1}|S_{1:j})$. By bringing this result into the above equation, we have:

$$
\begin{aligned}
g(S^*) &\leq g(S_{1:j}) + \sum_{k=1}^n \left[ \Delta(\{e_k^*\}|S_{1:j}) + \epsilon \right] \\