# OpenReview forum: "Bayesian Active Causal Discovery with Multi-Fidelity Experiments"
_NeurIPS.cc/2023/Conference — NeurIPS 2023 poster_

### Official Review · Reviewer_ug4Y · 2023-07-06

**Soundness:** 3 good
**Presentation:** 1 poor
**Contribution:** 3 good
**Rating:** 5
**Confidence:** 4

**Summary:**

The authors introduce the task of active causal discovery with multi-fidelity oracles which they go onto show is superior to many state-of-the-art methods for active causal discovery. They demonstrate their method in multiple settings and compare it alongside the aforementioned methods, to show superior performance almost across the board.

**Strengths:**

Please see the Questions section for the main review.

**Weaknesses:**

Please see the Questions section for the main review.

- The paper is very hard to follow for a number of reasons. First, the language is lacking in a lot of places and complex sentence construction makes following the arguments presented, somewhat harder still (this sentence for example on line 256: "Higher-fidelity models are more accuracy but cost much" should presumably be 'higher-fidelity models are more accurate but they cost more'). To help the authors I recommend Grammarly which is a great free tool that can pickup many of the language mistakes that are present in the manuscript. Second, the authors jump a lot in their narrative thus making quite difficult to understand the point they are trying to make. The manuscript would do well to be proof-read a few more times by different people. Most all the manuscript lacks a red thread, taking the reader through the paper one coherent argument at time, thus building a narrative as it goes along. At the end of section three, I still do not quite understand where the authors are heading with their method.

**Questions:**

## Abstract

- A very good abstract which I enjoyed reading but it is too long, much too long. The abstract is merely meant to summarise what you are doing in the paper, and pick out some main results and then leave the rest for the paper. I suggest about half the length of what you currently have.

## Introduction

- Typo: Markov equivalence class (not ‘equivalent’) - line 26.
- The paragraph starting on line 37 is excellent, very interesting. I would be interested to know if the authors have more information on this particular topic discussed in the paragraph i.e. do clinicians actually take these considerations intom account? References to that effect would be very informative.
- Do you know if there are more sources on multi-objective (multi-target), line 54, studies which you mention? This is an open problem for much of causal inference as you rightly know. It would be effectual to add such sources if you have them.

## Preliminary

- Typo / language: the section tile should be ‘Preliminaries’ as it is plural as you are describing multiple ideas here.
- Reference Pearl when you first introduce the SCM on line 78. It was his idea after all and he deserves credit whenever it is raised in academic context.
- You will need to give more information on the SCM. It is a strict definition which consists of a four-tuple. See Def 7.1.1 of Pearl’s book on causality. At present you are not giving the correct definition as there are pieces missing w.r.t. your description of the SCM.
- Your language for describing the adjacency matrix is a bit confusing. If i and j are denoted by a 1 in the matrix that means that there is an arc from i to j in the causal diagram or that there is an edge which exists i and points to j. Phrase it in this relation instead of saying a ‘causal relation’ as you have done, as you are fundamentally describing a DAG.
- Line 102: you cannot say ‘using causal language’ to define the do-operator because the do-operator is not unique for modelling interventions, there are many other frameworks such as the potential outcome framework. Please revise and provide adequate references.
- You have not discussed identifiability in section 2.2 which seems somewhat important because at present you are assuming that all your interventional distributions can be calculated from the observational distribution which is not always the case.

## The license model

- Explain what $H$ is in equation 2.
- Figure 1 does not make sense. What does the dashed arrow represent? Why does it point to another arrow? You are going to have to redo that figure with much more detail and clarification because this reviewer cannot make heads or tails of it. It looks like a first-order Markov model but I am sure that that has nothing to do with the problem you are studying but equally I do not quite understand what you are trying to show with that figure.

## Experiments

- Line 276: you cannot say “state-of-arts”. That phrasing is used as so: ‘the state-of-the-art model’ - it is an adjective that refers to a noun, in this case ‘model’.
- This section could be improved by adding the experimental topologies that are being targeted by the different causal discovery methods, to give the reader an idea of the complexity involved.

**Limitations:**

Please see the Questions section for the main review. I would be happy to increase my score but my concerns above would have to adequately addressed in order to do so. This is not nitpicking (which it surely will be construed as from the authors) but the paper is genuinely very difficult to follow for the reasons listed but could be much improved with some simple proof-reading and revisions.

---

> ### Author Rebuttal · Authors · 2023-08-09
>
> For reviewer ug4Y:
>
> Thanks so much for your detailed comments. We will try our best to alleviate your concerns in the following.
>
> > Reviews: **Abstract**
> >
> > A very good abstract which I enjoyed reading but it is too long, much too long. The abstract is merely meant to summarise what you are doing in the paper, and pick out some main results and then leave the rest for the paper. I suggest about half the length of what you currently have.
>
> Thanks for the comments. According to your comments, we plan to change our abstract by removing the discussions on the differences between single- and multi-fidelity oracles. In addition, we will also remove the challenges that we may face for multi-fidelity ACD. We believe the removed contents can be cleared delivered in the introduction.
>
> > Concerns on the **Introduction**
>
> 1. For the typo "Markov equivalence class": we will correct it in the final version.
>
> 2. For the topic discussed in the paragraph starting on line 37: we believe this example is very practical, and there are indeed many studies on simulation based drug-disease relation discovery. For example, [1-9] are all about this topic. We will add these references into the final paper.
>
> 3. For the termination of "multi-objective (multi-target)": there are indeed many papers on multi-target intervention, for example, [10-12]. We will add them into the final version.
>
> > Concerns on the **Preliminary**
>
> 1. For the typo and language problems: we will revise them carefully according to your suggestions.
>
> 2. For the reference of Pearl: we are sorry for this point. In the final version, we will definitely cite Pearl's paper.
>
> 3. For the definition of SCM: in the final version, we will present the correct definition according to Def 7.1.1 of Pearl’s book on causality.
>
> 4. For the adjacency matrix: thanks so much for your correction, in the final version, we will change "causal relation from $X_i$ to $X_j$" to "if $E_{ij} =1$, then there is an edge which exists from $i$ to $j$".
>
> 5. For the do-operator: we will remove the term "using causal language", and add Pearl's papers as references for the do-operator.
>
> 6. For the identifiability problem, we will add more discussions on the assumptions usually leveraged to derive identifiability.
>
>
>
> > Concerns on the **The license model**
>
>
> 1. For $H$: it is the notation of entropy, that is, $H(x) = E_x[log x]$.
>
> 2. For Figure 1: the dashed arrow means that $x$ is determined by $\phi$ and $(j,v)$ jointly. It should point to $x$. In the additional submitted one-page pdf, we have replotted this figure according to your comments in **Figure 1**.
>
>
>
> > Concerns on the **Experiments**
>
> Thanks for these comments.
>
> 1. For the term "state-of-arts": we will revise it in the final version.
>
> 2. For the topologies: in the submitted one-page pdf, we have added the structures of the real causal graph in **Figure 4**.
>
>
>
> **References**
>
> [1] Li, J., & Lu, Z. (2013). Pathway-based drug repositioning using causal inference. BMC bioinformatics, 14(16), 1-10.
>
> [2] Yang, J., Li, Z., Fan, X., & Cheng, Y. (2014). Drug–disease association and drug-repositioning predictions in complex diseases using causal inference–probabilistic matrix factorization. Journal of chemical information and modeling, 54(9), 2562-2569.
>
> [3] Peyvandipour, A., Saberian, N., Shafi, A., Donato, M., & Draghici, S. (2018). A novel computational approach for drug repurposing using systems biology. Bioinformatics, 34(16), 2817-2825.
>
> [4] Kelly, J., Berzuini, C., Keavney, B., Tomaszewski, M., & Guo, H. (2022). A review of causal discovery methods for molecular network analysis. Molecular Genetics & Genomic Medicine, 10(10), e2055.
>
> [5] Domingo-Fernández, D., Gadiya, Y., Patel, A., Mubeen, S., Rivas-Barragan, D., Diana, C. W., ... & Colluru, V. (2022). Causal reasoning over knowledge graphs leveraging drug-perturbed and disease-specific transcriptomic signatures for drug discovery. PLoS computational biology, 18(2), e1009909.
>
> [6] Chen, H., Zhang, Z., & Peng, W. (2017). miRDDCR: a miRNA-based method to comprehensively infer drug-disease causal relationships. Scientific reports, 7(1), 15921.
>
> [7] Subpaiboonkit, S., Li, X., Zhao, X., Scells, H., & Zuccon, G. (2019, November). Causality discovery with domain knowledge for drug-drug interactions discovery. In International Conference on Advanced Data Mining and Applications (pp. 632-647). Cham: Springer International Publishing.
>
> [8] Subpaiboonkit, S., Li, X., Zhao, X., & Zuccon, G. (2022, November). Causality Discovery Based on Combined Causes and Multiple Causes in Drug-Drug Interaction. In International Conference on Advanced Data Mining and Applications (pp. 53-66). Cham: Springer Nature Switzerland.
>
> [9] Tian, X. Y., & Liu, L. (2012). Drug discovery enters a new era with multi-target intervention strategy. *Chinese journal of integrative medicine*, *18*(7), 539-542.
>
> [10] Tigas, P., Annadani, Y., Ivanova, D. R., Jesson, A., Gal, Y., Foster, A., & Bauer, S. (2023, May). Differentiable Multi-Target Causal Bayesian Experimental Design. In *ICLR 2023-Machine Learning for Drug Discovery workshop*.
>
> [11] Guerrero, L. R., Ho, J., Christie, C., Harwood, E., Pfund, C., Seeman, T., ... & Wallace, S. P. (2017, December). Using collaborative approaches with a multi-method, multi-site, multi-target intervention: evaluating the National Research Mentoring Network. In *BMC proceedings* (Vol. 11, No. 12, pp. 193-200). BioMed Central.
>
> [12] Somvanshi, R. K., Zou, S., Kadhim, S., Padania, S., Hsu, E., & Kumar, U. (2022). Cannabinol modulates neuroprotection and intraocular pressure: A potential multi-target therapeutic intervention for glaucoma. *Biochimica et Biophysica Acta (BBA)-Molecular Basis of Disease*, *1868*(3), 166325.

---

> ### Author Response · Authors · 2023-08-13
> **Rebuttal by Authors**
>
> Dear reviewer ug4Y,
>
> Thanks again for your comments, which, we believe, are very important to improve our paper.
>
> In the rebuttal, we try our best to answer your questions one by one.  We will definitely improve our writing to make it more comfortable for a wider audience.  We believe most of the comments can be easily addressed in the final version.
>
> If you have further questions, we are very happy to discuss more about them.

---

> ### Comment · Reviewer_ug4Y · 2023-08-16
>
> Thanks again for a very interesting paper and the considerable effort that you have put in, as well for taking the time to respond to my comments. I am writing to confirm that I've read the author's rebuttal.

---

> > ### Author Response · Authors · 2023-08-16
> > **Thanks for the response**
> >
> > Dear reviewer ug4Y, thanks very much for your kind reply. We believe most of your concerns are about presentation. If our responses have alleviated your concerns, is it possible to consider adjusting your score? We really believe these concerns do not influence the major contribution of this paper, which can be quickly addressed in the final version.

---

### Official Review · Reviewer_yEiY · 2023-07-06

**Soundness:** 3 good
**Presentation:** 2 fair
**Contribution:** 3 good
**Rating:** 7
**Confidence:** 3

**Summary:**

This paper presents a method for active causal discovery with "multi-fidelity" oracles.
Here multi-fidelity refers to the option to request outcome labels for a given experiment from a set of oracles with different quality levels.
The method extends causal experimental design methods where experiments are defined by a given variable and value to this (to my knowledge) novel problem setting.
As such, their method views an experiment as consisting of a triple: (variable, value, fidelity) and defines a mutual information objective for experiment selection.
The method also includes a "cascading fidelity model" accounting for information shared between fidelity models.
They propose an "$\epsilon-$submodular" method for multiple-intervention experiments.
They empirically validate their results.

**Strengths:**

The paper makes novel non-obvious contributions, which I have summarized above. To my knowledge this is the first exploration of multi-fidelity active causal discovery, and they have provided a clever and principled solution. I cannot speak to the correctness of the improved greedy method, but the rest of the paper does not seem to have major flaws.

**Weaknesses:**

## Minor Concerns

I really only have minor concerns, which I hope will help to improve an already excellent paper.

1. **Writing Quality**. As an example, "which is fundamental for many real-world applications, ranging from health caring, education, to drug discovery, and protein synthesis," is a little awkward and could be improved as, "which is fundamental for many real-world applications, including health care, education, drug discovery, and protein synthesis." Another example would be  that "In specific" can be replaced by "Specifically" or "In particular." Another would be starting a sentence with a citation should use the \citet{} function rather than \cite{} or \citep{}, such as on line 34. There are other examples and I would suggest using a tool like Grammarly. I do not find this to be a deal breaker, though, and some sections are written very well (e.g., 2.1).

1. **Motivating Example**. The example of a high-fidelity experiment as an actual patient outcome vs. a low-fidelity experiment as a  simulated patient outcome is a little weak. At least need to explain why the simulator could be lower-fidelity. If the simulator has domain knowledge, why not incorporate that knowledge into the causal model rather than try to infer that knowledge? Is it a black box? As alternatives, maybe consider spacial or temporally measured outcomes as examples of low vs. high-fidelity. For example, perhaps a lower spatial or temporal resolution measurement vs. a higher spatial or temporal measurement? A satellite image at 10-mile resolution vs. 1-mile resolution. An MRI from a 1T machine vs. a 7T machine? Maybe alternatively, a radiologist's assessment of malignancy vs. a biopsy result?

1. **line 91**. covariance -> variance

1. **f**. You use $f$ to define the SCM functions and the acquisistion function, which is a little confusing.

1. **Active Causal Discovery**. Since you work on identifying the adjacency matrix and the  SEM parameters, I'd use causal experimental design. Only because causal discovery has focused more on just identifying the adjacency matrix. But, active causal discovery does have a nice ring to it.

1. **Duplicated references**. There are references with multiple but distinct entries.

1. **Licence** is a bit arbitrary as a name.



**Questions:**

What issues would arise if the cost $\lambda$ and the mutual information $I$ are not on similar scales?

Can you know the scale of the mutual information? Is your mutual information estimator reliable?


**Limitations:**

Yes

---

> ### Author Rebuttal · Authors · 2023-08-09
>
> For reviewer yEiY:
>
> Thanks for your detailed comments.
>
> > Reviews: As an example, "which is fundamental for many real-world applications, ranging from health caring, education, to drug discovery, and protein synthesis," is a little awkward and could be improved as, "which is fundamental for many real-world applications, including health care, education, drug discovery, and protein synthesis." Another example would be that "In specific" can be replaced by "Specifically" or "In particular." Another would be starting a sentence with a citation should use the \citet{} function rather than \cite{} or \citep{}, such as on line 34. There are other examples and I would suggest using a tool like Grammarly. I do not find this to be a deal breaker, though, and some sections are written very well (e.g., 2.1).
>
>
> Thanks for pointing out the writing problems, In the final version, we will carefully revise them according to your comments.
>
>
>
> > Reviews: Motivating Example. The example of a high-fidelity experiment as an actual patient outcome vs. a low-fidelity experiment as a simulated patient outcome is a little weak. At least need to explain why the simulator could be lower-fidelity. If the simulator has domain knowledge, why not incorporate that knowledge into the causal model rather than try to infer that knowledge? Is it a black box? As alternatives, maybe consider spacial or temporally measured outcomes as examples of low vs. high-fidelity. For example, perhaps a lower spatial or temporal resolution measurement vs. a higher spatial or temporal measurement? A satellite image at 10-mile resolution vs. 1-mile resolution. An MRI from a 1T machine vs. a 7T machine? Maybe alternatively, a radiologist's assessment of malignancy vs. a biopsy result?
>
> Thanks for this comments. In the patient example, we believe the simulator can be less accurate, since the models used for simulation can be not perfect, it may contain different approximation errors. However, the simulation method may cost little, since we do not have to make experiments on the real people.
>
> We believe the motivating example of spacial or temporally measured outcomes is also very interesting, we will definitely add it into our final paper.
>
>
>
> > Reviews: line 91. covariance -> variance. You use f to define the SCM functions and the acquisistion function, which is a little confusing.
>
> In the final version, we will revise these inappropriate aspects accordingly.
>
>
>
> > Reviews: Active Causal Discovery. Since you work on identifying the adjacency matrix and the SEM parameters, I'd use causal experimental design. Only because causal discovery has focused more on just identifying the adjacency matrix. But, active causal discovery does have a nice ring to it. Duplicated references. There are references with multiple but distinct entries. Licence is a bit arbitrary as a name.
>
> Thanks for this comment.  We will change ACD to causal experimental design in the final version. For the references, will double check them to make the paper more clear. For the model name Licence, we will change it in the final version.
>
>
>
> > Reviews: What issues would arise if the cost and the mutual information are not on similar scales?
>
> For the same oracle, we just need to obtain the ranking of the mutual information for different intervention variable and value pairs. The absolute value may be not that important.
> Actually, in our experiments, we find that the mutual information term is always in [0.05, 0.2], which does not differ too much from the cost.
>
>
>
> > Reviews: Can you know the scale of the mutual information? Is your mutual information estimator reliable?
>
> The scale of the mutual information term is always in [0.05, 0.2]. We can not ensure that the estimator is completely correct. The reliability of the estimator comes from the Law of Large Numbers.

---

> > ### Comment · Reviewer_yEiY · 2023-08-15
> > **Thank you for your reply.**
> >
> > Thank you for your reply and taking the time to review my comments. Again, I think this is a sufficiently novel (and correct) contribution for acceptance at NeurIPS and I will maintain my score.
> >
> > Regarding the last 2 points, I do think that because the mutual information estimate will generally depend on the estimator used, whereas the cost is something fixed in each setting, a discussion of the potential issues that this may lead to should be included in the limitations section of the paper.
> >
> > Good luck.

---

> > > ### Author Response · Authors · 2023-08-15
> > > **Thanks for the response**
> > >
> > > Dear Reviewer yEiY,
> > >
> > > Thanks very much for your feedback, we will definitely incorporate the discussion on the potential issues of the points you mentioned in the final version.
> > >
> > > Thanks

---

> ### Author Response · Authors · 2023-08-13
> **Rebuttal by Authors**
>
> Dear reviewer yEiY, thanks again for your insightful comments, which, we believe, are very important to improve our paper.  In the rebuttal and submitted one-page pdf, we have tried to answer your questions one by one. If you have further questions, we are very happy to discuss them.

---

### Official Review · Reviewer_am1k · 2023-07-07

**Soundness:** 3 good
**Presentation:** 3 good
**Contribution:** 3 good
**Rating:** 5
**Confidence:** 4

**Summary:**

In this manuscript, the authors propose a novel method for active causal discovery (ACD) in a multi-fidelity setting, where experiments with different cost and accuracy can be designed and performed for network intervention for the purpose of accurately learning the causal structure.
For this multi-fidelity ACD (MFACD), the manuscript proposes License - a Bayesian framework for "Multi-fidelity
active learning for causal discovery".
License adopts an information-theoretic acquisition function motivated by the popular Bayesian Active Learning by Disagreement (BALD) for the prediction of the best experiment, which consists of selecting the node for intervention, its value, and the fidelity.
Furthermore, a cascaded fidelity model is proposed to capture the correlations between the experimental outcomes across different fidelities.


**Strengths:**

The problem of ACD is widely studied in a number of fields, especially so in life science to uncover the regulatory relations among genes.
Network interventions are routinely performed in labs to unveil the causal structure of the network, hence designing experiments that can accurately uncover the causal relations among nodes and minimizing the overall cost of designing and carrying out the experiments is an important problem.
Multi-fidelity experiments may often be considered in practical settings to maximize the value of information acquired by the experiments based on a given experimental budget, and the MFACD problem tackled in this manuscript is therefore of practical importance as well as theoretical interest.

The proposed acquisition function for predicting the best experiment is reasonable, as its acquisition function is motivated by the widely popular active learning scheme - BALD - which is normalized by the experimental cost to assess the cost-normalized value of information that may be attained by a given experiment.

The evaluation results in the manuscript shows that the proposed License method may have potential advantages over some alternative schemes.


**Weaknesses:**

While the problem being tackled in this current study is very interesting and is both of practical importance as well as of theoretical significance, there are several major concerns regarding the current manuscript as summarized below.

Although the concept of designing multi-fidelity experiments to maximize the accuracy of ACD while minimizing the overall experimental cost is important, the type of multi-fidelity experiments considered in this work is not well motivated.

For example, the authors assume that there are multiple SCMs with different fidelities.
However, it is unclear how the accuracy of the SCM at a specific fidelity is defined and what parameters govern this accuracy.

And in practice, how does one construct or obtain multiple SCMs at different fidelities?
Where does the different experimental cost arise, if all SCMs are of the same form with different fidelities?
Since the manuscript doesn't give any practical example of how such multi-fidelity models may be constructed or accessed and how (and why) they may differ in terms of the acquisition cost (of the experimental outcomes), it is unclear whether the proposed method and the derived results may be applicable in any practical setting.

For example, in drug discovery, it is typical to assess the efficacy of a drug candidate against a specific target using multi-fidelity models, starting from a fast ML surrogate model, and then to a high-throughput docking model, then use molecular dynamics simulations to assess the binding affinity between the drug candidate and the target.
Naturally, there are differences in terms of the computational cost as well as the accuracy, hence decisions need to be made as to which model should be incurred to maximize the value of information on the computational investment made.
Unfortunately, the current manuscript does not provide any real example to motivate the proposed method and its problem setting, making it difficult to see how the proposed approach might be applied.

Furthermore, it is natural that multi-fidelity experimental outcomes will be correlated to each other (at least to a certain extent) and it is a good idea to explore this property when learning/constructing the multi-fidelity models and utilizing them in MFACD experimental design.
However, in the current study, it is unclear how leveraging such correlations across different fidelities actually translates to the efficacy of the designed experiments and/or savings in terms of the experimental cost.

The description of "extension to multi-target intervention" is also somewhat confusing.
Based on the technical descriptions in 3.3. and the equations (7)-(9), it appears that while the experiments (or interventions) are designed in sequence (hence referred to as "a series of experiments"), the interventions are being applied to the network "simultaneously".
However, as the actual "multi-target intervention" setting being investigated in the current study is not clearly described (or described in a potentially misleading manner), there is some ambiguity in the experimental setting which needs to be clarified.

Theorem 3 states that "For any two experiments es and et, if the corresponding samples xs and xt are
ε-independent given φM, {es, et} and D, then I(·; φM |·, D) is ε-submodular."
However, how can one actually guarantee whether a given pair of samples xs and xt are indeed ε-independent, considering the unknown and complex causal structure underlying these samples?
This needs to be elaborated more clearly.

Finally, the experimental results shown in the current study are very limited, and far from convincing to demonstrate the general applicability of the proposed License method to general MFACD problems.
Furthermore, the multi-fidelity SCMs used in these evaluations, the number of fidelities considered, and the respective experimental/intervention costs are not clearly described, which makes it difficult to understand how the overall evaluations have been carried out.




**Questions:**

Please see the questions and concerns in the above section.

**Limitations:**

The broader impacts of the current study and its limitations are not discussed in the current manuscript.

---

> ### Author Rebuttal · Authors · 2023-08-09
>
> For reviewer am1k:
>
> > Concerns on the multiple SCMs.
>
> 1. For the real-world scenarios, multi-fidelity oracles are very common, for example (as mentioned in the paper), to investigate the drug-disease causal relations, one can either conduct clinical tests (high cost but more accurate) or build simulators to obtain the medicine effects on the patients.
>
> 2. In our study, it is very hard to find public datasets, which can be used for causal discovery and simultaneously contain multi-fidelity oracles. Thus, we firstly find the commonly used causal discovery datasets. And then regard the ground truths in these datasets as the highest-fidelity oracle, and simulate multiple low-fidelity oracles by adding Gaussian noises on the ground truths.
>
> 3. The specific formulation of the above experiment settings are presented as follows: for a given intervention $(j,v)$, suppose we have $M$ oracles $\\{\phi_{1},\phi_{2},...,\phi_{M}\\}$, then the experiment results $\\{x_{j,v,1},x_{j,v,2},...,x_{j,v,M}\\}$ are specified as follows:
> >$$x_{j,v,m} = x_{j,v,M} + \delta_m,$$
> >$$\delta_m \sim N(0,\sigma_m),$$
>
> where $x_{j,v,M}$ is the ground truth, which can be directly obtained from the datasets.  Since $x_{j,v,m}$ is correlated with $x_{j,v,M}$ by the first line, their underlying oracles $\phi_{m}$ and $\phi_{M}$ are correlated in our simulation.  In our experiment, we set $\sigma_1 > \sigma_2 > ... >\sigma_M = 0$. Suppose the cost of $\phi_{m}$ as $\lambda_m$, then we set $\lambda_1 < \lambda_2 <...<\lambda_M$.
>
> 4. To demonstrate that our model is generally effective for different cost- and noisy-levels. We conduct experiments based on different sets of oracles.
> In specific, the experiments are conducted based on the settings presented in Table 1 of the submitted one-page pdf. The results are presented in Figure 3 of the submitted one-page pdf. From the results, we can see that our model can always perform better than the baselines on different sets of oracles.
>
> 5. The above experiment settings to simulate the oracles with different fidelities are very common in the field of multi-fidelity optimization [1-3]. We exactly follow the common practice in the multi-fidelity domain. We believe this is understandable, since it is hard to find public datasets with multi-fidelity oracles, but this research direction is meaningful and need to be improved.
>
> > Concerns on the effectiveness of modeling oracle correlations.
>
> To study whether the correlation modeling between different oracles are necessary, we first build a variant of our model by regarding different oracles as independent components, that is, removing the links between different $\phi$'s in Figure 1, and then compare our model with such variant. The results are presented in Table 3 of the submitted pdf. We can see, in most cases, our model can achieve better performance than its variant without modeling the correlations between different oracles.
>
> > Concerns on the description of "extension to multi-target intervention".
>
> 1. Multi-target intervention aims to (1) firstly determine a set of intervention variables, and then (2) leverage them to simultaneously query the oracles to obtain the experiments results.
>
> 2. For (1), we leverage equation (7) and (8) to determine the variables one by one within a budget $C$. This step only aims to determine a batch of variables. We still do not leverage them to conduct experiments. We use the greedy method (*i.e.,* determine the variables in sequence), because it has solid theoretical grantees and has been widely used before.
>
> 3. For (2), once the batch of variables has been determined, we use them to simultaneously intervene the oracles and obtain the experiment results.
>
> > Concerns on Theorem 3.
>
> 1. In the previous single-fidelity studies, the theoretical foundation behind the greedy method is "T1: if $x_s$ and $x_t$ are independent given $\Phi_M$, then the greedy method can be bounded." (see [4-6]).
>
> 2. In our multi-fidelity study, we find that, "$x_s$ and $x_t$ are actually not independent given $\phi_M$", which means that the theoretical foundation of the greedy method fails.
>
> 3. To extend T1 to multi-fidelity settings, we introduce the concepts of $\epsilon$-independent and $\epsilon$-submodular, and propose Theorem 3.
>
> 4. $\epsilon$-independent is a concept describing the data independent character. When $\epsilon \rightarrow 0$, $\epsilon$-independent equals to independent. When $\epsilon \rightarrow \infty$, xs and xt can be highly correlated. For any dataset, there should be always an $\epsilon$ which ensures that "$x_s$ and $x_t$ $\epsilon$-independent" (although $\epsilon$ can be a very large value).
>
> 5. Actually, theorem 3 is a general version of Theorem B.2 in [4]. When $\epsilon = 0$, our theorem reduces to Theorem B.2 in [4] as well.
>
> 6. The constraint in equation (10) aims to introduce inductive bias on the independent characters of the real data.
>
> > Concerns on the experiments.
>
> To improve our experiments, we conduct a large amount of further experiments on the additional evaluation metrics, different oracle settings, influence of the regularization coefficient $\lambda$ under line 172 in the appendix and ablation studies (see the submitted pdf).
>
> > The broader impacts of the current study and its limitations are not discussed in the current manuscript.
>
> Actually, we have discussed the impacts of the current study in the Appendix. For the limitations, we will add them in the final version.
>
> **References**
>
> [1] Batch Multi-Fidelity Bayesian Optimization with Deep Auto-Regressive Networks.
>
> [2] Batch Multi-Fidelity Active Learning with Budget Constraints.
>
> [3] Deep multi-fidelity active learning of high-dimensional outputs.
>
> [4] Interventions, where and how? experimental design for causal models at scale.
>
> [5] Batchbald: Efficient and diverse batch acquisition for deep bayesian active learning.
>
> [6] Abcd-strategy: Budgeted experimental design for targeted causal structure discovery.

---

> > ### Comment · Reviewer_am1k · 2023-08-18
> >
> > I would like to thank the authors for their careful and thorough responses to my review comments.
> > The point-by-point response above has addressed many of the previous concerns raised in my review.
> >
> > I understand that there exist practical limitations on how to evaluate and demonstrate the proposed active causal discovery scheme under a multi-fidelity setting and that the authors have decided to perform the experiments based on benchmark data commonly used for evaluating causal discovery techniques and also based on simulated data under reasonable modeling assumptions.
> > However, it is also important to recognize their limitations (as mentioned in my original review comments) and discuss them in the manuscript.
> > To a certain extent, this may be done by incorporating the arguments/explanations in the above rebuttal into the revised manuscript and appendix.
> >
> > Additionally, I still feel that the "practical" motivation for the proposed method and the current study could be further strengthened, as the problem and the work themselves are theoretically interesting but as the current evaluation setting does not appear to be strongly connected to "real" use cases.
> >
> > Finally, the additional experiments the authors have performed for the rebuttal add significant value to the current study, as they address a number of concerns raised in my original review and also highlight the merits of the proposed method more clearly.
> > I hope these results will be integrated into the main text as well as the appendix of this work.
> >
> > Overall, I would be happy to raise my evaluation score thanks to the clarifications and additional experimental results provided by the authors.
> > Thank you again.

---

> > > ### Author Response · Authors · 2023-08-18
> > > **Thanks for the response**
> > >
> > > Dear reviewer am1k,
> > >
> > > Thanks so much for your feedback. We will definitely incorporate the arguments/explanations about the experiment setup into our final version.  In addition, we will also present more strong motivations and the added experiments in the final paper.
> > >
> > > Thanks again.

---

> ### Author Response · Authors · 2023-08-13
> **Rebuttal by Authors**
>
> Dear reviewer am1k, thanks again for your detailed comments, which, we believe, are very important to improve our paper.
>
> In the rebuttal and submitted one-page pdf, we try to clarify the experiment settings, multi-target intervention, and Theorem 3 in detail.
>
> To alleviate your concerns, we also conduct a large number of experiments, including verifying whether the correlation of oracle modeling is important, experiments on different oracle settings, the influence of the key hyper-parameters, and more evaluation metrics.
>
> If you have further questions, we are very happy to discuss them. We really hope our efforts can alleviate your concerns.

---

### Official Review · Reviewer_HBUA · 2023-07-14

**Soundness:** 3 good
**Presentation:** 2 fair
**Contribution:** 2 fair
**Rating:** 6
**Confidence:** 3

**Summary:**

This paper proposes an approach for Bayesian active causal discovery with multi-fidelity observations. This approach has two main components: (1) a cascade probabilistic model handling the correlation between fidelity levels, and (2) a cost-aware information-theoretic acquisition function, which quantifies the mutual information (per unit of cost) between the causal graph and an observation at a given input location and fidelity level. An extension to the multi-target setting, where multiple nodes can be intervened simultaneously, is considered. In such a setting, a seemingly natural choice is to select the nodes to be intervened in a greedy fashion by iteratively maximizing the mutual information. However, submodularity does not hold, so the classical approximation guarantee is not obtained. To alleviate this issue, the notions of $\epsilon$-independence and $\epsilon$-submodularity. The high-level idea is to select points with low mutual information so that submodularity holds approximately. The proposed approach is shown to significantly outperform various baselines across three test problems.

**Strengths:**

1. The problem considered by this paper is of significant practical relevance.
2. The proposed approach is technically sound.
3. This paper is very well written overall.

**Weaknesses:**

My only significant concern about this paper is its empirical evaluation, as there are several details I could not figure out. See questions 1 and 2 below.

**Questions:**

1. How is $\epsilon$ chosen? More generally, how were all the algorithm hyperparameters chosen?

2. I believe some of the datasets used are, in principle, not multi-fidelity. How was this addressed?

3. It would be helpful to include figures showing the causal graph structure of the test problems considered.

4. In the context of causal optimization, the term "target" is often used to denote the node to be optimized. Thus, I believe the term "multi-target" is misleading in its use here. Perhaps "batch" would be more appropriate.

**Limitations:**

The potential negative societal impact was adequately discussed (in the supplement). The authors also mention some interesting ways to improve their method. However, I believe two important limitations were not addressed:

1. Computational cost of the proposed approach vs. the standard approach.
2. Robustness to the choice of the algorithm hyperparameters.

---

> ### Author Rebuttal · Authors · 2023-08-09
>
> For reviewer HBUA:
>
> Thanks for your comments. In the following, we try to alleviate your concern one by one:
>
> >  Question 1: How is episilon chosen? More generally, how were all the algorithm hyper parameters chosen?
>
> As can be seen in Appendix D, when training our model, the constraint involving $\epsilon$ is converted to a regularization to the objective. Actually, $\epsilon$ play similar roles as the regularization coefficient, that is, $\lambda$ in the equation below line 184. For $\lambda$ as well as all the other hyper-parameters, we determine them by grid search based on the validation set.
>
> > Question 2: I believe some of the datasets used are, in principle, not multi-fidelity. How was this addressed?
>
> 1. In our experiments, we regard the original ground truth as the highest fidelity oracle. Then we add Gaussian noises on the highest fidelity oracle to simulate the other oracles with different fidelities. We add more uncertain noises on the lower fidelity oracles to make it less accurate. At last, we manually set the costs of the oracles with different fidelities to ensure that "higher fidelity oracles cost more than the lower ones".
>
> 2. The specific formulations of the above experiment settings are presented as follows:
>
> In our experiments, we follow the common practice to simulate the oracles with different fidelities as follows:
>
> For a given intervention $(j,v)$, suppose we have $M$ oracles $\\{\phi_{1},\phi_{2},...,\phi_{M}\\}$, then the experiment results $\\{x_{j,v,1},x_{j,v,2},...,x_{j,v,M}\\}$ are specified as follows:
> $$
> x_{j,v,m} = x_{j,v,M} + \delta_m,
> $$
> $$\delta_m \sim N(0,\sigma_m),$$
> where $x_{j,v,M}$ is the ground truth, which can be directly obtained from the datasets.  Since $x_{j,v,m}$ is correlated with $x_{j,v,M}$ by the first line, their underlying oracles $\phi_{m}$ and $\phi_{M}$ are correlated in our simulation.  In our experiment, we set $\sigma_1 > \sigma_2 > ... >\sigma_M = 0$. Suppose the cost of $\phi_{m}$ as $\lambda_m$, then we set $\lambda_1 < \lambda_2 <...<\lambda_M$.
>
>
> 3. To demonstrate that our model is generally effective for different cost- and noisy-levels. We conduct experiments based on different sets of oracles.
>    In specific, the experiments are conducted based on the settings presented in Table 1 of the submitted one-page pdf. The results are presented in Figure 3 of the submitted one-page pdf. From the results, we can see that our model can always perform better than the baselines on different sets of oracles.
>
>
> 4. The above experiment settings to simulate the oracles with different fidelities are very common in the field of multi-fidelity optimization [1-3].
>
>
> We will definitely add the above experiment settings in the final paper.
>
>
> > Question 3: It would be helpful to include figures showing the causal graph structure of the test problems considered.
>
> Following your advice, we have added many examples of the test causal graph structures in Figure 4 of the submitted one-page pdf.
>
>
> >  Question 4: In the context of causal optimization, the term "target" is often used to denote the node to be optimized. Thus, I believe the term "multi-target" is misleading in its use here. Perhaps "batch" would be more appropriate.
>
> Thank you for your comments. In the final version, we will revise these inappropriate terminations.
>
> > Limitation 1: Computational cost of the proposed approach vs. the standard approach should be addressed.
>
> Actually, we have already compared the computation costs between our model and the baselines in Appendix H.2.
>
>
> > Limitation 2: Robustness to the choice of the algorithm hyper-parameters.
>
> We have conducted many experiments on the influence of the hyper-parameters, for example, the regularization coefficient $\lambda$ (see Figure 2 of the submitted one-page pdf), the oracle cost- and noisy-levels (see Figure 3 of the submitted one-page pdf), the DAG regularization coefficient $\beta$ (see Figure 3(b) in the main paper). From the results, we find that the model performance is robust to some parameters like $\lambda$, but may be sensitive to the other hyper-parameters like $\beta$.
>
>
>
>
> **References**
>
> [1] Li, S., Kirby, R., & Zhe, S. (2021). Batch Multi-Fidelity Bayesian Optimization with Deep Auto-Regressive Networks. *Advances in Neural Information Processing Systems*, *34*, 25463-25475.
>
> [2] Li, S., Phillips, J. M., Yu, X., Kirby, R., & Zhe, S. (2022). Batch Multi-Fidelity Active Learning with Budget Constraints. *Advances in Neural Information Processing Systems*, *35*, 995-1007.
>
> [3] Li, S., Kirby, R. M., & Zhe, S. (2020). Deep multi-fidelity active learning of high-dimensional outputs. *arXiv preprint arXiv:2012.00901*.

---

> > ### Comment · Reviewer_HBUA · 2023-08-14
> > **Post-rebuttal follow-up by Reviewer HBUA**
> >
> > Dear authors,
> >
> > Thank you for your thorough response. Several of my concerns have been adequately addressed. However, after learning that multi-fidelity observations were obtained by adding different levels of Gaussian noise, I am not convinced that the current empirical evaluation is representative of real-world problems, where observations at different fidelity levels are typically biased in very complex ways. I consider this a major concern, so I have decided to lower my score to 4.
> >
> > Best wishes,
> >
> > Reviewer HBUA

---

> > > ### Author Response · Authors · 2023-08-14
> > >
> > > Thanks for your response:
> > >
> > >
> > > (1) We admit that simulation based studies can not perfectly represent the real-world settings. However, in the field of multi-fidelity domain, simulation is a quite common strategy [1-6], since it is hard to find public available datasets.
> > >
> > > (2) We leverage Gaussian noises to simulate the low-fidelity oracles, since they widely exist in real-world scenarios.
> > >
> > > (3) We have conducted multiple different oracle settings (see the submitted one-page pdf) to demonstrate that the improvement of our model is not at random.
> > >
> > > (4) To alleviate your concern, we further conduct the experiment by building the oracles based on another type of method. In specific, we use two neural network (say A and B) to firstly learn the ground truth separately, and then use the learned models as the lower-fidelity oracles. The numbers of parameters of A and B are NA and NB (NA > NB), respectively. The accuracy of A is higher than B, and therefore, we regard A as a higher fidelity oracle.  We regard the normalized time cost for inferring the experiment results as the oracle cost. Since NA > NB, the time cost for A is large than B.
> > >
> > > - For the experiments of ER Graph
> > >
> > > More details of the neural networks:
> > >
> > > | Fidelity | Number of Parameters | Accuracy (oracle fidelity-level) | Time Cost (oracle cost) |
> > > | -------- | -------------------- | -------------------------------- | ----------------------- |
> > > | A (High) | 420                  | 93.22%                           | 29.68 ms                |
> > > | B (Low)  | 210                  | 91.37%                           | 18.46 ms                |
> > >
> > > Experiment results:
> > >
> > > | Mode        | SHD ↓     | AUPRC (%) ↑ | MSE (%) ↓ |
> > > | ----------- | --------- | ----------- | --------- |
> > > | AIT-REAL    | 24.25     | 16.50       | 4.49      |
> > > | AIT-RANDOM  | 25.75     | 17.97       | 4.21      |
> > > | CBED-REAL   | 27.75     | 20.14       | 5.64      |
> > > | CBED-RANDOM | 22.75     | 15.98       | 3.42      |
> > > | Licence     | **14.75** | **30.41**   | **2.12**  |
> > >
> > >
> > >
> > > - For the experiments of SF Graph
> > >
> > > More details of the neural networks:
> > >
> > > | Fidelity | Number of Parameters | Accuracy (oracle fidelity-level) | Time Cost (oracle cost) |
> > > | -------- | -------------------- | -------------------------------- | ----------------------- |
> > > | A (High) | 420                  | 93.20%                           | 44.53 ms                |
> > > | B (Low)  | 210                  | 91.52%                           | 30.47 ms                |
> > >
> > > Experiment results:
> > >
> > > | Mode        | SHD ↓     | AUPRC (%) ↑ | MSE (%) ↓ |
> > > | ----------- | --------- | ----------- | --------- |
> > > | AIT-REAL    | 27.25     | 16.44       | 3.89      |
> > > | AIT-RANDOM  | 27.50     | 21.65       | 3.67      |
> > > | CBED-REAL   | 25.50     | 24.48       | 4.88      |
> > > | CBED-RANDOM | 24.75     | 22.53       | 6.05      |
> > > | Licence     | **18.50** | **25.78**   | **2.99**  |
> > >
> > >
> > >
> > > **References**
> > >
> > > [1] Batch Multi-Fidelity Bayesian Optimization with Deep Auto-Regressive Networks.
> > >
> > > [2] Batch Multi-Fidelity Active Learning with Budget Constraints.
> > >
> > > [3] Deep multi-fidelity active learning of high-dimensional outputs.
> > >
> > > [4] Interventions, where and how? experimental design for causal models at scale.
> > >
> > > [5] Batchbald: Efficient and diverse batch acquisition for deep bayesian active learning.
> > >
> > > [6] Abcd-strategy: Budgeted experimental design for targeted causal structure discovery.

---

> > > ### Author Response · Authors · 2023-08-18
> > > **Further rebuttal**
> > >
> > > Dear Reviewer HBUA:
> > >
> > > We truly appreciate your feedback. Your comments and responses really have improved our paper a lot.  Here, we would like to further clarify our work for your consideration.
> > >
> > > To begin with,  we would like to highlight our major contributions:  (1) we believe this paper makes a first step towards multi-fidelity active causal discovery. (2) To solve this problem, we design a novel model for intervention conduction. (3) Our paper also has theoretical contributions. In specific, we find that the theories behind the traditional greedy method may not work, thus we develop many novel theories to extend the previous theories.
> > >
> > > Indeed, we can not ensure that our simulation method completely follows the real-world settings, but this is a common practice in the multi-fidelity domain, and maybe this is the best effort we can try to solve this novel problem.  As mentioned by reviewer Xcc7, simulation-based experiment settings could be understandable practical limitations. We will definitely incorporate the limitations of the simulation-based method into our final paper.
> > >
> > >
> > > Despite the above aspects, to alleviate your concerns about the simulation settings as much as possible, we have made two significant efforts: (1) for the Gaussian-based simulation, we experiment on six different settings (see the submitted one-page pdf). By this operation, we would like to demonstrate that the improvement of our model is **not sensitive to some specific simulation settings**.  (2) We add another type of method to simulate the lower fidelity oracles(see our rebuttal). By this operation, we would like to demonstrate that our model is **not sensitive to the simulation method**.
> > >
> > > We really hope that our efforts can alleviate your concerns.
> > >
> > > Yours faithfully, Authors

---

> > > > ### Comment · Reviewer_HBUA · 2023-08-19
> > > >
> > > > Dear authors,
> > > >
> > > > Thank you for conducting these additional experiments. My main concerns have been effectively addressed. Thus, I will raise my score to 6.
> > > >
> > > > Best wishes,
> > > >
> > > > Reviewer HBUA

---

> > > > > ### Author Response · Authors · 2023-08-19
> > > > > **Further responses**
> > > > >
> > > > > Dear reviewer HBUA，Thanks so much for your feedback.

---

> ### Author Response · Authors · 2023-08-13
> **Rebuttal by Authors**
>
> Dear reviewer HBUA, thanks again for your significant comments, which can definitely improve our paper.  In the rebuttal and submitted one-page pdf, we try to explain your questions one by one. In addition, we have added a large number of experiments to make our paper more solid.  If you have further questions, we are very happy to discuss them.

---

### Official Review · Reviewer_Xcc7 · 2023-07-26

**Soundness:** 3 good
**Presentation:** 2 fair
**Contribution:** 3 good
**Rating:** 6
**Confidence:** 2

**Summary:**

This paper addresses the problem of active causal discovery with multi-fidelity oracles, where experiments can be done based on different costs, precisions, and reliabilities. The paper formally defines the task of multi-fidelity active causal discovery and proposes a Bayesian framework consisting of a mutual information-based acquisition function and a cascading fidelity model. The paper also extends the framework to the multi-target intervention scenario and introduces a constraint-based fidelity model to validate the greedy method. The effectiveness of the proposed model is demonstrated through extensive experiments.


**Strengths:**

(1) The paper addresses an important and practical problem of active causal discovery with multi-fidelity oracles, which is more realistic than previous single-fidelity settings.
(2) The proposed Bayesian framework, including the mutual information-based acquisition function and the cascading fidelity model, provides a novel and practical solution to the multi-fidelity active causal discovery problem.
(3) The extension to the multi-target intervention scenario and the introduction of the constraint-based fidelity model further enhance the applicability of the proposed model.


**Weaknesses:**

(1) The evaluation and ablation studies for the proposed method are limited, and more comprehensive experiments could be conducted to strengthen the empirical findings.
(2) Have you considered other evaluation metrics besides SHD, AUPRC, and MSE? How does the proposed model compare to other state-of-the-art methods in terms of these metrics?

**Questions:**

Have you considered other evaluation metrics besides SHD, AUPRC, and MSE? How does the proposed model compare to other state-of-the-art methods in terms of these metrics?

**Limitations:**

No potential negative societal impact of their work.

---

> ### Author Rebuttal · Authors · 2023-08-09
>
> For reviewer Xcc7:
>
> Thanks for your overall positive comments on our paper. We try to alleviate your concerns as follows:
>
> > Concerns: More comprehensive experiments could be conducted to strengthen the empirical findings. Have you considered other evaluation metrics besides SHD, AUPRC, and MSE? How does the proposed model compare to other state-of-the-art methods in terms of these metrics?
>
> The reasons why we choose SHD, AUPRC, and MSE as our evaluation metrics:
>
> (1) These metrics are widely used in previous studies in the field of active causal discovery, particularly for the datasets used in our experiments [1-3].
>
> (2) These metrics provide a comprehensive evaluation of the model's performance from different perspectives. The first two metrics assess the accuracy of the learned topological structure, while the last one measures the performance of functional relations.
>
> To alleviate your concerns, we further conduct the following experiments:
>
> 1. More experiments on additional evaluation metrics (see Table 2 of the submitted one-page pdf).
>
> 2. More experiments on the influence of different oracle settings (see Table 1 and Figure 3 of the submitted one-page pdf). For a given intervention $(j,v)$, suppose we have $M$ oracles $\\{\phi_{1},\phi_{2},...,\phi_{M}\\}$, then the experiment results $\\{x_{j,v,1},x_{j,v,2},...,x_{j,v,M}\\}$ are specified as follows:
> > $$
> > x_{j,v,m} = x_{j,v,M} + \delta_m ,
> > $$
>  > $$\delta_m \sim N(0,\sigma_m) ,$$
>
> where $x_{j,v,M}$ is the ground truth, which can be directly obtained from the datasets.  Since $x_{j,v,m}$ is correlated with $x_{j,v,M}$ by the first line, their underlying oracles $\phi_{m}$ and $\phi_{M}$ are correlated in our simulation.  In our experiment, we set $\sigma_1 > \sigma_2 > ... >\sigma_M = 0$. Suppose the cost of $\phi_{m}$ as $\lambda_m$, then we set $\lambda_1 < \lambda_2 <...<\lambda_M$.
>
> To demonstrate that our model is generally effective for different cost- and noisy-levels. We conduct experiments based on different sets of oracles.
>
> In specific, the experiments are conducted based on the settings presented in Table 1 of the submitted one-page pdf. The results are presented in Figure 3 of the submitted one-page pdf. From the results, we can see that our model can always perform better than the baselines on different sets of oracles.
>
>
> 3. More experiments on the regularization coefficient $\lambda$ under line 172 in the appendix (see Figure 2 of the submitted one-page pdf).
>
> 4. More experiments on ablation studies (see Table 3 of the submitted one-page pdf).
>
>
> We sincerely thank you for your time to review our paper and give positive comments on it. We are glad to answer further questions, which, we believe, can definitely improve our paper.
>
>
> **References**
>
> [1] Tigas, P., Annadani, Y., Jesson, A., Schölkopf, B., Gal, Y., & Bauer, S. (2022). Interventions, where and how? experimental design for causal models at scale. *Advances in Neural Information Processing Systems*, *35*, 24130-24143.
>
> [2] Scherrer, N., Bilaniuk, O., Annadani, Y., Goyal, A., Schwab, P., Schölkopf, B., ... & Ke, N. R. (2021). Learning neural causal models with active interventions. *arXiv preprint arXiv:2109.02429*.
>
> [3] Zheng, X., Aragam, B., Ravikumar, P. K., & Xing, E. P. (2018). Dags with no tears: Continuous optimization for structure learning. *Advances in neural information processing systems*, *31*.

---

> > ### Comment · Reviewer_Xcc7 · 2023-08-17
> >
> > I appreciate the authors responding to my concerns; specifically, I appreciate the authors including a large number of additional experiments. I think this is a significant contribution. I will maintain my score.

---

> > > ### Author Response · Authors · 2023-08-17
> > > **Rebuttal by Authors**
> > >
> > > Thanks very much for your response.

---

> ### Author Response · Authors · 2023-08-13
> **Rebuttal by Authors**
>
> Dear reviewer Xcc7, thanks again for your important comments. In the rebuttal and submitted one-page pdf, we have followed your advice to conduct a large number of additional experiments. If you have further questions, we are happy to discuss them.

---

### Author Rebuttal · Authors · 2023-08-09

Dear reviewers:

   Thanks for your detailed reviews.  Additional tables and figures that mentioned in rebuttals are shown in the submitted one-page pdf.

---

### Author Response · Authors · 2023-08-17
**Further Rebuttal by Authors**

Dear Reviewers,

Thanks so much for your hard work in reading and reviewing our paper. All your comments are really helpful to improve our paper, and we appreciate them.

Here, we would like to further explain the following points:

**Our major contributions**:  (1) we believe this paper makes a first step towards multi-fidelity active causal discovery.  (2) When solving this problem, we design a novel Bayesian cascaded model to capture the oracle correlations.  (3) Our paper also has theoretical contributions. In specific, we find that the theories behind the traditional greedy method may not work, thus we develop many novel theories to extend the previous theories.


**Experiment settings**:  for the academic community, it is really hard to find multi-fidelity datasets. Most of the previous work leverage simulation methods (see the reference in the previous rebuttal).  We really appreciate HBUA for pointing out this problem. We will definitely follow your advice to find a more realistic method to build multi-fidelity data or collect real-world datasets.

**More explorations on the experiment settings**:  Despite the above considerations, we still want to follow the review's comments to make our paper more convincing. We have tried another strategy to build the oracles with different fidelities (i.e., learning surrogate models in the rebuttal to reviewer HBUA). The results suggest that our model can still perform better.

**We really hope that, as the first work towards multi-fidelity active causal discovery, our community can value more on its contributions.**

We greatly appreciate all feedback and are committed to participating in ongoing discussions, particularly when reviewers present comments or questions. Your input is highly valued and helps us improve our work.

---

### Decision · Program_Chairs · 2023-09-21

**Decision:**

Accept (poster)

**Comment:**

The authors consider the problem of causal discovery when the experiments may be performed at multiple fidelities. The reviewers agree that the problem is important and the solution proposed would be valuable to the NeurIPS community. The paper was also improved through the rebuttal process and the authors are strongly encouraged to incorporate this feedback into the final version.